
# The Δ¹⁷O and δ¹⁸O values of simultaneously collected atmospheric nitrates from anthropogenic sources – Implications for polluted air masses

Martine M. Savard[1*], Amanda Cole[2], Robert Vet[2], Anna Smirnoff[1]

[1] Geological Survey of Canada (Natural Resources Canada), 490 de la Couronne, Québec (QC), G1K 9A9, Canada
[2] Air Quality Research Division, Environment and Climate Change Canada, 4905 Dufferin St., Toronto (ON), M3H 5T4, Canada

*Correspondence to*: Martine M. Savard (martinem.savard@canada.ca)

**Abstract.** There are clear motivations for better understanding the atmospheric processes that transform nitrogen (N) oxides (NO$_x$) emitted from anthropogenic sources into nitrates (NO$_3^-$), two of them being that NO$_3^-$ contributes to acidification and eutrophication of terrestrial and aquatic ecosystems, and particulate nitrate may play a role in climate dynamics. For these reasons, oxygen isotope ratios (δ¹⁸O, Δ¹⁷O) have been applied to infer the chemical pathways leading to the observed distribution of wet (w-NO$_3^-$), particulate (p- NO$_3^-$), and the sum of p-NO$_3^-$ and gaseous HNO$_3$, while the gaseous form (HNO$_3$)

has never been separately characterized for ¹⁷O. Previous research studies have investigated w-NO$_3^-$, p-NO$_3^-$ or p-NO$_3^-$ + HNO$_3$ from non-polluted or polluted air masses, and inferred seasonal changes in the dominance of oxidation pathways to account for higher δ¹⁸O and Δ¹⁷O values in winter relative to summer. However, none of the polluted air studies collected samples specific to targeted emission sources. Here we have used a wind-sector-based, multi-stage filter sampling system and precipitation collector to simultaneously sample HNO$_3$ and p-NO$_3^-$, and co-collect w-NO$_3^-$, downwind from five different anthropogenic

sources.

Overall, the w- and p-NO$_3$ δ¹⁸O and Δ¹⁷O values show expected differences between cold and warm seasons, but only the Δ¹⁷O values of HNO$_3$ follow this pattern. The HNO$_3$ δ¹⁸O ranges are distinct from the w- and p-NO$_3^-$ patterns. Interestingly, the Δ¹⁷O differences between p-NO$_3^-$ and HNO$_3$ shifts from positive during cold sampling periods to negative during warm periods. The

summer pattern may be due to the presence of nitrates derived from NO$_x$ that has not yet reached isotopic equilibrium with O$_3$ and subsequent differences in dry deposition rates, while the larger proportion of p-NO$_3^-$ formed *via* the N$_2$O$_5$ pathway can explain the fall-winter pattern. Very low p-NO$_3^-$ Δ¹⁷O values observed during warm months may be due to this non-equilibrated NO$_x$, though contribution from RO$_2$ oxidation remains a possibility. Our results show that the isotopic signals of HNO$_3$, w-NO$_3^-$ and p-NO$_3^-$ are not interchangeable and that their differences can further our understanding of NO$_x$ oxidation and

deposition. Future research should investigate all tropospheric nitrate species as well as NO$_x$ to refine our understanding of nitrate worldwide and to develop effective emission reduction strategies.





# 1 Introduction

Anthropogenic $NO_x$ (NO and $NO_2$) emissions are oxidized to nitrate in the atmosphere in the form of gaseous ($HNO_3$), wet (precipitation or w-$NO_3^-$) or particulate (p-$NO_3^-$) forms, $HNO_3$ being one of the main precursors of p-$NO_3^-$. All these species may have detrimental effects on human health and aquatic and terrestrial ecosystems through inhalation, acidification and

excess nitrogen deposition. In addition, aerosols may play a significant role in regional climate dynamics as they interact with clouds and solar radiation (e.g., IPCC, 2013). For these reasons, understanding the chemical processes controlling the transport and fate of atmospheric reactive N is required to help develop effective emission reduction strategies and drive climate models (in the present article, we use *nitrates* to refer to p-$NO_3$, $HNO_3$ and w-$NO_3$ together).

Triple oxygen isotopes ($\delta^{18}O$ and $\delta^{17}O$) may serve for deciphering atmospheric oxidation pathways of $NO_x$ leading to ambient nitrate. Michalski et al. (2003) performed the first measurement of $\delta^{17}O$ values in atmospheric nitrate. The authors found that atmospheric nitrate is highly enriched in $^{18}O$ and $^{17}O$, likely due to the transfer of anomalous oxygen atoms from ozone ($O_3$) via the $NO_x$-ozone photochemical cycle and oxidation to nitrate. During its formation, $O_3$ inherits abnormally high $\delta^{18}O$ and $\delta^{17}O$ values through mass independent fractionation. The specific $\delta^{17}O$ departure from the terrestrial mass dependent

fractionation line, named $^{17}O$ anomaly, is expressed as $\Delta^{17}O = \delta^{17}O - 0.517 \times \delta^{18}O$ (Thiemens, 1999). Further investigations suggested that the $\delta^{18}O$ and $\delta^{17}O$ values of w-$NO_3^-$ and p-$NO_3^-$ reflect several reactions taking place after the atmospheric emission of $NO_x$, *i.e.*, atmospheric oxidation pathways transforming $NO_x$ into secondary products (Hastings et al., 2003; Michalski et al., 2003; Michalski et al., 2004; Morin et al., 2007; Savarino et al., 2007; Alexander et al., 2009). Seasonal $\delta^{18}O$ differences in w-$NO_3^-$ samples (less variable and lower values during summer) have been interpreted to be due to changes in

these chemical pathways (Hastings et al., 2003). Modeling and validation based on sparse existing data provide hope regarding a global understanding of atmospheric nitrate (Alexander et al., 2009), and further measurements need to be done on the ground, particularly at mid-latitudes.

Additional studies dealing with triple oxygen isotope characterizations have addressed questions of methodology (Kaiser et al.,
2007; Smirnoff et al., 2012), transfer of the ozone $^{17}O$ anomaly to atmospheric nitrate (Liang and Yung, 2007; Savarino et al., 2008; Michalski et al., 2014), or sources and chemical pathways of high (Arctic) and low (Taiwan) latitude nitrate (Morin et al., 2008; Guha et al., 2017, respectively). Triple oxygen isotope characterizations of field $NO_3^-$ samples are not yet widespread. The few existing studies have chiefly characterized w-$NO_3^-$ or the sum of p-$NO_3^-$ and $HNO_3$ (Michalski et al., 2004; Morin et al., 2007; Morin et al., 2008; Alexander et al., 2009; Morin et al., 2009; Proemse et al., 2012; Guha et al., 2017), and suggested

these indicators would be useful to trace atmospheric nitrate in water (Kendall et al., 2007; Tsunogai et al., 2010; Dahal and Hastings, 2016). Elliott et al. (2009) measured $\delta^{18}O$, but not $\Delta^{17}O$, in United States CASTNET (Clean Air Status and Trends Network) samples of simultaneously-collected p-$NO_3^-$ and $HNO_3$ as well as in nearby NTN (National Trends Network) precipitation samples. Even rarer are the nitrate $\delta^{18}O$ and $\Delta^{17}O$ values of field samples downwind from $NO_x$-emitting sources at mid-latitudes (Kendall et al., 2007; Proemse et al., 2013). In those studies, $\delta^{18}O$ and $\Delta^{17}O$ values were suggested to be useful

to apportion the contribution of emission sources to regional atmospheric nitrate loads. However, the signals of precursor $NO_x$ emitted from the same sources may quickly get modified through isotopic equilibration with $O_3$, so that the original source signals may be difficult to recognize.



In the past, due to sampling challenges, $HNO_3$ and $p\text{-}NO_3^-$ were generally collected together (without differentiation). Therefore, no studies have separately and simultaneously collected and analyzed the $HNO_3$ and $p\text{-}NO_3^-$ $\delta^{18}O$ and $\Delta^{17}O$ values, and discussed these isotopic characteristics of nitrate collected downwind of anthropogenic emitters at mid-latitudes. It is clear
that investigating the causes of isotopic fractionation of these nitrate species and identifying the reaction pathways responsible for their transformation would contribute to our understanding of nitrogen worldwide.

While national reported $NO_x$ emissions in Canada declined steadily from 2000 to 2015, emissions in the Province of Alberta have remained relatively constant since 2004 (Environment and Climate Change Canada, 2016). Pioneering work was
accomplished measuring nitrate on emitted PM2.5 (particulate matter less than 2.5 μm) and in bulk and throughfall precipitation samples (wet and some dry deposition on ion exchange resin collectors) collected at or downwind of the Athabasca oil-sands mining operations in Alberta (Proemse et al., 2012; Proemse et al., 2013). However, the Edmonton area in Central Alberta, known to generate the highest $NO_x$ emissions in Canada, and the area of southern Alberta, characterized by dense gas compressor station and agricultural emissions, have never been investigated.

Here we have characterized nitrate downwind of five targeted emission sources in central and southern Alberta, namely: (1) coal-fired power plants, (2) city traffic, (3) chemical industries and metal refining, (4) fertilizer plant and oil refinery, and (5) gas compressors plus cattle and swine feedlots. To this end, we employed wind-sector-based active samplers to collect $HNO_3$ and $p\text{-}NO_3^-$ as well as $w\text{-}NO_3^-$ downwind of the source types. The objective of this work was to assess the atmospheric $NO_x$
reaction pathways and processes responsible for the distribution of $HNO_3$, and $w\text{-}$ and $p\text{-}NO_3^-$ in mid-latitudinal regions.

## 2 Methodology

### 2.1 Regional context

This research project investigated nitrates ($p\text{-}NO_3^-$, $HNO_3$ and $w\text{-}NO_3^-$) from three main emission source areas: the Genesee and Edmonton areas of central Alberta, and the Vauxhall area of southern Alberta. These areas experience a continental climate,
but the mean annual temperature at Vauxhall is slightly higher (5.6 °C) and total annual precipitation lower (320 mm) than in central Alberta (3.9°C; 537 mm; Fig. SI-1). Autumn is generally the wettest season and winter the driest. The sampling sites were at altitudes between 645 and 820 m (altitude above sea level), and in continental regions devoid of the influence of marine air masses (negligible halogen oxides).

The rural Vauxhall area was selected for collecting nitrates emitted from multiple small gas compressor stations scattered throughout southern Alberta and reduced N from cattle and swine feedlots (see Savard et al., 2017). The other anthropogenic emissions are from three sites in central Alberta: coal-fired power plants (CFPP) at the Genesee site, 55 km southwest of Edmonton; traffic-dominated emissions at Terrace Heights, a residential area near downtown Edmonton; and an industrial area in Fort Saskatchewan, northeast of Edmonton, where sampling two different wind sectors allowed separating different
industries. In Fort Saskatchewan, sampling in the northwest sector targeted emissions from a mixture of sources of which the



largest were a chemical plant and metal refinery (referred to as chemical plus metal industries), while the north sector point emissions were dominated by a fertilizer plant and an oil refinery (referred to as fertilizers plus oil).

## 2.2 Sampling protocols

Collection of nitrate samples took place between 30 September 2010 and 20 January 2014. Active air sampling was carried out
using a modified version of Environment Canada's standard CAPMoN (Canadian Air and Precipitation Monitoring Network) sampling protocol, which is recognized and used worldwide to measure ambient $NO_3^-$ ($HNO_3$ and p-$NO_3^-$). A description of the samplers and sample handling protocols is available in the literature (Sirois and Vet, 1999). A 'conditional sampling' method was employed to maximize the collection of nitrogen compounds from the sources of interest, in which the sampling pumps and precipitation collector were activated when the site wind vane registered winds from the direction of the targeted
sources. The CAPMoN sampling system was installed and operated at different sites, each at varying distances from the targeted point (<1 to 25 km), and diffuse sources (3 to >125 km; Table 1). Back trajectories run using the HYSPLIT model (Stein et al., 2015; Rolph, 2017) for every hour of sampling verified that very few air masses within the sampling sector passed over other sources outside of the targeted ones in the preceding 24 hours. Ambient air was pulled through a three-stage filterpack system to collect, sequentially, particulate matter, gaseous nitric acid ($HNO_3$), and gaseous ammonia ($NH_3$; see (Savard et al., 2017),
for exact location of sampling sites and $\delta^{15}N$ values of reduced and oxidized atmospheric N species). Here we report on the simultaneously sampled $HNO_3$ and p-$NO_3^-$, along with co-sampled w-$NO_3^-$ in rain and snow samples collected in CAPMoN wet-only precipitation samplers. Note that precipitation events did not occur regularly (see Fig. SI 1), so that the number of aqueous samples collected was fewer than the gas and particulate samples. Both the air and precipitation samplers were only active when the wind direction was from the desired source sector and the wind speed was greater than 0.55 m/s. Four identical
air-sampling systems operated simultaneously at each site, with samples pooled when necessary to provide sufficient filter loadings for isotope analysis and, when possible, measured separately to estimate sampling precision. In contrast to the four gas-and-particle sampling systems, there was a single precipitation collector at each site, and therefore precision was not determined for precipitation samples. Individual sample deployment times ranged from 5 to 113 days, and total air sampling time within the wind-direction sectors ranged from 21 to 360 hours. The variable cumulative periods reflected the frequency
of the wind flow from the targeted source sectors and the amount of time required to obtain sufficient mass loadings on the filters.

Two or three replicate samples for most species were pooled at Genesee and Vauxhall, the first two sampling sites, subject to the requirement that sampled air volumes be within 15 % of each other, thereby eliminating samples that experienced flow
problems. At the sites sampled later in the Edmonton area, improvements to the laboratory analytical procedure allowed for smaller sample amounts and eliminated the need for sample pooling. Replicate isotopic values at these sites could be determined by analyzing each of the four samples individually.





### 2.3 Analytical procedures

We characterized the triple oxygen isotopic ratios ($\delta^{17}O$, $\delta^{18}O$) of $HNO_3$, $w\text{-}NO_3^-$, and $p\text{-}NO_3^-$, along with their $\delta^{15}N$ values, and those of $NH_3$, $w\text{-}NH_4$, $p\text{-}NH_4$ and $NO_x$ (all N isotopic results are in Savard et al., 2017). The present article deals solely with the $\delta^{18}O$ and $\delta^{17}O$ ($\Delta^{17}O$) values obtained for the three nitrate species. We treated the samples with the chemical conversion

and thermal decomposition of $N_2O$ protocols, providing the ability to simultaneously analyze low-concentration N- and O-containing species (Smirnoff et al., 2012).

A notable challenge in the analysis of the filter-based atmospheric samples is their small extraction volumes; only 6-10 mL of extract solution was normally available for the measurement of concentrations and isotopic analysis. In addition, concentrations

of these low volume samples were generally low (7.1-21.4 $NO_3$ μmol/L) for the protocols preconized here. Therefore, not all samples could be diluted to produce volumes sufficient for reduction of $NO_3^-$ to $NO_2$ and subsequent conversion to $N_2O$, the final product before isotope analysis. Samples with an initial concentration below (2.3 μmol/L) could not be treated, and some were combined to produce volumes sufficient for analyses (same sampling period but combination of several collected samples). All extracted $N_2O$ was analyzed with the pre-concentration/gold furnace-IRMS system developed at the Geological

Survey of Canada (Smirnoff et al., 2012). This approach allows the spectrum of $\delta^{15}N$, $\delta^{17}O$ and $\delta^{18}O$ ratios from O-bearing N-species to be determined in samples containing as little as 37.5 nmol of N (15 mL final solution). The USGS-34, USGS-35, USGS-32 nitrate reference materials were used and processed exactly the same way as the samples, *i.e.*, converted from nitrate to nitrite, then to $N_2O$. The laboratory analytical precision (average of replicates) determined during the present study using the described analytical procedures was 0.6 ‰ for $\delta^{18}O$ and $\delta^{17}O$ values in gaseous (n=12) and solid nitrates (n=20). For $w\text{-}NO_3$,

analytical duplicates gave 0.6 and 0.5 ‰, for $\delta^{18}O$ (n=3) and $\delta^{17}O$ (n=4) values, respectively. The $\Delta^{17}O$ values are defined as ln $(1+\delta^{17}O/1000)$ − 0.516 x ln $(1+\delta^{18}O/1000)$, relative to Vienna Standard Mean Ocean (VSMOW).

## 3 Results and interpretation

### 3.1 Isotopic reproducibility when using the CAPMoN filterpack sampling system

Data obtained using at least two of the four identical CAPMoN sample collection streams at each sampling site were used to calculate the reproducibility of each isotopic value measured. With four or fewer samples collected during each sampling period, a non-parametric approach was deemed most appropriate. Therefore, for each period in which multiple (2-4) values were obtained, a median isotopic value was calculated, then the two to four absolute deviations from this median were

calculated (Table 2). This operation was repeated for each of the 18 sampling periods in which multiple measurements were obtained. Although there were four replicates in 18 periods, the pooling of simultaneously collected samples and the QC steps described earlier reduced the total number of replicates for each compound (Table 3). The median absolute deviation (MAD) for each compound was then calculated from the 15-38 absolute deviations. Finally, the MAD was scaled by dividing by 0.6745 to give the modified median absolute deviation (M.MAD). This scaling factor ensures that the M.MAD will be consistent with

the standard deviation in the event that the distribution is Gaussian (Randles and Wolfe, 1979; Sirois and Vet, 1999). This suite



of parallel tests indicates that all measured species show coherent and reproducible $\delta^{18}O$ and $\delta^{17}O$ results, with the M.MAD varying between 0.7 and 2 ‰ (Table 2). These estimations encompass the precision of the entire method, including errors due to sampling, chemical treatments and instrumental analysis.

A potential complication of the air sampling method can arise if there was significant volatilization of $NH_4NO_3$ on the particle filter into $HNO_3$ and $NH_3$, with subsequent collection on the downstream gas filters. This could result in isotopic fractionation between the particle and gaseous components, which would become artificially high and low, respectively, with stronger effects at higher temperatures (summer) relative to lower temperatures (winter). We find the overall p-$NO_3^-$ isotopic ratios to be higher during winter than during summer (see Section 3.4). Moreover, the p-$NO_3^-$ minus $HNO_3$ isotopic difference is negative during
summer, opposite to the expected isotopic artefact if particulate volatilization were influencing the final signal of the samples. We therefore concluded that collecting particles and gaseous components simultaneously provides the best information for understanding atmospheric reactive nitrogen. Finding that the sampling protocols are adequate for isotopic work is in agreement with a previous study using a comparable method that found minimal fractionation between p-$NO_3^-$ and $HNO_3$ (Elliott et al., 2009).

### 3.2 Averages and ranges of triple oxygen isotopic results in Alberta

The average $\delta^{18}O$ and $\Delta^{17}O$ values of $HNO_3$ (gas), w- and p-$NO_3^-$ show no apparent systematic ordering (Table 3), in contrast to what was found for $\delta^{15}N$ values in the same samples (Savard et al., 2017). In addition, there is no systematic tendency when looking at the samples collected downwind from the five-targeted sources: CFPP $HNO_3$ and p-$NO_3^-$ have the highest $\delta^{18}O$ and
$\Delta^{17}O$ averages, but not the highest w-$NO_3^-$ values; chemical industries show the lowest $\delta^{18}O$ and $\Delta^{17}O$ averages for w- and p-$NO_3^-$, but not for $HNO_3$. At first sight, this observation suggests that the oxygen isotopes in the three nitrate species are not predominantly source-dependent (see Section 3.3).

Considering all nitrate species, the Alberta $\delta^{18}O$ and $\Delta^{17}O$ values range between +48.4 and +83.2 ‰, and between 13.8 and
30.5 ‰, respectively (Table 4). These ranges indicate that ozone partly transferred its isotopic anomaly to nitrates during NOx cycling and oxidation (nitrate derived through $O_2$ oxidation would show $\delta^{18}O$ and $\Delta^{17}O$ values of 23.5 and 0 ‰, respectively). When examining the existing $\delta^{18}O$ and $\Delta^{17}O$ data for w- and p-$NO_3^-$ in the literature, the ranges for our mid-latitude samples are within those previously reported (Table 4). The worldwide compilation of documented data is broadening the $\delta^{18}O$ range of atmospheric $NO_3^-$ previously suggested to be between 60 and 95 ‰ (Hastings et al., 2003; Kendall et al., 2007).


Previous studies that report triple isotope oxygen results in atmospheric $NO_3$ samples are scarce (Table 4). To our knowledge, triple oxygen isotopic characterization specific to $HNO_3$ has never been documented before; previous studies only reported values for the sum of $HNO_3$ and p-$NO_3^-$ (Table 4). The $HNO_3$ range is within the broad spectrum of p-$NO_3$ values compiled for remote to contaminated sites. Elliott et al. (2009) have reported $HNO_3$ oxygen results for $\delta^{18}O$ values only, with a range of
+51.6 to +94.0 ‰ (mean of 77.4), with simultaneously-sampled p-$NO_3^-$ $\delta^{18}O$ values between +45.2 and +92.7 ‰ (mean of 75.2). Those ranges are broader than the $HNO_3$ and p-$NO_3^-$ values obtained in the present study. The $\delta^{18}O$ and $\Delta^{17}O$ ranges we




document here for HNO$_3$ in Alberta are narrower than those of the simultaneously collected p-NO$_3^-$ (Fig. 1), suggesting that there are additional isotopic fractionation processes when HNO$_3$ is transformed to p-NO$_3^-$, or that p-NO$_3$ is derived from different pathways with more variation in isotopic signatures.

### 3.3 Covariations of $\Delta^{17}$O and $\delta^{18}$O values in nitrates from individual sources

The p-NO$_3^-$, w-NO$_3^-$ and HNO$_3$ values co-vary when identified by source type in the $\delta^{18}$O and $\Delta^{17}$O space (Fig. 1). The isotopic range for a single source can be as large as 6 ‰ for $\Delta^{17}$O values and 19 ‰ for $\delta^{18}$O values in HNO$_3$, 12 and 17 ‰ in w-NO$_3^-$, and 7 and 21 ‰ in p-NO$_3$. Each source type clearly exhibits nitrate $\Delta^{17}$O and $\delta^{18}$O with a specific grouping. The CFPP w-NO$_3^-$ results show a range similar to the HNO$_3$ results, but lower $\delta^{18}$O values than the HNO$_3$ and p-NO$_3^-$ groups. The few other precipitation samples show $\delta^{18}$O and $\Delta^{17}$O values generally higher than the p-NO$_3^-$ and HNO$_3$ samples, again with exception

of the chemical and metal industries.

    The HNO$_3$ samples from a given source type tend to have a higher $\delta^{18}$O value for a given $\Delta^{17}$O value than p-NO$_3^-$ (or *vice versa*; Fig. 1). These observations suggest that the contribution of oxidation pathways leading to HNO$_3$ and p-NO$_3^-$ are not identical, or that there is an isotope fractionation in the conversion of HNO$_3$ to p-NO$_3^-$. Therefore, separate measurements of

the gaseous and particle nitrate forms may provide additional constraints on oxidant concentrations.

    Regarding the potential for identifying nitrate sources, it appears that using $\delta^{18}$O and $\Delta^{17}$O values for such a task is not feasible, as previously suggested in the literature (Michalski et al., 2003). This interpretation stems from the fact that nitrate species show either continuous trends regardless of their sources (p- and w- NO$_3$) or overlapping source results (HNO$_3$; Fig. 1).

The individual range of points identified by source may partly reflect different initial ambient conditions and rates of changes in ambient conditions during NO$_x$ oxidation (Fig. 1; see Section 3.5). Specifically, each isotopic range may depict the progressively changing influence of ozone due to ambient conditions through time. Indeed, the atmospheric samples were collected repeatedly over several weeks or months at a given site (near a given source), and consecutively from one site to the

other over more than three years; samples undeniably incorporate N-species produced under significantly changing ambient conditions.

### 3.4 Seasonal $\delta^{18}$O and $\Delta^{17}$O trends in nitrates

    The $\Delta^{17}$O and $\delta^{18}$O results from all sources combined and identified by sampling period for HNO$_3$, w-NO$_3^-$ and p-NO$_3^-$ clearly

show higher $\delta^{18}$O and $\Delta^{17}$O values during cold periods relative to warm periods (Fig. 2), with the exception of HNO$_3$ $\delta^{18}$O values, which were similar in cold and warm periods. As mentioned above, the collection of several samples lasted over periods overlapping fall and winter periods and, in such cases, the results are labelled as covering the two seasons; note that for many fall cases, the average sampling temperatures were below 0ºC. Nevertheless, plotting by sampling period can be regarded as a general repartition of results between warm and cold months, which show lower and higher isotopic values, respectively, in

both the w- and p-NO$_3^-$.




A series of reactions listed in Table 5 summarizes the main atmospheric processes taking place during the production of nitrates in contaminated air masses. First, during anthropogenic combustion of fossil fuels $NO_x$ (NO and $NO_2$) is produced through reactions of air $N_2$ with atmospheric $O_2$ at high temperatures (reactions R1; Table 5). Then, $NO_x$ cycles between NO and $NO_2$

through a series of reactions involving sunlight (R5), $O_3$ (R2, R4), and peroxy ($HO_2$) or alkyl peroxy ($RO_2$) radicals (R3; Morin et al., 2007; Fang et al., 2011; Michalski et al., 2014; here we use $RO_2$ to refer collectively to $HO_2$ and $RO_2$).

The oxidation of $NO_x$ (specifically $NO_2$) to $HNO_3$ further incorporates additional O atoms from different oxidants (R6-R8; Table 5). Production of nitrate via R6 is restricted to daytime (since OH is generated through photochemistry), whereas

production through reactions R4, R7 and R8 dominates at night. In addition, $N_2O_5$ is thermally unstable, so the p-$NO_3^-$ contribution of the R4-R7-R8 pathway is larger during winter than during summer. We have neglected contributions from BrO cycling due to the location far from the coast, and from reactions of $NO_3^-$ with hydrocarbons (R12) since they are predicted to have a minimal contribution to nitrate formation in this region (Alexander et al., 2009). Finally, $HNO_3$ in the gas phase can be irreversibly scavenged by wet surfaces or precipitation (R9) and calcium carbonate on particles (R11), and can equilibrate with

solid ammonium nitrate where there is excess ammonia available (R10).

It has been previously suggested that the $\delta^{18}O$ and $\Delta^{17}O$ values of w- and p-$NO_3^-$ formed during summer are lower than those during winter due to higher contribution from the $N_2O_5$ path (R4, R7-R8) during that season (e.g., Hastings et al., 2003; Morin et al., 2008). As an early take on the data identified by sampling periods, it seems that all the studied w- and p-$NO_3^-$ show $\delta^{18}O$

and $\Delta^{17}O$ trends following the expected patterns for warm and cold months (Fig. 2). In the case of the less commonly studied $HNO_3$, the summer $\Delta^{17}O$ values are lower than the fall-winter, fall and spring ones, suggesting that the processes leading to the summer isotopic ratios perhaps include $O_3$ contributions similar to winter ones, but with lower $\Delta^{17}O$ values (see section 4.1).

### 3.5 Isotopic differences between $HNO_3$ and p-$NO_3^-$

Regarding the specific forms of nitrate, it is pertinent to mention that the $HNO_3$ concentrations (from 0.01 to 0.15 µg N/m$^3$; average of 0.06) are slightly lower than those of p-$NO_3^-$ (from 0.02 to 0.35 µg N/m$^3$; average of 0.12). For context, the median concentrations at all CAPMoN sites, which represent non-urban areas across Canada, range from 0.07 to 1.1 µg N/m$^3$ for $HNO_3$ and from 0.03 to 2 µg N/m$^3$ for p-$NO_3^-$ (Cheng and Zhang, 2017). In the Alberta samples, $HNO_3$ is present at such significant proportions, that if they had not been differentiated, the low end of the isotopic range obtained for p-$NO_3^-$ would have been

significantly higher in both $\delta^{18}O$ and $\Delta^{17}O$ (Table 4, see undifferentiated category). Hence, as far as the isotopic characteristics are concerned, an important feature to keep in mind is that the $HNO_3$ of central and southern Alberta has distinct properties relative to simultaneously-sampled p-$NO_3^-$ and co-sampled w-$NO_3^-$. In practical terms, the relationships between the simultaneously sampled $HNO_3$ and p-$NO_3^-$ are of four types (Fig. 3): (i) $HNO_3$ $\delta^{18}O$ and $\Delta^{17}O$ are both lower than p-$NO_3^-$; (ii) $HNO_3$ has lower $\Delta^{17}O$ but higher $\delta^{18}O$ values than p-$NO_3^-$; (iii) $HNO_3$ has higher $\delta^{18}O$ values and similar $\Delta^{17}O$ ones relative to

p-$NO_3^-$; and (iv) $HNO_3$ has higher $\delta^{18}O$ and $\Delta^{17}O$ values than p-$NO_3^-$ (Fig. 3).



The fall-winter isotopic results belong to group (i), fall results, to groups (i), (ii) and (iii), and the spring and summer results, to groups (ii), (iii) and (iv). Elliott et al. (2009) reported simultaneously sampled p-NO$_3^-$ and HNO$_3$ in northeastern USA with similar seasonal changes of $\delta^{18}$O differences (no $\Delta^{17}$O measurement). The HNO$_3$ $\delta^{18}$O were generally similar or lower than the p-NO$_3^-$ values during winter and fall, and slightly to much higher during spring and summer, with the spring and autumn p-

NO$_3^-$-HNO$_3$ relationships being roughly intermediate between the winter and summer ones. The average $\delta^{18}$O difference of p-NO$_3^-$ minus HNO$_3$ reported between winter and summer (15 ‰) by Elliott et al. (2009) agrees with the difference for fall-winter and summer obtained here (12 ‰).

The marked shifts in isotopic differences between the HNO$_3$ and p-NO$_3^-$ reported here likely reflect changes in the dominant

processes leading to the production of the different sampled nitrate forms (see Section 4.1). These isotopic differences also imply that the analysis of samples that combine p-NO$_3^-$ and HNO$_3$ can mislead when attempting to understand in detail the chemical pathways involved in a specific region where HNO$_3$ is significantly present; and that the isotopic signals of HNO$_3$ and p-NO$_3^-$ are not interchangeable in such cases.

## 4 Discussion

### 4.1 Dominant oxidation pathways producing HNO$_3$, w-NO$_3^-$ and p-NO$_3^-$

The Alberta nitrate values do not fall on a single line but, rather, show a vertical extent in the $\delta^{18}$O and $\Delta^{17}$O space (Fig. 2) that exceeds the precision of the data (Section 2.3 and Table 2). This observation differs from several studies that measured bulk

nitrate or a single nitrate species and reported $\delta^{18}$O and $\Delta^{17}$O sets as linear.

Considering the relevant oxidation reactions shown in Table 5, anthropogenic atmospheric nitrates incorporate O atoms from three main molecules, O$_2$ (via RO$_2$ - R3, and possibly source NO$_x$- R1), O$_3$ (via NO$_2$, NO$_3^-$ and N$_2$O$_5$ – R2, R4, R7-R8) and H$_2$O (via OH, R5-R6). These molecules carry distinct isotopic signals that will partly determine the final $\delta^{18}$O and $\Delta^{17}$O values

of the nitrate products. The $\delta^{18}$O and $\Delta^{17}$O values of O$_2$ are 23.5 and 0 ‰, respectively. Anthropogenic emitters involving combustion (O$_2$) may generate primary NO$_x$ at or near sources that tend to carry low $\delta^{18}$O and $\Delta^{17}$O values. This primary NO$_x$ (>90 % emitted as NO) cycles through NO-NO$_2$-O$_3$-NO numerous times before it is removed in R6. OH typically has negative $\delta^{18}$O values and a $\Delta^{17}$O value equal to 0 ‰ as it rapidly exchanges O isotopes with water vapour (Dubey et al., 1997; Röckmann et al., 1998). We obtained the average of precipitation $\delta^{18}$O values for each sampling period at the studied sites (OPIC, 2017),

and calculated the vapour signal using water-vapour fractionation factors (Clark and Fritz, 1997). Next, using fractionation factors between OH in equilibrium with H$_2$O vapour (Walters and Michalski, 2016), we calculated that the $\delta^{18}$O values would range between -83 and -62 ‰. Peroxy radicals mostly derive from O$_2$ at mid latitudes (Michalski et al., 2003; Morin et al., 2007; Alexander et al., 2009), but they have a non-zero $\Delta^{17}$O signal (1-2‰) due to the role of ozone in the HO$_x$ cycle (Morin et al., 2011). However, their $\delta^{18}$O values are difficult to measure, so they can only be inferred based on assumptions (+23.9 ‰;

Fang et al., 2011; Guha et al., 2017). The $\delta^{18}$O and $\Delta^{17}$O values of bulk O$_3$ are generally between 90 and 120 ‰, and 30 and 34 ‰, respectively, but the transferable signals are suggested to be around 130 and 39 ‰ at mid-latitudes (Vicars and Savarino,



2014). Moreover, $NO_x$ modelled at isotopic steady state with tropospheric $O_3$ yields 117 and 45 ‰ in $\delta^{18}O$ and $\Delta^{17}O$, respectively (Michalski et al., 2014). This neglects the contribution of NO oxidation by $RO_2$ (R3), which will reduce the steady-state $\Delta^{17}O$ and $\delta^{18}O$ of $NO_x$ below the $O_3$-only oxidation value. The foregoing review of isotopic signals provides context to the interpretation of our data, keeping in mind that mass-dependent fractionation has likely played a role in determining nitrate 5 $\delta^{18}O$ values.

Generally, the winter to summer isotopic differences are thought to be due to the high oxygen isotopic values of $N_2O_5$ due to interaction with $O_3$ (Johnston and Thiemens, 1997; Michalski et al., 2003; Morin et al., 2008; Vicars et al., 2012) and low values of OH in isotopic equilibrium with atmospheric $H_2O$ (Dubey et al., 1997). According to Table 5, these two reaction 10 pathways produce nitrates via R4-R7-R8 with 2/3 O from $NO_2$, 1/6 O from $O_3$ and 1/6 O from $H_2O$, and nitrates via R6 with 2 out of 3 O atoms coming from $NO_2$ and 1 added O from OH (e.g., Michalski et al., 2003). Using these proportions with the Alberta $\Delta^{17}O$ values of p-$NO_3^-$ and $HNO_3$ in weighted averages allows us to make a rough estimation of the maximum and minimum $\Delta^{17}O$ values of $NO_2$ oxidized to nitrates in the air masses sampled. The calculations assume the O from $O_3$ contributes a signal of ~39 ‰ as was recently measured (Vicars et al., 2014) and that $\Delta^{17}O$ of OH and $H_2O$ are zero. The estimated $NO_2$ 15 $\Delta^{17}O$ values for fall-winter (34-45 ‰ daytime, 25-36 ‰ nighttime) and summer sets (25-34 ‰ for daytime; 15-24 ‰ for nighttime) represent the extremes assuming daytime oxidation takes place 100 % through the OH pathway and nighttime oxidation takes place entirely through the $N_2O_5$ pathway. One should keep in mind that the Alberta results are for nitrates collected during multi-week sampling periods. Each nitrate sample therefore contains *a priori* a mixture of O from the pathways operating during daytime (R6) and nighttime (R4-R7-R8). Assuming a 50 % contribution from each pathway for summer, we 20 generate values ranging from 20 to 29 ‰. Alternatively, assuming the domination of the $N_2O_5$ pathway during winter (90 %; Michalski et al., 2014), the range shrinks to 26-37 ‰. Fall and spring values should fit between these summer and winter estimated ranges.

The estimated $NO_2$ $\Delta^{17}O$ ranges indicate that the potential parent $NO_2$ generally had a smaller [17]O anomaly than $O_3$ (39 ‰; 25 Vicars and Savarino, 2014) or $NO_2$ in isotopic equilibrium with $O_3$ alone (45 ‰; Michalski et al., 2014). Two mechanisms could be responsible for the difference with the latter. One is the competition of R3 with R2 in oxidizing NO to $NO_2$, since $RO_2$ will dilute the $\Delta^{17}O$ relative to an ozone-only equilibrium. The relative reaction rates of R2 and R3 have previously been assumed to control the $NO_2$ isotopic composition (e.g., Alexander et al., 2009) based on the assumption of isotopic steady state. A larger contribution of $RO_2$ is expected in the $NO_2$ precursors for summer relative to winter, since biogenic VOCs that are 30 major sources of $RO_2$ radicals are much higher in the summer. This suggestion is consistent with the lower $\Delta^{17}O$ ranges in summer reported here. A second possibility is that the $NO_2$ did not reach isotopic steady state with $O_3$ and $RO_2$, retaining some of its original signature (assumed to be $\Delta^{17}O=0$). Most studies have assumed that since photochemical steady state is established within a few minutes after emission of $NO_x$ from a combustion source, isotopic steady state is also reached quickly. However, recent modeling by Michalski et al. (2014) suggests that isotopic equilibration of $NO_x$ with $O_3$ could take up to a few hours at 35 the relatively low $NO_x$ and $O_3$ concentrations in rural Alberta. At the measured average wind speeds on site of 8-19 km hr$^{-1}$, transit times could be 9 minutes to almost 4 hours. Therefore, the nitrates measured at our sites may partly derive from NOx





that had not yet reached isotopic steady state with $O_3$ (and $RO_2$). The extent of the $\Delta^{17}O$ difference between the summer and fall-winter $HNO_3$ clusters (Fig. 2B) likely reflects various combinations of these two mechanisms.

A challenging question is why do the $HNO_3$ to p-$NO_3^-$ isotopic differences shift seasonally (Figs. 3, 4)? One factor that may
influence the relationship between $HNO_3$ and p-$NO_3^-$ is mass-dependent isotopic equilibrium between $NH_4NO_3$ and $HNO_3$ (R4, R7-R8 and R9-R10); however, this mechanism would result in higher $\delta^{18}O$ in p-$NO_3^-$ and unchanged $\Delta^{17}O$ values and, therefore, cannot be solely responsible for any of the observed patterns (Fig. 3). Alternately, the trend for cold months (trend $i$) could be due to the fact that the heterogeneous $N_2O_5$ pathway is likely to contribute more to p-$NO_3^-$ than to $HNO_3(g)$, which would result in a higher contribution from ozone and explain why $\delta^{18}O$ and $\Delta^{17}O$ values are both higher in p-$NO_3^-$. A previous study
addressing why p-$NO_3^-$ on coarse particles is more enriched than on fine particles invoked a similar explanation (Patris et al., 2007).

For some of the spring and summer samples, both $\delta^{18}O$ and $\Delta^{17}O$ were lower in p-$NO_3^-$ than in $HNO_3$ (trend $iv$), therefore the mechanism above cannot dominate the fractionation; nor can a mass-dependent process be responsible. We suggest a different
fractionation process because $HNO_3$ dry deposits to surfaces more rapidly than p-$NO_3^-$ (Zhang et al., 2009; Benedict et al., 2013), which would only apply in the situation where $NO_2$ is not in isotopic equilibrium with $O_3$ in a fresh plume. The first nitrates formed in the plume shortly after emission from the $NO_x$ source have low $\delta^{18}O$ and $\Delta^{17}O$ values since $NO_x$ has not yet reached isotopic steady state with $O_3$ and $RO_2$. Those nitrates that form as p-$NO_3^-$ or that partition to p-$NO_3^-$ remain in the plume for longer than $HNO_3$, which deposits more rapidly upon contact with vegetation or other surfaces. As the plume travels,
the $NO_x$ becomes more enriched, and the newly formed nitrates take on higher $\delta^{18}O$ and $\Delta^{17}O$ values. However, p-$NO_3^-$ collected downwind will be a mixture of the low-$\delta^{18}O$ and -$\Delta^{17}O$ nitrate formed earlier and the high-$\delta^{18}O$ and -$\Delta^{17}O$ nitrate formed more recently, while $HNO_3$ will have less of the low- $\delta^{18}O$ and -$\Delta^{17}O$ nitrate. The fact that we find the lowest values in summer p-$NO_3^-$ samples collected from various industries at distance less than 16 km supports this suggestion (Table 1). Seasonal changes in the planetary boundary layer height may also affect the impingement of emission plumes on the
measurement sites, and thereby the relative amounts of fresh vs background nitrates.

The two mechanisms that we have proposed – differential $N_2O_5$ contribution resulting in higher $\Delta^{17}O$ values in p-$NO_3^-$ than in $HNO_3$, and differential deposition resulting in lower $\Delta^{17}O$ values in p-$NO_3$ – would essentially compete against each other, with local conditions and chemistry influencing the results. In winter, when the $N_2O_5$ pathway is most important, the first
mechanism dominates, as supported by the observation that p-$NO_3^-$ concentrations are higher during that season. Conversely, in summer, when the $N_2O_5$ pathway is less important and dry deposition is likely faster due to minimal snow cover, higher surface wetness and high leaf areas, the second mechanism is more important. The local reactant concentrations, wind speeds and radiative fluxes (which control the time to reach isotopic equilibrium) would also be factors in the second mechanism. Therefore, we would expect intermediate trends ($ii$, $iii$) in other seasons. In addition to these non-mass-dependent fractionation
processes, mass-dependent fractionation in formation and loss of nitrate likely contributes to the observed $\delta^{18}O$ differences. For instance, kinetic fractionation may be involved in the production of trend $iii$.



In summary, the $\delta^{18}O$ and $\Delta^{17}O$ patterns of the measured nitrates follow the generally described seasonal isotopic trend from high, during cold periods, to low, during warm periods. However, examining the isotopic relationship of $HNO_3$ to p-$NO_3^-$ (Fig. 3), and by extension w-$NO_3^-$, reveals the complexity of anthropogenic $NO_x$ oxidation mechanisms. The final isotopic values are derived from the $O_3$-$NO_x$ (Leighton) cycle - –possibly not yet at isotopic steady state - followed by the OH and $N_2O_5$ ($O_3$)

oxidation pathways in proportions that vary with the periods of sampling (Fig. 4). The negative isotopic difference between p-$NO_3^-$ and $HNO_3$ during warm months likely reflects differential removal rates from plumes containing $NO_x$ at disequilibrium with $O_3$. The isotopic signals from $NO_x$ to the final nitrate species follow the isotopic trajectory of the dominant reactions, so we conceive the final oxidation trends as lines resulting from adding vectors that represent these main trajectories (Fig. 4).

**4.2 Correlations with other meteorological parameters and co-pollutants**

The distribution and proportion of $HNO_3$ and p-$NO_3^-$ in polluted air masses can vary daily and seasonally with temperature, relative humidity (RH) and concentration of co-contaminants (Morino et al., 2006). For that reason, we compared the isotopic ratios of the $HNO_3$ and p-$NO_3$ samples with meteorological and air quality parameters measured routinely at nearby monitoring stations where available (within 5 km). We found that the p-$NO_3^-$ and $HNO_3$ $\delta^{18}O$ and $\Delta^{17}O$ values correlate with RH, with p-

$NO_3$ values showing stronger statistical links than $HNO_3$ (Table 6). The $N_2O_5$ hydrolysis reaction (R8) rate increases with humidity (Kane et al., 2001), which may explain this positive correlation. Significant inverse relationships exist between temperature and p-$NO_3^-$ $\delta^{18}O$, p-$NO_3^-$ $\Delta^{17}O$, and $HNO_3$ $\Delta^{17}O$. These negative links likely arise since $N_2O_5$ is more stable under cold conditions, leading to a higher contribution of R8. The stronger links with p-$NO_3^-$ may be due to R8 taking place on surfaces (such as particles) with liquid water, which is likely to retain the $HNO_3$ as p-$NO_3^-$ rather than release it to the gas

phase. Therefore, in winter R8 may contribute more to p-$NO_3$ than to $HNO_3$(g), supporting our interpretation of trend *i* (Fig. 4). Moreover, the highest $\delta^{18}O$ and $\Delta^{17}O$ values were found for fall-winter samples with proportions of 33 to 66 % for $HNO_3$ and p-$NO_3^-$, respectively, and collected at high RH (76 %) and low temperature (-10ºC). In contrast, the lowest p-$NO_3^-$ isotopic values were found for samples with similar proportions of $HNO_3$ and p-$NO_3^-$, and sampled during moderately humid (60-63 %) and warm (8-20ºC) periods. In other words, the $HNO_3$ and p-$NO_3^-$ results obtained here partly support the previous

suggestion that temperature and RH amounts can shift nitrate partitioning between $HNO_3$ and p-$NO_3^-$ (Morino et al., 2006).

Concentrations of oxidants, co-contaminants (e.g., $SO_4^-$ aerosols) and $NO_x$ influence the dominance and rates of the discussed reactions (Brown et al., 2006; Michalski et al., 2014). For instance, high $NO_x/O_3$ ratios do not favor NO oxidation by peroxy radicals ($RO_2$ family) with typically low isotopic values, explaining the positive link between $\delta^{18}O$ and $\Delta^{17}O$ values with

$NO_x/O_3$ ratios (Table 6). It appears however that the PM and $SO_2$ concentrations were not correlated with the production of the measured nitrates. Surprisingly, only the p-$NO_3^-$ $\Delta^{17}O$ and $\delta^{18}O$ values correlated with the fraction of each sample collected during daylight hours (i.e., between the sunrise and sunset times on the day at the middle of each sampling period, either at Edmonton or Lethbridge), which was expected for $HNO_3$ as well due to the daytime-only OH pathway. However, daylight hours do not take into account light intensity, which can influence significantly the oxidation rate through this pathway, and

consequently both the $\delta^{18}O$ and $\Delta^{17}O$ values.



### 4.3 - $\delta^{18}O$ and $\Delta^{17}O$ in w- and p-NO$_3^-$ from polluted areas

Atmospheric nitrates measured in central and southern Alberta were sampled downwind of well-identified anthropogenic sources to verify the potential role of emitted NO$_x$ isotopic signals through to final nitrate isotopic ratios (primarily N isotopes; see Savard et al., 2017). As expected, the measured oxygen isotopes of the various nitrate groups show no source-specific

isotopic characteristics, but instead are consistent with exchange with O$_3$ and oxidation through the well-known OH and N$_2$O$_5$ oxidation paths. However, NO$_2$ not in isotopic equilibrium with O$_3$, and/or NO reacted with RO$_2$ significantly influenced the overall results. Co-contaminants in the emissions and sampled plumes at short distances from the sources may have favoured these two mechanisms, and quantifying RO$_2$ and/or HO$_2$ would help distinguish between the two mechanisms. The question remains: are these overall effects observable in triple oxygen isotopes of nitrates from other polluted sites?

The full $\Delta^{17}O$ and $\delta^{18}O$ ranges for p-NO$_3^-$, w-NO$_3^-$ and HNO$_3$ (between 13.8 and 20.5 ‰, 48.4 and 83.2 ‰; Table 4) compare well with the isotopic ranges obtained for bulk deposition NO$_3^-$ samples collected downwind from oil sands mining operations in the lower Athabasca region farther north in Alberta (Proemse et al., 2013). Moreover, the isotopic clusters in cold and warm months delineated here essentially overlap with the data sets of winter and summer from the lower Athabasca region (Fig. 5).

This correspondence exists despite the slightly different climatic conditions (SI-Fig. 1), different source types, and very different sampling methods (bulk/throughfall deposition samples using open ion exchange resin collectors, vs. wind sector-specific active sampling on filters and precipitation-only collectors). Notably, many points carry relatively low $\delta^{18}O$ and $\Delta^{17}O$ values.

Previous work in the Athabasca region reported very low $\delta^{18}O$ and near-zero $\Delta^{17}O$ values for p-NO$_3^-$ sampled directly within oil-sands industrial stacks, i.e., in the emissions measured in-stack and diluted with ambient air (Proemse et al., 2012). These values appear near the O$_2$-end of the O$_2$-O$_3$-NO$_x$ line (Fig. 4). Similar isotopic signatures are very likely produced in source emissions of NO$_x$ in the studied Edmonton and Vauxhall areas (e.g., CFPP, gas compressors, industries). This source signature may persist into p-NO$_3$ collected close to the sources. Within the first few hours in the atmosphere (or less, in more polluted

areas), the NO$_x$ $\delta^{18}O$ and $\Delta^{17}O$ values rapidly increase due to isotope exchange with O$_3$ via many iterations of the photochemical NO$_x$-O$_2$-O$_3$ cycle (R2, R3, R5 and O3 formation, Table 5; Michalski et al., 2014) and reach isotopic equilibrium. Though the e-folding lifetime for NO$_x$ oxidation to nitrates may be longer than these few hours, depending on the NO$_x$/VOC ratio, only a fraction of the oxidized source NO$_x$ will create a measureable contribution to the ambient nitrate where the background air is very low in nitrate. This is likely the case in the oil sands region, where Proemse et al. (2013) reported the lowest $\Delta^{17}O$ values

within 12 km of the emission sites, and where direct stack emissions of p-NO$_3^-$ were ~5000 times lower than NO$_x$ emissions (Wang et al., 2012).

In a methodological test study, we obtained low values for w-NO$_3^-$ sampled near a high traffic volume highway in Ontario, Canada (Smirnoff et al., 2012). Low $\delta^{18}O$ and $\Delta^{17}O$ values in atmospheric nitrates during warm months (65 and 20 ‰ or less,

respectively) have been reported for other parts of the world as well (Table 4). Authors of these studies have invoked peroxy radicals to account for low $\delta^{18}O$ values in w-NO$_3^-$ from a polluted city (Fang et al., 2011), in p-NO$_3^-$ from Taiwan collected partly from air masses influenced by pollutants (Guha et al., 2017) and from a polluted coastal site in California (Michalski et





al., 2004; Patris et al., 2007; Table 4). However, sampling in these three other regions did not use collection restricted to air masses transported from targeted anthropogenic sources. So uncertainties persist regarding the ultimate sources of nitrates with low isotopic values.

Although a few low values are also reported for seemingly non-polluted areas of the Arctic and Antarctic regions (unknown cause; Morin et al., 2008; Morin et al., 2009), and of coastal California (Patris et al., 2007), the information from the literature, integrated with the interpretation proposed for the Alberta low $\delta^{18}O$ and $\Delta^{17}O$ values in summer nitrates, may reflect the involvement of air masses that favour $RO_2$ reactions and/or include nitrates from oxidation of $NO_x$ with light isotopes in plumes. In such cases, we hypothesize that the $NO_x$-$O_3$ isotopic equilibrium was not reached (see Section 4.1). Keeping in mind that

other hydrocarbon and halogen pathways may play a role in determining the isotopic nitrate characteristics in other parts of the world, we propose that, in general, the warm-periods isotopic ranges appear to be lower in polluted areas, regardless of the anthropogenic source types. Given these points, our nitrate $\delta^{18}O$ and $\Delta^{17}O$ values do not represent indicators of specific sources but, more likely, in the case of summer solid phases, they may reflect anthropogenic nitrates in general. Further research work on plume $NO_x$ to nitrates chemical mechanisms may help to validate this suggestion.

### 4.4 Comparison with high-latitude p-NO$_3^-$

An interesting aspect of the Alberta p-NO$_3^-$ cold-period $\Delta^{17}O$ ranges is that they compare relatively well with the range obtained for Canadian Arctic winter (Fig. 6), during which nighttime conditions and the $N_2O_5$ pathway prevail without interruption (Morin et al., 2008). This observation supports the suggestion that the $N_2O_5$ pathway produces around 90 % of nitrates during

mid-latitudinal cold months (Michalski et al., 2003; Section 4.1). The $\delta^{18}O$ ranges of cold months are similar in Alberta and in the Arctic. This similarity goes against previous suggestions that at higher latitudes, nitrate $\delta^{18}O$ annual means should be higher than at mid-latitudes due to local ambient conditions and atmospheric chemistry affecting the proportions of species involved in producing nitrate (Morin et al., 2009), namely, the sole influence of the $N_2O_5$ pathway during the Arctic winter (Fang et al., 2011).

The $\Delta^{17}O$ departure between the Alberta and Arctic winter parallel lines is about 3‰. Such difference is slightly larger than the one calculated for winter $NO_3^-$ at 80 and 40º N latitudes (about 2 ‰; Morin et al., 2008). In contrast, the warm-months and summer data sets for Alberta and the Arctic, respectively, show different isotopic ranges (Fig. SI-3), possibly due to the plume effects described in Section 4.3. Moreover, contrary to a previous suggestion, the winter-summer contrast in nitrate $\Delta^{17}O$ values

is similar at the mid- and high-latitudinal sites (about 6 ‰ here, and 5 ‰ in Morin et al., 2008). This similarity is likely coincidental as it may reflect the fact that during warm months the anthropogenic influence may lower the summer $\Delta^{17}O$ values in Alberta (see Section 4.1), whereas the seasonal departure in Arctic samples comes from the dominant OH and $N_2O_5$ pathways during summer and winter, respectively. Finally, the $\Delta^{17}O$ averages for the Alberta summer and winter results approximately fits within ranges predicted for the studied area by global modeling (Alexander et al., 2009), suggesting that global modeling

of nitrate distribution worldwide is promising. In the future, the assumption of $NO_x$ isotopic steady state with $O_3$ should be explored, given recent findings (Michalski et al., 2014), the critical importance of $NO_x$ isotope characteristics on resulting



nitrate isotopic values (Alexander et al., 2009), and the suggestion regarding the evolution of $NO_x$-$NO_3^-$ signals in early anthropogenic plumes (present study).

**5 Conclusion**

The $HNO_3$, w-$NO_3$ and p-$NO_3$ from five different sources in central and southern Alberta, simultaneously collected with wind sector-based conditional sampling systems produced similar $\delta^{18}O$ and $\Delta^{17}O$ trends, regardless of their sources types (CFPP, various industries, city traffic, and gas compressors). This confirms previous observations that regional ambient conditions (e.g., light intensity, RH, temperature) dictate the triple isotopic characteristics and oxidation pathways of nitrates.

The gaseous form of nitrate ($HNO_3$) having distinct isotopic characteristics relative to the wet and particulate forms imply that understanding nitrate formation and loss requires characterizing the nitrate species individually, and disavows the assumption that isotopic values of the nitrate phases are invariably interchangeable. The Albertan nitrate production operated mostly through the well-known OH and $N_2O_5$ oxidation pathways, possibly prior to their reaching isotopic equilibrium with $O_3$ in

some samples, though we also suggest contributions from $RO_2$ oxidation of NO, as well leading to low oxygen isotopic values. Particulate-$NO_3^-$ generally shows higher $\delta^{18}O$ and $\Delta^{17}O$ values than $HNO_3$ in the fall-winter period as the heterogeneous $N_2O_5$ pathway favours the production of p-$NO_3^-$. Moreover, $HNO_3$ has higher $\delta^{18}O$ and $\Delta^{17}O$ values during warm periods, which we propose is due to faster dry deposition rates relative to p-$NO_3^-$ in the event that $NO_x$ is oxidized before reaching isotopic steady state with $O_3$. These mechanisms conferring p-$NO_3^-$ with relatively low isotopic values are prone to happen in anthropogenic

polluted air masses. An interesting deduction arising from this interpretation and from a comparison with nitrate isotopes from other polluted areas of the world is that relatively low $\delta^{18}O$ and $\Delta^{17}O$ values may reflect nitrates produced from undifferentiated anthropogenic $NO_x$ emissions.

Future research needs to explore the assumption of $NO_x$ isotopic equilibration with $O_3$, given the critical importance of $NO_x$

isotopes on resulting nitrate isotopic values. More field sampling and state-of-the-art isotopic analyses of all tropospheric nitrate species, as well as $NO_x$, are required for refining our understanding of atmospheric nitrate worldwide. This endeavour is fundamental for developing effective emission-reduction strategies towards improving future air quality.

*Acknowledgements*. The authors are grateful for the technical support provided by Marie-Christine Simard and Jade Bergeron of the Geological Survey of Canada, and by Syed Iqbal, Rachel Mintz, Daniel McLennan, Matthew Parsons, Mike Shaw Amy Hou of Environment and Climate Change Canada; and for the constructive pre-submission review by Drs. Geneviève Bordeleau from the Geological Survey of Canada, and Felix Vogel and Jason O'Brien from ECCC. This research has been financially supported by the Clean Air Regulatory Agenda of Environment and Climate Change Canada, and the Environmental

Geoscience program of Natural Resources Canada (NRCan contribution number: 20170310). The first author dedicates this research article to Pauline Durand for her support.





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



**Table 1. Settings and conditions for wind sector-based simultaneous sampling of atmospheric nitrates.**

| Site (coordinates) | Sources | Distance Km (mean) | Sector direction; opening | Sampling period; | n | Avg T (°C) | Context |
|---|---|---|---|---|---|---|---|
| Genesee (114.14° W, 53.31° N) | Coal-fired power plants | 7–25 | NW, 35° | 30/09/2010 – 21/06/2011 | 6 | 11.7, 12.2, 5.5, -9.8, -0.9, 12.2 | 3 plants |
| Vauxhall (112.11° W, 50.06° N) | Gas compressors and cattle and swine feedlots | 12-125+; 7.5-45+ | W, 65° | 25/10/2011 – 13/12/2011 | 3 | 2.6, 0.7,-3.5 | 65+ compressors; 200+ feedlots |
| Terrace Heights (113.44° W, 53.54° N) | Urban traffic | <1-15 (4) | W, 150° | 24/07/2012 – 25/10/2012 | 4 | 20.3, 15.6, 7.9, -1.8 | Park in residential area, 3.5 km east of downtown core |
| Fort Saskatchewan (113.14° W, 53.72° N) | Chemical industries and metal refining | 3-7 (4) | NW, 88° | 12/04/2013 – 06/09/2013 | 4 | 4.3, 15.7, 16.3, 17.7 | Chemical plant and metal refinery largest NOₓ sources; fertilizer plant largest NH₃ source |
| Fort Saskatchewan (113.14° W, 53.72° N) | Fertilizers plant and oil refinery | 9-14 (11) | N, 27° | 20/09/2013 – 20/01/2014 | 1 | -8.1 | Fertilizer plant largest NH₃ and NOₓ source, oil refinery major NOₓ source |

N: number of sampling sessions. Avg T: average temperature during each of the consecutives sampling sessions.

5   **Table 2. Isotopic reproducibility (modified median absolute deviation) established using 2 to 4 parallel active CAPMoN sampling setups in seven separate sampling periods, resulting in (n) total samples.**

| N compound (n) | $\delta^{18}O$ | $\delta^{17}O$ |
|---|---|---|
| *Teflon filters* p-NO₃⁻ (19) | 2 | 1 |
| *Nylon filters* HNO₃ (18) | 1 | 0.7 |

10   **Table 3. Average oxygen isotopic ratios (‰) for NO₃⁻ sampled as gas (HNO₃), w (precipitation) and p (particulate matter) relative to VSMOW.**

| Matrix | Gas | w | p | Gas | ..w | ..p |
|---|---|---|---|---|---|---|
| Source | $\delta^{18}O$ | | | $\Delta^{17}O$ | | |
| Coal-fired power plants | 69.7 (5) | 66.1 (4) | 70.7 (4) | 25.1 (5) | 25.4 (4) | 26.6 (4) |
| Fertilizers plant & oil refinery | 63.2 (1) | 71.4 (1) | 69.5 (1) | 19.3 (1) | 26.0 (1) | 23.8 (1) |
| Chemical industries & metal refining | 65.7 (4) | 61.9 (2) | 54.6 (4) | 21.8 (4) | 21.4 (2) | 18.5 (4) |
| Gas compressors | 65.0 (2) | - | 63.1 (3) | 24.5 (2) | - | 26.4 (3) |
| City traffic | 65.7 (3) | 67.2 (2) | 59.6 (3) | 21.2 (3) | 24.4 (2) | 22.5 (3) |
| **Mean** | **66.8** | **66.0** | **62.6** | **23.0** | **24.3** | **23.4** |

(n): number of sampling periods characterized





**Table 4. Compilation of triple oxygen isotopic ranges obtained for atmospheric and emitted nitrates.**

| δ$^{18}$O (‰) | Δ$^{17}$O (‰) | Regional context | Location | Authors |
|---|---|---|---|---|
| **HNO$_3$** | | | | |
| 62.4-81.7 | 19.3-29.0 | Various contaminated sites | Alberta, Canada | *This study* |
| **p-NO$_3^-$** | | | | |
| 43-62 | 20-27 | Coast, Trinidad Head | California, USA | Patris et al. (2007) |
| 78-92 | 29.8-35.0 | High Arctic (Alert, Ellesmere Is.) | Nunavut, Canada | Morin et al. (2007) |
| 62-112 | 19-43 | Coast | Antarctica | Savarino et al. (2007) |
| 15.6-36.0 | -0.2 to 1.8 | Oil-sands mining stacks, PM 2.5 | Alberta, Canada | Proemse et al. (2012) |
| 49-86 | 19-27 | Coast (onboard sampling) | California, USA | Vicars et al. (2013) |
| 10.8-92.4 | 2.7-31.4 | Mt. Lulin, partly polluted air masses | Central Taiwan | Guha et al. (2017) |
| 48.4-83.2 | 13.8-30.5 | Various contaminated sites | Alberta, Canada | *This study* |
| **w-NO$_3^-$** | | | | |
| 66.3-84.0 | 20.2-36.0 | Shenandoah National Park | Virginia, USA | Coplen et al. (2004) |
| 70-90 | 20-30 | Bi-monthly sampling across state | New England, USA | Kendall et al. (2007) |
| 68-101 | 20.8-34.5 | Rishiri Island, polluted air masses | Northern Japan | Tsunogai et al. (2010) |
| 51.7-72.8 | 18.9-28.1 | Highway traffic emissions | Ontario, Canada | Smirnoff et al. (2012) |
| 35.0-80.7 | 15.7-32.0 | Oil-sands mining (with some dry dep) | Alberta, Canada | Proemse et al. (2013) |
| 57.4-74.4 | 19.2-30.1 | Various contaminated sites | Alberta, Canada | *This study* |
| **Undifferentiated and Bulk NO$_3^-$** | | | | |
| 60-95 | 21-29 | Polluted coastal area & Remote land | California, USA | Michalski et al. (2004) |
| 57-79 | 22-32 | High Arctic | Nunavut, Canada | Morin et al. (2008) |
| 36-105 | 13-37 | Marine boundary layer | 65S to 79N Atlantic | Morin et al. (2009) |
| 56.6-82.3* | 16.7-30.2* | Various contaminated sites | Alberta, Canada | *This study* |

Note: isotopic values rounded at unit are from published graphs (except for O values with actual precision at unit in Morin et al., 2007).
*Calculated using weighted averages of HNO$_3$ and p-NO$_3$ isotopic results.

**Table 5. Main reactions producing atmospheric nitrates (Zel'dovich, 1946; Lavoie et al., 1969; Erisman and Fowler, 2003; Michalski et al., 2003; Morino et al., 2006; Morin et al., 2007; Stroud, 2008; Michalski et al., 2014) Reactions 1, 9-12 can occur any time.**

| Daytime - Summer | Nighttime - Winter |
|---|---|
| (R1) $O_2 + Q \rightarrow O + O + Q$ ; $N_2 + O \rightarrow NO + N$ ; $N + O_2 \rightarrow NO + O$ | |
| (R2) $O + O_2 + M \rightarrow O_3$ ; $NO + O_3 \rightarrow NO_2 + O_2$ | |
| (R3) $NO + RO_2 \rightarrow NO_2 + RO$ | |
| | (R4) $NO_2 + O_3 \rightarrow NO_3 + O_2$ |
| (R5) $NO_2 + h\nu$ (sunlight) $\rightarrow NO + O$ | |
| | (R7) $NO_2 + NO_3^- \Leftrightarrow N_2O_5$ |
| (R6) $NO_2 + OH + M \rightarrow HNO_3 + M$ | (R8) $N_2O_5 + H_2O$(surface) $\rightarrow 2HNO_3$ (aq)* |
| (R9) $HNO_3$(g) $\Leftrightarrow HNO_3$(aq)* $\rightarrow NO_3^-$(aq)* + H$^+$(aq) | |
| (R10) $HNO_3$(g) + $NH_3$(g) $\Leftrightarrow NH_4NO_3$(s) | |
| (R11) $HNO_3$(g) + $CaCO_3$(s) $\rightarrow Ca(NO_3)_2$(s) + HCO$_3$ | |
| (R12) $NO_3^-$ + HC;(CH$_3$)$_2$S $\rightarrow HNO_3$ + products | |

*Q is a stable molecule of high energy; M is either $O_2$ or $N_2$; $RO_2$ stands for both $HO_2$ and alkyl peroxy. HC stand for hydrocarbons. *This aqueous nitrate may be on a particle.*





**Table 6. Correlations of NO$_3^-$ isotopic results (‰) with meteorological parameters and concentration (or ratio) of co-contaminants.**

| | Relative Humidity | | Temperature | | Daylight (fraction) | | PM | SO$_2$ | O$_3$ | | NOx/O$_3$ | |
|---|---|---|---|---|---|---|---|---|---|---|---|---|
| | r | R$^2$ | r | R$^2$ | r | R$^2$ | r | r | r | R$^2$ | r | R$^2$ |
| HNO$_3$ | | | | | | | | | | | | |
| δ$^{18}$O | **0.8** | 0.59 | -0.4 | | -0.3 | | 0.1 | 0.0 | -0.29 | | 0.4 | 0.12 |
| n | 8 | | 15 | | 15 | | 13 | 13 | 13 | | 13 | |
| Δ$^{17}$O | 0.6 | | **-0.5** | 0.24 | -0.4 | | 0.4 | 0.3 | -0.03 | | 0.2 | 0.06 |
| n | 8 | | 15 | | 15 | | 13 | 13 | 13 | | 13 | |
| p-NO$_3^-$ | | | | | | | | | | | | |
| δ$^{18}$O | **0.9** | 0.79 | **-0.6** | 0.34 | **-0.6** | 0.35 | 0.1 | 0.5 | **-0.61** | 0.38 | **0.6** | 0.39 |
| n | 7 | | 15 | | 15 | | 12 | 12 | 12 | | 12 | |
| Δ$^{17}$O | **0.9** | 0.73 | **-0.6** | 0.34 | **-0.7** | 0.44 | 0.0 | 0.5 | -0.47 | | **0.7** | 0.45 |
| n | 7 | | 15 | | 15 | | 12 | 12 | 12 | | 12 | |

In **bold** are the significant correlation coefficients, equal or above the 95 % significance value





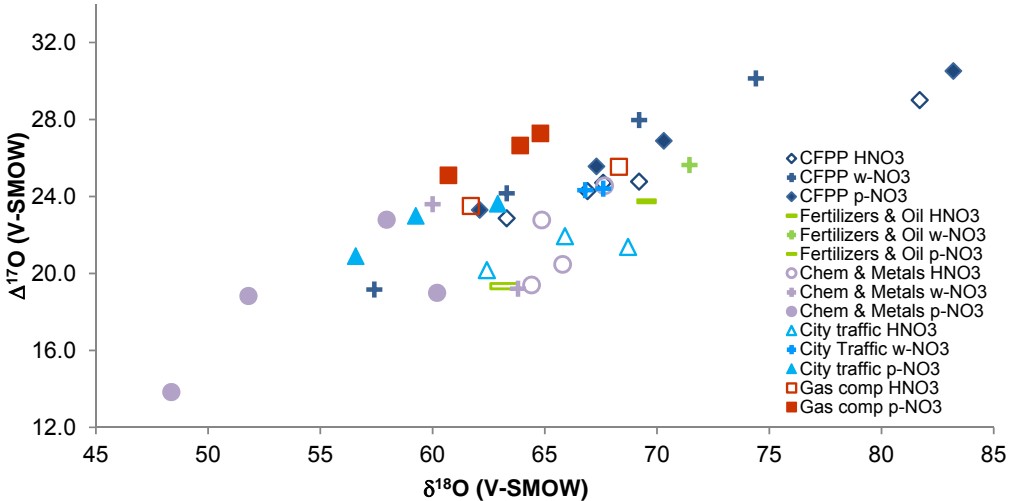

**Figure 1: Triple oxygen isotopic results (‰) obtained for simultaneously sampled atmospheric HNO₃ (empty symbols), w-NO₃⁻**

**(crosses) and p-NO₃⁻ (full symbols) downwind of the various sources.**







**Figure 2: Triple O isotopic results (‰) obtained for simultaneously collected atmospheric HNO₃ (A), w-NO₃⁻ (B) and p-NO₃⁻ (C), in**

5    **Alberta, identified by sampling periods (cold months - blue; warm months - red).**





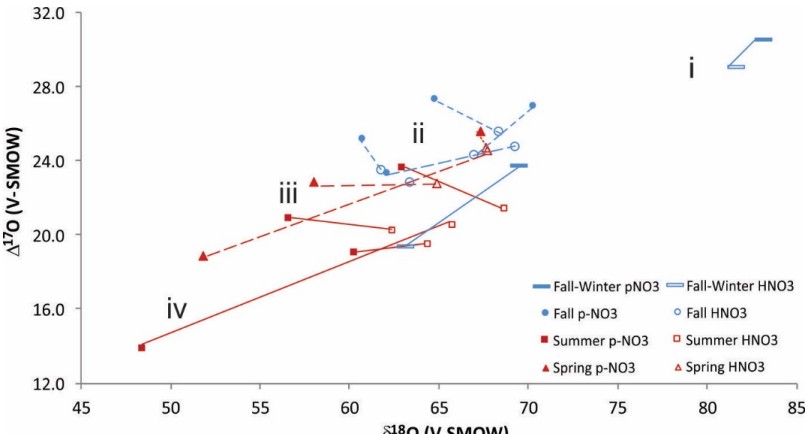

**Figure 3: Line-connected $\delta^{18}O$ and $\Delta^{17}O$ values (‰) for simultaneously collected HNO₃ and p-NO₃⁻ from cold and warm sampling periods.**

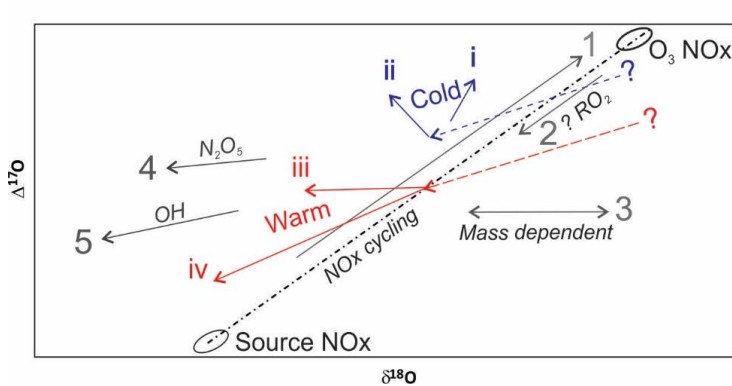

**Figure 4: Schematic outline of main steps in the production of Alberta nitrates: NOₓ-O₂-O₃ photochemical cycle (1) and reaction with RO₂ (2) modify NOₓ source signals (R2, R3, R9); oxidation of NOₓ produces HNO₃ along the N₂O₅ (4) or OH (5) pathways. The grey line represents NOₓ from photochemical cycling with O₂ and O₃ (Michalski et al., 2014). The direction of arrows 1 to 5 indicates how the isotopic values would evolve along the different chemical reactions; the positions of these arrows are arbitrary.**

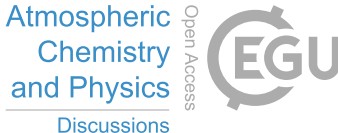



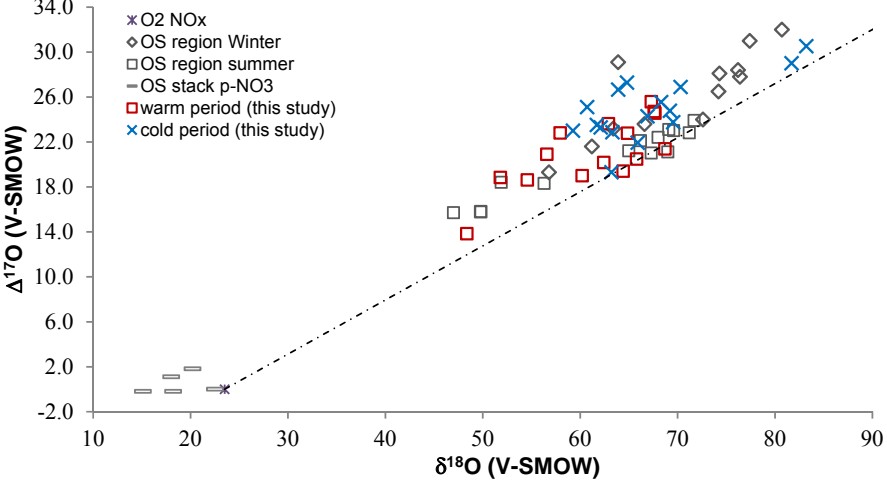

**Figure 5: Isotopic ratios (‰) for atmospheric p-NO₃⁻, w-NO₃⁻ and HNO₃ samples in cold and warm periods from central and southern Alberta (this study), compared with previously published winter and summer bulk and throughfall deposition samples from the oil sands (OS) region from northern Alberta (Proemse et al., 2013), and p-NO₃⁻ in-stack emissions data for an OS**

5 **upgrader located in the same region (Proemse et al., 2012). The grey dotted line connects NOₓ from theoretical combustion with O₂ isotopic composition and at isotopic equilibrium with tropospheric O₃ (Michalski et al., 2014).**

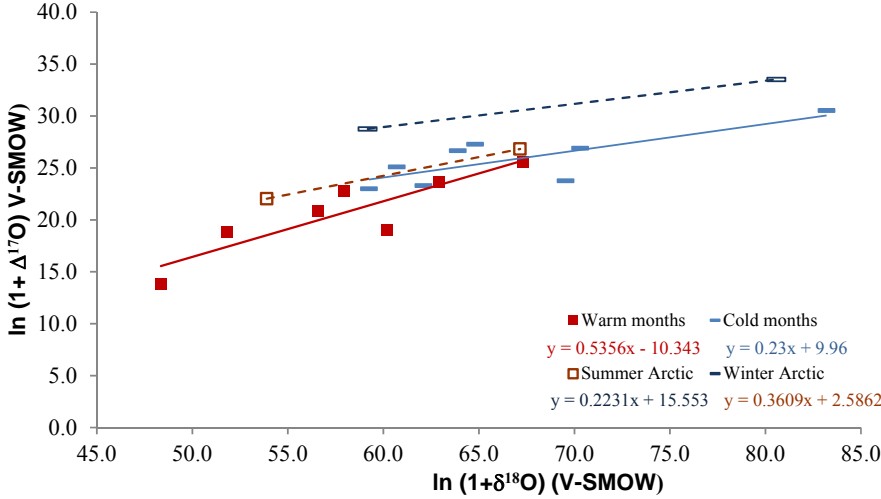

**Figure 6: Isotopic results (‰) for p-NO₃⁻ identified by sampling periods (solid lines), compared with summer and winter trends**

10 **obtained for Arctic sites (dashed lines; derived from ln (1+ δ) in Morin et al., 2008).**