# Peer review of "The $\Delta^{17}\text{O}$ and $\delta^{18}\text{O}$ values of atmospheric nitrates simultaneously collected downwind of anthropogenic sources – Implications for polluted air masses"

_Atmospheric Chemistry and Physics, 2017_

## Referee Comment (RC1) · Anonymous Referee #3 · 21 Jan 2018

The manuscript presents a new data set on the isotopic oxygen composition of nitrates in the Alberta region, Canada. It focuses specifically on the speciation of nitrate (aerosols, gases, wet phases) in conjunction with a potential source effect. The manuscript can be considered as the second part of a previous manuscript published in Atmospheric Environment (Savard et al. 2017, doi: 10.1016/j. atmosenv. 2017.05.010) which dealt only with the 15N/14N ratio of the same samples. As a first question, I wonder why the authors did not submit this second part to AE for coherency reasons or add this part to above mentioned reference.

Generally speaking, I find the article unclear and confusing, with too many figures and

tables that are not all very informative and easy to read. The explanations given are often ad hoc and not supported by strong observations, experiments or theory. Overall, the article is not of sufficient interest with new and strong novelty to recommend its publication in ACP. A major flaw of the paper is the angle taken by the authors to present and interpret their data in relation with a source effect as they did in Savard et al. (2017). It is well accepted by the community that the oxygen isotopes of nitrate are driven by oxidations and not by source effect, an idea back up by a large number of experiments and observations from the first studies (Michalski et al. 2003) to most recent ones (Guha et al. 2017). The authors should have eliminated the source effect in one or two sentences and concentrated on the oxidation mechanism by adding ancillary data such as NOx, O3 concentrations, photo-dissociation rates such JNO2 an/ d/or modeling.

The sampling protocols are poorly described. Blanks are not given, neither pumped volumes. No filter breakthrough, saturation, interference, efficiency is evaluated (see Talbot et al. 1990 for the use of nylon filter), especially in response to RH which is known to greatly influence volatilization of p-NO3 (Cheng et al., 2012) and HNO3 collection efficiency (Appel et al., 1980) on filters. Actually, such samplings artefacts can alternatively be an argument to explain the tight correlation observed between HNO3/p-NO3 isotopes and RH (Table6). It is also surprising to see the use of filter pack system to differentiate p-NO3 and HNO3 collection as most modern systems and networks use impregnated denuder systems (Cheng et al., 2012, ChemComb (Thermo Fisher scientific), MARGA (Metrohm) or URG gas-aerosols denuder samplers)) to avoid loss p-NO3 by H2SO4 acidification or gain of HNO3 by adsorption on collected alkaline aerosols. Location descriptions and context refers systematically to the Savard et al, 2017 papers which does not help to contextualize what the data plotted really mean. Samples cover different total air sampling time, from 21 to 360h and deployment times. We don't know if the sampling is dominated by nighttime or daytime chemistry, if they are rich/poor NOx/O3 atmospheres. Replicated samples were pooled at two sites (Genesee and Vauxhall) making even more difficult to know what plotted data really represent.

Section 3.3 is useless considering what the authors say in the first line of 3.2. It is thus detrimental to the understanding of the study to see an idea accepted by the whole community, namely that oxygen isotopes of nitrate are controlled by oxidation, starting to appear in the middle of the discussion. Discussion about source-driven effect should be evacuated as soon as possible with no more than one/two sentences, such as "we did not observe any significant correlations between O-isotopes and source types or wind direction".

The discussion about the different oxidation pathways to explain the season trends is classic and does not bring any new idea or interpretation. The only original observation is the difference in isotopic compositions between HNO3 and nitrate but it is questionable given the above reserve mentioned. Moreover, there is no systematic trend about HNO3 being enriched or depleted as function of season and with respect to p-NO3. In figure 3, there is few cases where summer p-NO3 have higher  $\Delta$ 170 than HNO3. It is thus difficult to understand why authors want to explain the greater  $\Delta$ 170 of HNO3 in summer over p-NO3. Furthermore, the discussion falls short to give an acceptable explanation (lines 10 to 25 of page 11). The idea that NO2 is not in isotopic equilibrium with O3 in summer is odd. First if equilibrium is not reached, it should be amplified in winter, not in summer when O3 is at max (Angle et al., 1989) and photolysis at its peak. Moreover, NO2 is the precursor of HNO3 and p-NO3, if not in equilibrium it should impact equally HNO3 and p-NO3. To twist this basic idea, the authors claim that HNO3 is faster scavenged from the atmosphere than p-NO3 but they have no quantitative data to show that is realistic in their environmental context. Neither the authors tested the hypothesis that NO2 is indeed not in equilibrium with O3. If Michalski et al. (2014) showed that the time-scale for equilibrium is strongly dependent on local sunlight conditions and NOx/O3 ratio and can be longer than 1h, they fall short to tell us why isotope equilibrium will take longer than chemical steady state (is it due to the time for ozone or NO2 to reach its isotopic equilibrium composition? or unrealistic O/O3/NO/NO2 ratios after model initialization since chemical steady state will be reached in min and will radically change the NO2/O3 ratio?). In another study, Morin et al. (2011) using

СЗ

a true atmospheric model modeled  $\Delta$ 170 of NO2 using different realistic atmospheric conditions and environments. They showed that NO2 is largely at isotopic equilibrium except during few night hours but with little impact on prognosticated  $\Delta$ 170 of nitrate (1 to 2 ‰ at most). Clearly, this section needs more and deeper investigations and critical review of published works.

Explanation of correlations with meteorological parameters are ad hoc and rough with a weak constrain on possible mechanisms. For instance, correlations with RH and T can be the result of the winter/summer meteorology. Summer is more oxidant but also warmer, sunnier and lower RH. Should all correlations be interpreted, as much of them are not independently related? Correlations with co-pollutants are contradictory as mentioned by the authors (lines 27-35, page 12) and lead to no strong conclusions. In this regard and in my view, the authors should have reported O3, NOx and JNO2 time-series to give some context. Only gross correlations are reported with most the variables interdependent.

There is other imperfection that bother me. For instance, what was a hypothesis at the beginning (the none equilibrium of NO2 with O3) has now become a certainty (line 6 page 13). Finally, the idea that low values of  $\Delta$ 17O can be linked to the rapid oxidation of anthropogenic NOx is attractive but would have merited more investigation such as following for example the NOx/NO3- ratio to give some clue about the aging of the air masses.

For all these reasons, I do not support the publication in acp.

Talbot, R. W., Vijgen, A. S., and Harriss, R. C.: Measuring tropospheric HNO3: Problems and prospects for nylon filter and mist chamber techniques, Journal of Geophysical Research: Atmospheres, 95, 7553-7561, 10.1029/JD095iD06p07553, 1990. Cheng, Y., Duan, F.-k., He, K.-b., Du, Z.-y., Zheng, M., and Ma, Y.-I.: Sampling artifacts of organic and inorganic aerosol: Implications for the speciation measurement of particulate matter, Atmos. Environ., 55, 229-233, https://doi.org/10.1016/j.atmosenv.2012.03.032, 2012. Appel, B. R., Wall, S. M., Tokiwa, Y., and Haik, M.: Simultaneous nitric acid, particulate nitrate and acidity measurements in ambient air, Atmospheric Environment (1967), 14, 549-554, https://doi.org/10.1016/0004-6981(80)90084-0, 1980. Michalski, G., Scott, Z., Kabiling, M., and Thiemens, M. H.: First measurements and modeling of  $\Delta$ 170 in atmospheric nitrate, Geophys. Res. Lett., 30, 1870, 10.1029/2003gl017015, 2003. Guha, T., Lin, C. T., Bhattacharya, S. K., Mahajan, A. S., Ou-Yang, C.-F., Lan, Y.-P., Hsu, S. C., and Liang, M.-C.: Isotopic ratios of nitrate in aerosol samples from Mt. Lulin, a high-altitude station in Central Taiwan, Atmos. Environ., 154, 53-69, https://doi.org/10.1016/j.atmosenv.2017.01.036, 2017. Angle, R. P., and Sandhu, H. S.: Urban and rural ozone concentrations in Alberta, Canada, Atmospheric Environment (1967), 23, 215-221, https://doi.org/10.1016/0004-6981(89)90113-3, 1989. Michalski, G., Bhattacharya, S. K., and Girsch, G.: NOx cycle and the tropospheric ozone isotope anomaly: an experimental investigation, Atmos. Chem. Phys., 14, 4935-4953, 10.5194/acp-14-4935-2014, 2014. Morin, S., Sander, R., and Savarino, J.: Simulation of the diurnal variations of the oxygen isotope anomaly ( $\Delta$ 170) of reactive atmospheric species, Atmos. Chem. Phys., 11, 3653-3671, 10.5194/acp-11-3653-2011, 2011.

---

## Author Comment (AC1) · 29 Jan 2018

Point-by-point REPLY to Anonymous Referee #3 The manuscript presents a new data set on the isotopic oxygen composition of nitrates in the Alberta region, Canada. It focuses specifically on the speciation of nitrate (aerosols, gases, wet phases) in conjunction with a potential source effect. The manuscript can be considered as the second part of a previous manuscript published in Atmospheric Environment (Savard et al. 2017, doi: 10.1016/j. atmosenv. 2017.05.010) which dealt only with the 15N/14N ratio of the same samples.

1- As a first question, I wonder why the authors did not submit this second part to

[Figure]

AE for coherency reasons or add this part to above mentioned reference. rep- The option of adding this data set and interpretation to the AE paper on $\delta 15N$ values of all N-species investigated (NH3/NH4 and all nitrates) was not feasible, as it would have made a much too long article. We are convinced there is a natural separation of the two articles. They are addressing different questions: the AE article aims at evaluating the source fingerprinting potential of $\delta 15N$ values in all forms of N emission (reduced and oxidized) from various anthropogenic sources; whereas the article submitted to ACP aims at understanding better the NOx oxidation pathways and the NOx/HNO3/p-NO3 relationships.

2- Generally speaking, I find the article unclear and confusing, with too many figures and tables that are not all very informative and easy to read. The explanations given are often ad hoc and not supported by strong observations, experiments or theory. Overall, the article is not of sufficient interest with new and strong novelty to recommend its publication in ACP. rep- This article presents the first $\Delta 17O$ values in HNO3, simultaneously sampled with p-NO3. These measurements are difficult to obtain as they require elaborated field collection campaigns and state-of-the-art analytical systems. The data presented are new and they prompted a new interpretation in terms of non-equilibrated NOx-O3, suggested for the first time for field samples. For these reasons, we believe the article is worth publishing in ACP.

3- A major flaw of the paper is the angle taken by the authors to present and interpret their data in relation with a source effect as they did in Savard et al. (2017). It is well accepted by the community that the oxygen isotopes of nitrate are driven by oxidations and not by source effect, an idea back up by a large number of experiments and observations from the first studies (Michalski et al. 2003) to most recent ones (Guha et al. 2017). rep- The anonymous reviewer agrees with a key statement of the introduction in the originally submitted article that O isotopes should reflect oxidation pathways (see Introduction second paragraph; and section 4.4 line 4). Confirming no direct source effect on the O isotopes was expected, and this confirmation IS NOT the main contribution highlighted in the article. The original introduction clearly states the rationale for sampling downwind from anthropogenic source: ÂñIn those studies, $\delta$18O and 17O values were suggested to be useful to apportion the contribution of emission sources to regional atmospheric nitrate loads. However, the signals of precursor NOx emitted from the same sources may quickly get modified through isotopic equilibration with O3, so that the original source signals may be difficult to recognize.Âż In the new version of the article to be available when the open discussion period is over, we further explain the pertinence of evaluating source effects, not in terms of distinguishing the ultimate sources among themselves, but for assessing if low $\triangle$17O values previously suggested as indicative of anthropogenic emissions are characterizing some or all anthropogenic emissions sampled here. Do low $\triangle$17O values reflect a larger role of RO2 in the oxidation of anthropogenic NOx emissions in fresh plumes? This question is of interest to the scientific community and as it is still debated in the literature as (Proemse et al., 2013; Guha et al., 2017).

4- The authors should have eliminated the source effect in one or two sentences and concentrated on the oxidation mechanism by adding ancillary data such as NOx, O3 concentrations, photo-dissociation rates such JNO2 and/or modeling. rep- We have used O3 and NOx mixing ratios and presented our statistics in Table 6. The fraction of each sample collected during daylight hours (correlations also shown in Table 6) was judged to be a reasonable proxy for the amount of sample collected during active photochemistry. Detailed jNO2 calculations are of limited value for effort considering that we do not have radiation data on site to account for cloud cover. We do recognize the importance of modelling, but it was not the purpose of our research, and our data can be made available for modellers when the article is accepted (a table with all pertinent information can be placed in the supplemental information).

5- The sampling protocols are poorly described. Blanks are not given, neither pumped volumes. No filter breakthrough, saturation, interference, efficiency is evaluated (see Talbot et al. 1990 for the use of nylon filter), especially in response to RH which is

known to greatly influence volatilization of p-NO3 (Cheng et al., 2012) and HNO3 collection efficiency (Appel et al., 1980) on filters. Actually, such samplings artefacts can alternatively be an argument to explain the tight correlation observed between HNO3/p-NO3 isotopes and RH (Table6). It is also surprising to see the use of filter pack system to differentiate p-NO3 and HNO3 collection as most modern systems and networks use impregnated denuder systems (Cheng et al., 2012, ChemComb (Thermo Fisher scientific), MARGA (Metrohm) or URG gas-aerosols denuder samplers)) to avoid loss p-NO3 by H2SO4 acidification or gain of HNO3 by adsorption on collected alkaline aerosols. rep-The filter pack system is based on the ones used by two long-standing networks (Environment and Climate Change Canada's CAPMoN and the U.S. Environmental Protection Agency's CASTNET), but we can certainly provide more background about the historical testing of these filters and the rationale for their use in this study. For example, Anlauf et al. (1986) found that breakthrough was ∼3% for filter loadings up to 3 times higher than the maximum loading in this study. Filter loadings and pumped volumes can be reported with the tabulated sample and ancillary data mentioned above.

Denuders were considered but were not used, partly because of the lack of capacity and established quality control protocols at the CAPMoN laboratory. Also because of the higher potential complications due to the longer deployments in these remote locations (necessary to collect sufficient material for isotopic analysis at low ambient concentrations) compared to the typical urban networks with high concentrations that allow using denuders. We had concerns about: (a) the likely positive artefact of "passive" sampling due to diffusion into the denuder during the periods without pumping in this sector-based approach; (b) the likelihood of capturing coarse PM on the denuder if no size-selective inlet was used (which was not wanted due to the desire to capture p-NO3 on coarse PM); and (c) the higher potential for condensation and dripping within the denuders during multiple day/night cycles and resulting loss of coating/sample. While we acknowledge that small part of the HNO3 is likely volatilized p-NO3, as discussed in the last paragraph of section 3.1, fractionation during this process would

be negligible during winter sampling and bias the HNO3 $\delta$18O values low relative to p-NO3 in summer, while the observations showed the opposite seasonal pattern. In addition, this would be a mass-dependent process and therefore have no effect on the $\Delta$17O signals, so it cannot explain the correlations between RH and $\Delta$17O values.

6- Location descriptions and context refers systematically to the Savard et al, 2017 papers which does not help to contextualize what the data plotted really mean. Samples cover different total air sampling time, from 21 to 360h and deployment times. We don't know if the sampling is dominated by nighttime or daytime chemistry, if they are rich/poor NOx/O3 atmospheres. rep-We can present a location map in the supplementary information and further describe the locations and contexts of sampling if judged pertinent by the reviewers and editor. However, the main point with our sampling protocol has nothing to do with the targeted source types, but with the fact that it allows collection of true anthropogenic plumes, without changes in NOx/nitrate source, which could modify the potential oxidation pathways. The submitted article therefore describes this aspect clearly.

We have explored the relationship between the isotopic results and daylight fraction and found a significant inverse correlation with isotopic values of p-NO3, but not with HNO3 (Table 6). We can provide all the data, including available O3 and NOx concentrations, in a summary table placed in supplemental information if judged necessary (opinion of the editor/reviewers required here).

7- Replicated samples were pooled at two sites (Genesee and Vauxhall) making even more difficult to know what plotted data really represent. Each point on the plots represents a single sampling period at a given site, whether several samples were pooled or not. Where samples were not pooled, the individual data were used to estimate the reproducibility of the combined sampling and analytical approach, but in the end, the average values were plotted. In brief, last paragraph of section 2.2 of the submitted article clearly explains what the data represent.

8- Section 3.3 is useless considering what the authors say in the first line of 3.2. It is thus detrimental to the understanding of the study to see an idea accepted by the whole community, namely that oxygen isotopes of nitrate are controlled by oxidation, starting to appear in the middle of the discussion. Discussion about source-driven effect should be evacuated as soon as possible with no more than one/two sentences, such as "we did not observe any significant correlations between O-isotopes and source types or wind direction". rep-In the new version of the article to be posted when the article is accepted, we have now placed former Figure 1 in supplementary information (replaced in article by a location map), removed the emphasis previously put on the individual sources and merged together sections 3.2 and former section 3.3 for which the length is now much reduced.

9- The discussion about the different oxidation pathways to explain the season trends is classic and does not bring any new idea or interpretation. The only original observation is the difference in isotopic compositions between HNO3 and nitrate but it is question-able given the above reserve mentioned. rep-As mentioned in reply to point 5, as well as in the text, the recognized sampling artefacts cannot cause $\Delta 17O$ differences between pNO3 and HNO3.

10- Moreover, there is no systematic trend about HNO3 being enriched or depleted as function of season and with respect to p-NO3. In figure 3, there is few cases where summer p-NO3 have higher _17O than HNO3. It is thus difficult to understand why au-thors want to explain the greater _17O of HNO3 in summer over p-NO3. Furthermore, the discussion falls short to give an acceptable explanation (lines 10 to 25 of page 11). rep-The data show both positive and negative values of $\Delta 17O(HNO3)$- $\Delta 17O(pNO3)$, with a somewhat positive trend with temperature. While we hypothesize that nega-tive values may be due to the larger contribution of the N2O5+H2O heterogeneous reaction to p-NO3, we felt it was necessary to propose a mechanism for the positive differences also observed in most spring and summer samples. At this stage, to our knowledge, the best hypothesis for explaining higher $\Delta 17O$ values in HNO3 is that the

deposition of HNO3 is greater than the one of p-NO3, a difference in rates that is much stronger during summer than winter. This mechanism combined with the NOx-O3 isotopic equilibrium can explain our data set. If readers want to suggestion different lines of interpretation, we will gladly receive them.

11- The idea that NO2 is not in isotopic equilibrium with O3 in summer is odd. First if equilibrium is not reached, it should be amplified in winter, not in summer when O3 is at max (Angle et al., 1989) and photolysis at its peak. rep-Agreed, but we should note that there was not a full year of data at any single site, and the two sites where summer samples were primarily gathered were the closest to the NOx sources (Table 1). This will be added to the discussion on p. 10 regarding the NO2 estimated $\Delta$17O values.

12- Moreover, NO2 is the precursor of HNO3 and p-NO3, if not in equilibrium it should impact equally HNO3 and p-NO3. To twist this basic idea, the authors claim that HNO3 is faster scavenged from the atmosphere than p-NO3 but they have no quantitative data to show that is realistic in their environmental context. rep-We refer the reader again to articles cited in the manuscript (Zhang et., et al 2009; Benedict et al., 2013) showing the higher dry deposition rate for HNO3.

13- Neither the authors tested the hypothesis that NO2 is indeed not in equilibrium with O3. If Michalski et al. (2014) showed that the time-scale for equilibrium is strongly dependent on local sunlight conditions and NOx/O3 ratio and can be longer than 1h, they fall short to tell us why isotope equilibrium will take longer than chemical steady state (is it due to the time for ozone or NO2 to reach its isotopic equilibrium composition? or unrealistic O/O3/NO/NO2 ratios after model initialization since chemical steady state will be reached in min and will radically change the NO2/O3 ratio?). In another study, Morin et al. (2011) using a true atmospheric model modeled _17O of NO2 using different realistic atmospheric conditions and environments. They showed that NO2 is largely at isotopic equilibrium except during few night hours but with little impact on prognosticated _17O of nitrate (1 to 2 ‰ at most). Clearly, this section needs more and deeper investigations and critical review of published works. rep-The field measurements reported here were not designed to test this hypothesis (they predate the Michalski et al. 2014 paper), but we look forward to other field data to further understanding of the real-world applicability of their model. Our understanding of Michalski et al.'s results (see Fig. 8 of their paper, which bracket the concentrations of NO2 and O3 observed in the current study) is that it is simply a matter of the number of interactions between NO2 and O3 required to achieve a statistical redistribution of the heavy isotopes – by necessity, due to the fewer number of heavy isotopes, this will be many more than are required to achieve chemical equilibrium. However, we would argue that it is not necessary to provide an explanation for others' findings when they are well presented in the original paper. Note that Morin et al. (2011) did not model any fresh NOx emissions and they used a 24-hour model spin up before reporting isotopic composition of NO2. Therefore, we are not able to compare their modeling results with the nitrates collected within minutes to hours of fresh NOx emissions.

Considering that this article represents the first investigation of simultaneously sampled nitrates in precipitation, gas and particulate forms for their $\delta$18O and $\Delta$17O values, we think it deserves to be available to the large readership of ACP. We have now clarified the lines of interpretation the article provides placing less emphasis on the types of sources from which the plumes were sampled.

14- Explanation of correlations with meteorological parameters are ad hoc and rough with a weak constrain on possible mechanisms. For instance, correlations with RH and T can be the result of the winter/summer meteorology. Summer is more oxidant but also warmer, sunnier and lower RH. Should all correlations be interpreted, as much of them are not independently related? rep-We judge it pertinent to suggest an interpretation for these correlations as they relate to reactions summarized in Table 5. The text describing this interpretation is short.

15- Correlations with co-pollutants are contradictory as mentioned by the authors (lines 27-35, page 12) and lead to no strong conclusions. In this regard and in my view, the authors should have reported O3, NOx and JNO2 time-series to give some context.

Only gross correlations are reported with most the variables interdependent. rep-We are not convinced that time series of O3, NOx and jNO2 would be meaningful in interpreting these integrated and intermittent samples, which is why we used average values over the sampling times for O3 and NOx analysis. However, those average values can be reported in the proposed data table to be placed in the supplementary information. The use of daylight fraction rather than jNO2 is discussed in point 4.

16- There is other imperfection that bother me. For instance, what was a hypothesis at the beginning (the none equilibrium of NO2 with O3) has now become a certainty (line 6 page 13). rep- Good point. The previous sentence was : Âń However, NO2 not in isotopic equilibrium with O3, and/or NO reacted with RO2 significantly influenced the overall results.Âż The sentence now reads: However, NO2 not in isotopic equilibrium with O3, and/or NO reacted with RO2 may have significantly influenced the overall results.Âż

17- Finally, the idea that low values of _17O can be linked to the rapid oxidation of anthropogenic NOx is attractive but would have merited more investigation such as following for example the NOx/NO3- ratio to give some clue about the aging of the air masses. rep- This is a good suggestion. A technique for actively sampling integrated NO2 and NO concentrations was developed with some success through the course of the study, but since it was an evolving methodology we have acceptable NO2 concentrations only at 2 of the 4 sites, both in the Edmonton urban area. Therefore, we could provide this ratio for the samples, where available on site, in the supplemental data table.

Cited references Anlauf, K.G., Fellin, P., Wiebe, H.A., Schiff, H.I., Mackay, G.I., Braman, R.S., Gilbert, R. A comparison of three methods for measurement of atmospheric nitric acid and aerosol nitrate and ammonium (1985) Atmospheric Environment (1967), 19 (2), pp. 325-333.

Anlauf, K.G., Wiebe, H.A., Fellin, P. Characterization of Several Integrative Sampling

Methods for Nitric Acid, Sulphur Dioxide and Atmospheric Particles (1986) Journal of the Air Pollution Control Association, 36 (6), pp. 715-723.

---

## Referee Comment (RC2) · Anonymous Referee #3 · 6 Feb 2018

Bullet points refer to first authors' reply.

1-I maintain that this article should have been submitted to AE as a Part II for coherency but this is a minor comment

2-I don't think that "new and novel" data are sufficient arguments to guaranty their publications. New and novel does not mean correct and I have major reserves about their correctness (see point 5)

3-I don't think that the authors demonstrated in any way that they have collected nitrate from specific sources whatever O isotopes track or not these sources. To pretend that, they need to provide observations that either NOx, nitrate (or any other tracers, CO, O3)

are different than background atmosphere. According to the set-up of their experiment, I have serious doubts that sampling air from hours to days will guaranty a permanent sampling of the plume emissions. Conditional sampling based on wind direction is not enough. In this way, I found the title misleading, firstly because as said above, there is no guaranty they have sampled specific anthropogenic sources and secondly, as they mentioned, the scrambling of the oxygen atoms erases source fingerprints.

4-Giving the Pearson's correlation in a table is not enough to judge the correctness of the correlation. Readers need to see the dispersion of the data and species time-series within the sampling time windows to connect sources with sampling.

5-It is wrong to think that denuders are best used in urban area. Denuders to collect HNO3 are used in the most remote regions of world (eg Antarctica, Jourdain and Legrand, 2002, Legrand et al., 2017). Denuders that are operational at 1m3/h exists (URG or Thermo Chemcomb), thus minimizing the collection time. Proper set up can limit passive sampling and restricted it to gas diffusion, exactly their purpose. The denuder tubes are the norm to collect acid gases with minimal interferences. They are promoted by the largest atmospheric aerosol networks (EMEP, EPA-method IO4-2). The method used by the authors (1st filter for p-NO3 and 2nd nylon filter for HNO3) is not the reference set up used to separate p-NO3 and HNO3. It is a set up used mainly to collect total nitrate. The difference in $\Delta17O$ between p-NO3 and HNO3 is not a guaranty that the different phases are sampled correctly. Finally, as already mentioned, the fact that a method is published and accepted does not exempt the authors to show us that they can correctly reproduce it. Authors should be able to provide the data and demonstrate that blanks, interferences, efficiencies etc. can be quantified and/or corrected (Finlayson-Pitts&Pitts, 2000).

Jourdain, B., and Legrand, M.: Year-round records of bulk and size-segregated aerosol composition and HCl and HNO3 levels in the Dumont d'Urville (coastal Antarctica) atmosphere: Implications for sea-salt aerosol fractionation in the winter and summer, J. Geophys. Res., 107, 4645, 10.1029/2002jd002471, 2002. Legrand, M., Preunkert,

[Figure]

S., Wolff, E., Weller, R., Jourdain, B., and Wagenbach, D.: Year-round records of bulk and size-segregated aerosol composition in central Antarctica (Concordia site) – Part 1: Fractionation of sea-salt particles, Atmos. Chem. Phys., 17, 14039-14054, 10.5194/acp-17-14039-2017, 2017. EMEP manual for sampling and chemical analysis, Norwegian Institute for Air Research, Kjeller, NorwayEMEP/CCC-Report 1/95, 2001. Compendium of Methods for the Determination of Inorganic Compounds in Ambient Air (EP A/625/R-96/010a) – method IO4-2 Finlayson-Pitts, B. J., and Pitts, J. N.: Chemistry of the upper and lower atmosphere: Theory, experiments and applications, Academic Press, San Diego, CA, 969 pp., 2000.

6-If the main point of the paper has nothing to do with targeted source types, title of the paper should not give the opposite impression. The authors did not convince me that they have sampled "true" anthropogenic plumes. Nothing in the presented data indicate such thing

7-When I said what the data mean, I mean what atmospheric context are they representing? Not how have they been obtained? Plotting altogether data that represent averaged hours, averaged days, mix of nighttime or daytime in different proportion etc. does not help the reader to contextualize the observations.

9- I will give one example where $\Delta$17O of nitrate can be modified. If a nitrate particles seating on the filter is hit by a sulfuric acid droplet and the pH of this sulfuric acid is low enough, then isotopic exchange between HNO3 and H2O can be triggered. I'm not saying it is what is happening with the author's sampling system but again my main point is that $\Delta$17O cannot be at the same time the causal and the effect, i.e. the variable to be explained and the variable to explain: the observed difference between $\Delta$17O HNO3 and p-NO3 can't be used as an argument to validate a sampling system. Where is the constrain showing me that such difference simply exists and it is not an artifact? For me it is a self-realization observation.

10- Again I do not see any systematic trend in $\Delta$17O difference between p-NO3 and

HNO3 with season (fig3). In summer, two out of four have Δ17O nitrate > Δ17O HNO3 and in winter they have only two events, a very weak statistic. I may not see the same data than the authors and any help from the other reviewers will be welcome. I have no explanation (as I'm not convinced by the correctness of the data by the way) but I can easily found one if I pile up few none demonstrated hypothesis, like the authors did with 1- HNO3 is formed from non-equilibrated NOx/O3 system and 2- HNO3 is faster scavenged. I can propose the formation of lower Δ17O p-NO3 by the heterogeneous reaction 2NO2 + H2O(s) –> HNO3(ads) + HONO (Finlayson-Pitts, 2009), or higher Δ17O HNO3 by NO3 + RH –> HNO3 in gas phase nighttime oxidation.

Finlayson-Pitts, B. J.: Reactions at surfaces in the atmosphere: integration of experiments and theory as necessary (but not necessarily sufficient) for predicting the physical chemistry of aerosols, PCCP, 11, 7760-7779, 10.1039/b906540g, 2009.

13- Well, I disagree again with the authors. One of the strongest argument used in this paper is to claim that NOx-O3 are not in isotopic equilibrium, using mainly Michalski paper as support. So, it is up to the authors to first question Michalski's paper and its conclusions. In Michalski, the atmospheric application of their model is really poorly described. It is not mentioned if at initialization, ozone has already its isotopes at equilibrium (as it should be in the atmosphere considering the life-time of O3 vs NOx). Yet ozone formation is the only reaction creating 17O-excess, and since chemical steady state is quickly reached, equilibrium of Δ17O among all species can't be reached faster than O3 own equilibrium time in Michalski's model. Clearly, the limiting step in Michalski's model to propagate Δ17O is ozone formation and not NOx/O3 interaction. If ozone is in isotopic equilibrium, any new population of NO2 formed by O3+NO (modulo the two-to-one atom transfer) will have the same isotopic composition that the O-atom transfer (if kinetic fractionation is neglected). It is thus simply a question of reservoir of NO2 versus flux of NO2 to reach equilibrium. Isotopic abundance has nothing to do here. Let's imagine that O3 is already in isotopic equilibrium, further formation/destruction have no effect on ozone Δ17O. Let's imagine further that NOx

and O3 are in chemical/isotopic equilibrium (new O3 formed has the same isotopic composition than consumed O3 as O3 isotope is controlled by pressure and temperature only). Suddenly, a new pool of NO is emitted. NO will be converted to NO2 by O3 contained in the surrounding atmosphere upon mixing and thus NO2 will be formed at the rate of the Leighton cycle in this system. The characteristic time of the isotopic transfer from O3 to NO2 is simply twice the time of the Leighton cycle. Obviously, a plume model is necessary to calculate air mass mixing but as a first approximation, we can assume that the plume is continuously replenished by surrounding O3 so that O3 stays constant. The characteristic time, Tau, at which the non-equilibrated isotopic NOx reservoir is replaced by the isotopic equilibrated NO2 is simply twice the size of NO2 reservoir divided by the speed of Leighton cycle, either NO+O3 reaction or JNO2 depending on the chemistry context, as one of these reactions is the limiting step. Using Michalski first simulations, NO = 23 ppbv (assumed NO2/NOx = 0.3 for fresh plume), NO2 = 10 ppbv, O3 = 50 ppbv and k = 2e-14 molecules cm-3 s-1, J = 0,007 s; then Tau = 2/J = 4,8 min. In 20 min NO2 is at 98 % in isotopic equilibrium. Using Michalski second simulations NO2= 0,03ppb, NO=0,003 ppb (assumed NO2/NOx = 0.9 for remote place), O3 = 5 ppb, Tau = 2 [NO2]/(k[NO][O3]) = 120 min; 8h to reach 98 % of equilibrium. Apparently, a much less favorable situation (due to the very low NO, strongly limiting the recycle speed) but this simulation at low ozone, 5 ppb, is taken as an illustration of Morin's observation (Morin et al., 2007). However, such situation corresponds to an ozone depletion event (due to the high concentration of bromine) for which NOx are recycled through the BrO + NO and not NO+O3 reaction. In a more rural situation (Rohrer et al., 1998), NO2 = 1,4 ppb, NO = 0,3 ppb, O3 = 25 ppb, Tau = 11 min

Rohrer, F., Brüning, D., Grobler, E. S., Weber, M., Ehhalt, D. H., Neubert, R., Schüßler, W., and Levin, I.: Mixing Ratios and Photostationary State of NO and NO2 Observed During the POPCORN Field Campaign at a Rural Site in Germany, Journal of Atmospheric Chemistry, 31, 119-137, 10.1023/a:1006166116242, 1998.

In summary, authors' reply did not change my position and did not convince me. Because the idea that 1- HNO3 has a different $\Delta 17O$ composition than p-NO2 and 2- NOx is not in isotopic equilibrium are strong and important conclusions, before propagating these idea in the literature, strong lines of evidence should be provided. I don't think the current work carries such guaranty.

---

## Author Comment (AC2) · 15 Feb 2018

1-I maintain that this article should have been submitted to AE as a Part II for coherency but this is a minor comment

REP- We do not see the advantage for the readership in this proposition. Publishing two articles dealing with distinct issues of atmospheric science isotopic applications, with several months in between, is commonly done through different journals, even if reporting data from a single region.

2-I don't think that "new and novel" data are sufficient arguments to guaranty their

publications. New and novel does not mean correct and I have major reserves about their correctness (see point 5)

REP- We mean 'New and novel' implying that the data is QA/QC checked, i.e., correct (also see point 5).

3-I don't think that the authors demonstrated in any way that they have collected nitrate from specific sources whatever O isotopes track or not these sources. To pretend that, they need to provide observations that either NOx, nitrate (or any other tracers, CO, O3) are different than background atmosphere. According to the set-up of their experiment, I have serious doubts that sampling air from hours to days will guaranty a permanent sampling of the plume emissions. Conditional sampling based on wind direction is not enough. In this way, I found the title misleading, firstly because as said above, there is no guaranty they have sampled specific anthropogenic sources and secondly, as they mentioned, the scrambling of the oxygen atoms erases source fingerprints.

REP- We did not claim a "permanent sampling" of plume emissions, as we agree that would be unrealistic. The goal was to isolate emissions from sources at their respective location, with the emissions subject to some atmospheric processing (i.e. not stack sampling). By necessity there will be contributions from background nitrate as well. For comparison, background particle nitrate and nitric acid concentrations at Wood Buffalo National Park in northern Alberta, where CAPMoN began sampling in 2014, averaged 0.071 and 0.089 $\mu$g m-3, respectively, for >2 years of monitoring. Concentrations from the conditional sampling at the sampling sites studied here were 4-40 times higher, suggesting that the collected samples have significantly greater p-NO3 and HNO3 concentrations than background. Moreover, back trajectory runs using the HYSPLIT model for every hour of sampling were used to rule out significant air mass transfer from other potential emission sources outside of the targeted wind sector, as discussed in section 2.2 of the original article.

4-Giving the Pearson's correlation in a table is not enough to judge the correctness of

the correlation. Readers need to see the dispersion of the data and species time-series within the sampling time windows to connect sources with sampling.

REP- We have carefully examined the data dispersion before interpreting the statistical correlations. The main graphs (or stats) illustrating the dispersion can be shown in the Supl. Info. If required by the editor and reviewers.

5 (merged with comment 9; see below) -It is wrong to think that denuders are best used in urban area. Denuders to collect HNO3 are used in the most remote regions of world (eg Antarctica, Jourdain and Legrand, 2002, Legrand et al., 2017). Denuders that are operational at 1m3/h exists (URG or Thermo Chemcomb), thus minimizing the collection time. Proper set up can limit passive sampling and restricted it to gas diffusion, exactly their purpose. The denuder tubes are the norm to collect acid gases with minimal interferences. They are promoted by the largest atmospheric aerosol networks (EMEP, EPA-method IO4-2). The method used by the authors (1st filter for p-NO3 and 2nd nylon filter for HNO3) is not the reference set up used to separate p-NO3 and HNO3. It is a set up used mainly to collect total nitrate. The difference in 17O between p-NO3 and HNO3 is not a guaranty that the different phases are sampled correctly. Finally, as already mentioned, the fact that a method is published and accepted does not exempt the authors to show us that they can correctly reproduce it. Authors should be able to provide the data and demonstrate that blanks, interferences, efficiencies etc. can be quantified and/or corrected (Finlayson-Pitts&Pitts, 2000). Jourdain, B., and Legrand, M.: Year-round records of bulk and size-segregated aerosol composition and HCl and HNO3 levels in the Dumont d'Urville (coastal Antarctica) atmosphere: Implications for sea-salt aerosol fractionation in the winter and summer, J. Geophys. Res., 107, 4645, 10.1029/2002jd002471, 2002. Legrand, M., Preunkert, S., Wolff, E., Weller, R., Jourdain, B., and Wagenbach, D.: Year-round records of bulk and size-segregated aerosol composition in central Antarctica (Concordia site) – Part 1: Fractionation of sea-salt particles, Atmos. Chem. Phys., 17, 14039-14054, 10.5194/acp-17-14039-2017, 2017. EMEP manual for sampling and chemical analysis, Norwegian Institute for Air Research, Kjeller, NorwayEMEP/CCC-Report 1/95, 2001. Compendium of Methods for the Determination of Inorganic Compounds in Ambient Air (EP A/625/R-96/010a) – method IO4-2 Finlayson-Pitts, B. J., and Pitts, J. N.: Chemistry of the upper and lower atmosphere: Theory, experiments and applications, Academic Press, San Diego, CA, 969 pp., 2000. 9- I will give one example where 17O of nitrate can be modified. If a nitrate particles seating on the filter is hit by a sulfuric acid droplet and the pH of this sulfuric acid is low enough, then isotopic exchange between HNO3 and H2O can be triggered. I'm not saying it is what is happening with the author's sampling system but again my main point is that 17O cannot be at the same time the causal and the effect, i.e. the variable to be explained and the variable to explain: the observed difference between 17O HNO3 and p-NO3 can't be used as an argument to validate a sampling system. Where is the constrain showing me that such difference simply exists and it is not an artifact? For me it is a self-realization observation.

REP- We have responded to points 5 and 9 together since we interpreted them as raising closely-related issues. We acknowledge that it is possible to use denuders in remote areas, our point was that there are specific and well-regarded networks of rural and remote stations that continue to use filter-based sampling. Since our system was using the established methods of one of those networks (CAPMoN), and evaluation of the method blanks, collection efficiencies and interferences have been previously reported, it seems excessive to us to require repetition of these tests in every report using the same method. Where we developed a new method (for NO and NO2 active sampling, not reported here), blanks and breakthrough tests were done and evaluated before reporting results. Again, denuders were considered but we chose not to use them for several reasons: (1) we were not certain of the denuder capacity or the ambient levels of HNO3 in this region prior to the study; (2) given the potential for long periods without flow in the conditional sampling setup, denuders open to the atmosphere would be likely to passively sample during non-pumped periods, while (3) denuders with size-selective impactors at the inlets would result in screening out nitrate on some particles, with the size cutoff varying as the pumps cycled on and off in (sometimes) 5-

min periods. Note that isotopic results based on collection with filter packs are not new. For instance, isotopic values for dry deposition (pNO3 and HNO3) actively collected with filter packs over a week have previously been reported in eastern USA (Elliott et al., JGR, vol 114, 2009).

Our primary concern with this system was the volatilization that is well documented, and that would affect both the O and N isotopes in a mass dependent and highly temperature dependent way. Therefore, as we stated, we evaluated the relative HNO3 and pNO3 $\delta$18O and $\delta$15N values (as well as $\delta$15N in NH3 and pNH4), and their pattern with temperature, to judge whether this was strongly affecting the results. We did not draw conclusions about the artifact based on $\Delta$17O values, just stated that mass-dependent volatilization would not affect the value, which is correct. While the reviewer does suggest a possible mechanism that would affect $\Delta$17O (exchange with H2O due to highly acidic particles), this scenario is unlikely in this region. Where we analyzed a complete suite of major ion data from the particle filter (2 of the 4 sites), the charge balance was always positive due to both relatively high Ca2+ and NH4+. In any case, this scenario would similarly influence pNO3 collected in a denuder-filter pack sampling system.

6-If the main point of the paper has nothing to do with targeted source types, title of the paper should not give the opposite impression. The authors did not convince me that they have sampled "true" anthropogenic plumes. Nothing in the presented data indicate such thing

REP- The main point of the article is not to address potential differences between various anthropogenic sources, but to examine isotopic trends in anthropogenic sources sampled at different periods, with the specific objective of verifying if low $\Delta$17O values exist in such contexts. Yes, they do and this finding has implications for interpreting isotopic data collected downwind from anthropogenic sources. The title refers to this aspect, which the article largely discusses.

[Figure]

7-When I said what the data mean, I mean what atmospheric context are they representing? Not how have they been obtained? Plotting altogether data that represent averaged hours, averaged days, mix of nighttime or daytime in different proportion etc. does not help the reader to contextualize the observations.

REP- Merged parallel samples (Genesee and Vauxhall) constitutes a physical average of atmospheric characteristics at a given area, which can be compared with the calculated average through 4 parallel samples (4 other sites) which only had as a goal to determine the reproducibility of our sampling and analytical protocols. We have judged this type of care determinant and crucial in guaranteeing the quality of the data. Not clear what the reviewer means in the second point. There is no other way to plot the data since each sample is integrated over a variety of conditions. We would agree that higher-frequency field measurements would add to our understanding of the processes, though it would be challenging to collect enough material for isotopic analysis as methods currently stand.

10- Again I do not see any systematic trend in 17O difference between p-NO3 and HNO3 with season (fig3). In summer, two out of four have 17O nitrate > 17O HNO3 and in winter they have only two events, a very weak statistic. I may not see the same data than the authors and any help from the other reviewers will be welcome. I have no explanation (as I'm not convinced by the correctness of the data by the way) but I can easily found one if I pile up few none demonstrated hypothesis, like the authors did with 1- HNO3 is formed from non-equilibrated NOx/O3 system and 2- HNO3 is faster scavenged. I can propose the formation of lower 17O p-NO3 by the heterogeneous reaction 2NO2 + H2O(s) –> HNO3(ads) + HONO (Finlayson-Pitts, 2009), or higher 17O HNO3 by NO3 + RH –> HNO3 in gas phase nighttime oxidation. Finlayson-Pitts, B. J.: Reactions at surfaces in the atmosphere: integration of experiments and theory as necessary (but not necessarily sufficient) for predicting the physical chemistry of aerosols, PCCP, 11, 7760-7779, 10.1039/b906540g, 2009.

REP- The trends are various and each deserves attention. We discuss all of them in

the article.

13- Well, I disagree again with the authors. One of the strongest argument used in this paper is to claim that NOx-O3 are not in isotopic equilibrium, using mainly Michalski paper as support. So, it is up to the authors to first question Michalski's paper and its conclusions. In Michalski, the atmospheric application of their model is really poorly described. It is not mentioned if at initialization, ozone has already its isotopes at equilibrium (as it should be in the atmosphere considering the life-time of O3 vs NOx). Yet ozone formation is the only reaction creating 17O-excess, and since chemical steady state is quickly reached, equilibrium of 17O among all species can't be reached faster than O3 own equilibrium time in Michalski's model. Clearly, the limiting step in Michalski's model to propagate 17O is ozone formation and not NOx/O3 interaction. If ozone is in isotopic equilibrium, any new population of NO2 formed by O3+NO (modulo the two-to-one atom transfer) will have the same isotopic composition that the O-atom transfer (if kinetic fractionation is neglected). It is thus simply a question of reservoir of NO2 versus flux of NO2 to reach equilibrium. Isotopic abundance has nothing to do here. Let's imagine that O3 is already in isotopic equilibrium, further formation/destruction have no effect on ozone 17O. Let's imagine further that NOx and O3 are in chemical/isotopic equilibrium (new O3 formed has the same isotopic composition than consumed O3 as O3 isotope is controlled by pressure and temperature only). Suddenly, a new pool of NO is emitted. NO will be converted to NO2 by O3 contained in the surrounding atmosphere upon mixing and thus NO2 will be formed at the rate of the Leighton cycle in this system. The characteristic time of the isotopic transfer from O3 to NO2 is simply twice the time of the Leighton cycle. Obviously, a plume model is necessary to calculate air mass mixing but as a first approximation, we can assume that the plume is continuously replenished by surrounding O3 so that O3 stays constant. The characteristic time, Tau, at which the non-equilibrated isotopic NOx reservoir is replaced by the isotopic equilibrated NO2 is simply twice the size of NO2 reservoir divided by the speed of Leighton cycle, either NO+O3 reaction or JNO2 depending on the chemistry context, as one of these reactions is the limiting step.

Using Michalski first simulations, NO = 23 ppbv (assumed NO2/NOx = 0.3 for fresh plume), NO2 = 10 ppbv, O3 = 50 ppbv and k = 2e-14 molecules cm-3 s-1, J = 0,007 s; then Tau = 2/J = 4,8 min. In 20 min NO2 is at 98 % in isotopic equilibrium. Using Michalski second simulations NO2= 0,03ppb, NO=0,003 ppb (assumed NO2/NOx = 0.9 for remote place), O3 = 5 ppb, Tau = 2 [NO2]/(k[NO][O3]) = 120 min; 8h to reach 98 % of equilibrium. Apparently, a much less favorable situation (due to the very low NO, strongly limiting the recycle speed) but this simulation at low ozone, 5 ppb, is taken as an illustration of Morin's observation (Morin et al., 2007). However, such situation corresponds to an ozone depletion event (due to the high concentration of bromine) for which NOx are recycled through the BrO + NO and not NO+O3 reaction. In a more rural situation (Rohrer et al., 1998), NO2 = 1,4 ppb, NO = 0,3 ppb, O3 = 25 ppb, Tau = 11 min Rohrer, F., Brüning, D., Grobler, E. S., Weber, M., Ehhalt, D. H., Neubert, R., Schüßler, W., and Levin, I.: Mixing Ratios and Photostationary State of NO and NO2 Observed During the POPCORN Field Campaign at a Rural Site in Germany, Journal of Atmospheric Chemistry, 31, 119-137, 10.1023/a:1006166116242, 1998.

REP- We do not want to discuss the fundamentals of Michalski et al.'s paper here, this is not the place. However, we trust that the conclusion of Michalski's experiments open up the possibility of seeing isotopic disequilibrium in natural samples under certain conditions. In fact, given the unknowns, the back-of-envelope calculations above (20 min and 8 h to 98% of equilibrium in the two scenarios) are roughly in agreement with the timescales shown in Michalski et al. (Fig. 8), so it is not clear why the reviewer is not comfortable with the results of their simulations. Given that transit times from the closest point sources to our measurement sites averaged 25 minutes (range 9-55), and that we were sampling the fraction of NOx that had been converted to nitrate and therefore "frozen" in Δ17O at the point of conversion, contributions from unequilibrated NOx are not ruled out by the tau of 11 minutes suggested by the reviewer for similar conditions.

We would like to be clear that we are not claiming to present definitive evidence of

this phenomenon in the atmosphere. Indeed, we do suggest in a few places in the manuscript that the contribution from enhanced RO2 could also give a similar result, as has been previously hypothesized. However, since the possibility of incomplete NOx equilibration retained in nitrate field samples was a new idea, it was highlighted. We will carefully review the wording of the document to be sure not to overstate our confidence in the mechanism, as was suggested in the earlier comments.

In summary, authors' reply did not change my position and did not convince me. Because the idea that 1- HNO3 has a different 17O composition than p-NO2 and 2- NOx is not in isotopic equilibrium are strong and important conclusions, before propagating these idea in the literature, strong lines of evidence should be provided. I don't think the current work carries such guaranty.

REP- Point 1 refers to measurements; the difference in isotopic signals is an observation, it is not an idea inferred through an interpretation. We have shown that the data are valid. Point 2 is a suggestion for which all arguments are exposed in the article; the reader gets substantial information allowing for a personal opinion to be made; this suggestion may create a debate (indeed, it already has) and spur further testing of the hypothesis through additional measurements and plume modelling, a healthy outcome in science.

---

## Referee Comment (RC3) · Anonymous Referee #1 · 16 Feb 2018

The authors present oxygen stable isotope composition ($\delta$18O and $\Delta$17O) of atmospheric-derived nitrate (nitric acid (HNO3(g)) + particulate nitrate (p-NO3-) + wet-deposited nitrate (w-NO3-)) from serval locations in the Alberta, CA. The authors suggest that they have speciated HNO3(g) and p-NO3- utilizing a filter pack method, providing separate $\delta$18O and $\Delta$17O of these two nitrate phases in order to better understand their oxidation formation pathways. The authors find that the different sampled regions did not have an influence on $\Delta$17O and $\delta$18O of nitrate but this is an expected result since the authors collected nitrate downwind of emissions sources, allowing for NOx oxidation to have an influence on $\delta$18O and $\Delta$17O, rather than directly from emission plumes. One interesting point is that the authors think that lower than expected

$\Delta$17O values during the summer may be explained due to non-equilibration of NOx with O3 during the summer, but this conclusion is not properly justified in text. This manuscript certainly has a lot of potential but there are major methodology and interruption flaws. Specifically, I have serious doubts that the authors truly achieved nitrate phase speciation due to the potential of p-NO3- volatilization. Additionally, I found the authors conclusion not often properly justified. It is also unclear whether the nitrate that was sampled from targeted emission sources was entirely derived from these targeted sources due to atmospheric lifetime and transit times, as suggested by the authors and explicitly in the title. It is my opinion that this manuscript does not expand upon the knowledge of atmospheric nitrate dynamics and/or its isotopic compositions, and that due to serious methodology concerns, their data should be interrupted as bulk nitrate (p-NO3- + HNO3(g)) rather than speciated, requiring significant reframing. For these reasons, I suggest that this manuscript should be rejected to ACP. My specific comments justifying this decision are listed below:

Comment 1: Title: I find the title to be somewhat misleading specifically "from anthropogenic sources – Implications for polluted air masses" As written this title implies direct plume emissions were sampled rather than air mass originating from anthropogenic emission regions. Due to the numerous sampling sites, I think the authors should instead highlight that this work is spatially-resolved.

Comment 2: Intro: I think the introduction could use a better framework. If speciating HNO3(g) and p-NO3- is a main motivation of this study (however likely not achieved), then I think the introduction needs to have a better critical analysis of HNO3(g) and p-NO3- dynamics in the atmosphere. What might $\delta$18O and $\Delta$17O of speciated nitrate tell the atmospheric community that isn't already known? Will this speciated work shed light into their separate formation pathways or will the phase separation reflect the inorganic thermodynamics of trace gases and PM driven by RH, temperature, [sulfate], [ammonia], etc.

Comment 3: Sampling Methodology: I think it would be useful to quantitatively demonstrate that the collected nitrate truly originated from the targeted upwind region. Specifically, the lifetimes both for NOx oxidation to nitrate and nitrate lifetime should be considered. Based on these lifetimes, how much of the sampled nitrate reasonable originated from NOx oxidized from the upwind region? Was the emitted NOx converted to nitrate from these emission regions are is the sampled nitrate a mixture of "background" nitrate originated from NOx emitted further upwind that the targeted region due to a relatively long nitrate lifetime (3-5 days). I think this is incredible important especially considered the sampling distance from the targeted sources (1 to 125 km).

Comment 4 (MAJOR): Sampling Protocols: I'm not convinced that nitrate speciation (HNO3 and p-NO3-) was actually achieved with the sampling filter pack method. The problem, as the authors have pointed out, that the collected p-NO3- can easily volatize as HNO3 that is collected on the filter designated for HNO3(g) collection. Additionally, there is the possibility of gaseous reactions on the PM filter especially as the filter accumulates PM. The authors point out that they don't expect volatilization of p-NO3- to play a major role on their results because p-NO3- isotope ratios are higher during winter than during summer and that the difference between p-NO3- and HNO3 is opposite than expected, but this is not proper justification. First, I wonder what the expected p-NO3- and HNO3 isotope difference is? This process is likely driven by an equilibrium effect rather than a kinetic effect since the volatilization of p-NO3- as HNO3 is due to the system being at non-equilibrium. I think this would change the authors expectation that the difference p-NO3- and HNO3 is driven by a kinetic effect (I assumed this was the authors assumption). Additionally, which "isotopic ratios" did the authors use to evaluate the p-NO3- and HNO3 difference? I'm assuming mass-dependent $\delta$18O but this information is not provided in text. Also, suggesting that this filter pack method has previously been used for isotopic analysis of p-NO3- and HNO3 (Elliot et al., 2009) does not mean the sampling method is correct and optimal for this isotopic analysis. Elliot et al., 2009 did not quantify this method for phase separation for isotopic analysis; thus, this argument should not be used to justify the work in this manuscript. Also, there is a general lack of information regarding the authors sampling protocols. What types

of filters were used (and size)? How were these filters prepared and processed? Were field and laboratory blanks taken? At what flow rates was sampling conducted? How was the requirement that sample air volumes be within 15% quantitatively determined and could the authors elaborated on the experienced flow problems, specifically was this related to mechanical malfunctions or filter build-up? If samplers were placed out for an extended period of time, how might potential for passive HNO3(g) absorption on the Nylon filter play a role in their results?

Comment 5: Analytical procedures: I also mind a general lack of information in the analytical procedures. First, how were concentrations measured? Importantly, was nitrite detected and removed from the samples? Even if NO2- was minor say 5% relative to NO3-, due to its rapid exchange with water below pH of 10, this could have artificially lowered the measured $\Delta$17O of the interpreted HNO3(g) or p-NO3-. As an example, a 5% NO2- contribution would have lowered the $\Delta$17O of the targeted nitrate (assuming $\Delta$17O $\sim$ 30‰ by 1.5‰ and could be the primary driver behind the $\Delta$17O difference in the "speciated" nitrate. Additionally, which chemical conversion method was used? Azide/Acetic Acid Buffer? If so, please cite the appropriate references (i.e. McIlvin and Altabet, 2005). How might using the long-form $\Delta$17O definition (ln(1+d17O/1000)-0.516*ln(1+$\delta$18O/1000)) impact the authors comparison to data using the more commonly used linear $\Delta$17O definition (d17O-0.52*$\delta$18O) (i.e. Table 4)?

Comment 6: Sections 3.2-3.3: I'm a little unsure as to why the authors are spending so much time on the $\delta$18O + $\Delta$17O "source effect". Their collections were from polluted air masses not direct stack emissions, such that $\delta$18O and $\Delta$17O should effectively be wiped of any "source effect" due to the rapid equilibration of NOx and its oxidants. The authors suggest in 3.2, that due to elevated $\delta$18O + $\Delta$17O there isn't a source effect (this is not surprising or a novel finding but expected), but extend the discussion of source effects in 3.3 despite ruling them out in 3.2. This seems a bit odd to me, and I think it would serve this manuscript better to simplify these sections into 1, removing source effect discussions. Instead the authors should focus on the unique oxidation

chemistries of the polluted air masses to understand how differences in NOx oxidation cycling and post NO2 reactions would have altered $\delta18O + \Delta17O$ rather than a source-derived $\delta18O + \Delta17O$ effect, but again this is complicated as previously mentioned because of the nitrate lifetime problem. We don't know that the nitrate sampled is from the targeted source area.

Comment 7 (MAJOR): Section 4.1: The Alberta nitrate $\delta18O$ and $\Delta17O$ relationships appear linear despite the author's claim to the contrary. Can the authors include regression statistics so that their argument is supported quantitively rather than qualitatively? Much of the authors $\Delta17O$ range calculations and justifications are ad hoc. Can the authors properly justify their assumptions made in this calculation, specifically "50% contribution from each pathway for summer"? Additionally, can the authors propagate the error made in the suggested $\Delta17O$ ranges? The authors indicate that the $\Delta17O$ range "shrinks" during winter but their calculations indicate a larger range during the winter (winter: 26 to 37‰ summer: 20 to 29‰. The authors conclusion that $\Delta17O$ of NO2 is not equal to the asymmetrical O3 is not new but rather expected, due to VOC oxidation contributions that have an NO oxidation branching ratio of 70 to 80%. Perhaps the authors should retry their calculations utilizing a more realistic approximated $\Delta17O$ of NO2. Also, could the authors compare their $\Delta17O$ in this region with the global $\Delta17O$ model? Again, the calculated transit times of 9 minutes to 4 hours, indicates that not all of the sampled nitrate is derived from the targeted upwind region due to the chemical lifetimes of NOx and atmospheric lifetime of nitrate. I find it hard to believe any of the interpretation on $\delta18O$ and $\Delta17O$ differences between p-NO3- and HNO3(g) because this speciation was likely not truly achieved given the method concerns already raised in this review and others. I recommend that this speciation discussion should be removed and $\delta18O$ and $\Delta17O$ interruption should focus on total nitrate relative to wet-nitrate (which was hardly discussed in this manuscript!)

Comment 8: In general, I find the figures and tables difficult to read and interrupt (especially Figs. 3 and 4)

Comment 9: The authors findings that meteorological parameters often correlate with HNO3, p-NO3- and their isotopic compositions isn't surprising (particularly phase separation) due to the well-established thermodynamic equilibrium of HNO3 and p-NO3- that determines this phase separation. This point however, directly contradicts that authors claim that reaction pathways (i.e. NO2 + OH vs N2O5 heterogenous rxn) had a significant role on the observed speciation and isotopic composition in Section 4.1.

Comment 10: Overall, I'm not convinced that during the summer, source effects lowered the anthropogenic originating nitrate $\Delta17O$ values. The simplest explanation for this observation should be NO oxidation contributions from RO2. Until the authors can explicitly rule out the "oxidation chemistry effect" by modeling or empirical evidence, I don't think the authors suggested conclusion should be drawn.

---

## Author Comment (AC3) · 19 Feb 2018

Comment 1: Title: I find the title to be somewhat misleading specifically "from anthropogenic sources – Implications for polluted air masses" As written this title implies direct plume emissions were sampled rather than air mass originating from anthropogenic emission regions. Due to the numerous sampling sites, I think the authors should instead highlight that this work is spatially-resolved.

REP – We can modify the title to: "The D17O and d18O values of simultaneously collected atmospheric nitrates from anthropogenic air masses"

[Figure]

Comment 2: Intro: I think the introduction could use a better framework. If speciating HNO3(g) and p-NO3- is a main motivation of this study (however likely not achieved), then I think the introduction needs to have a better critical analysis of HNO3(g) and p-NO3- dynamics in the atmosphere. What might _18O and _17O of speciated nitrate tell the atmospheric community that isn't already known? Will this speciated work shed light into their separate formation pathways or will the phase separation reflect the inorganic thermodynamics of trace gases and PM driven by RH, temperature, [sulfate], [ammonia], etc.

REP – We can do that in the final version.

Comment 3: Sampling Methodology: I think it would be useful to quantitatively demonstrate that the collected nitrate truly originated from the targeted upwind region. Specifically, the lifetimes both for NOx oxidation to nitrate and nitrate lifetime should be considered. Based on these lifetimes, how much of the sampled nitrate reasonable originated from NOx oxidized from the upwind region? Was the emitted NOx converted to nitrate from these emission regions are is the sampled nitrate a mixture of "background" nitrate originated from NOx emitted further upwind that the targeted region due to a relatively long nitrate lifetime (3-5 days). I think this is incredible important especially considered the sampling distance from the targeted sources (1 to 125 km).

REP – We aimed at isolating emissions from sources at their respective location, with the emissions subject to some atmospheric processing (i.e., not stack sampling). By necessity, it is clear that background contributed to the sampled load, but not in significant proportions. For comparison, background particle nitrate and nitric acid concentrations at Wood Buffalo National Park in northern Alberta, where CAPMoN began sampling in 2014, averaged 0.071 and 0.089 $\mu$g m-3, respectively, for >2 years of monitoring. Concentrations from the conditional sampling at the sampling sites studied here were 4-40 times higher, suggesting that the collected samples have greater p-NO3 and HNO3 concentrations than background. Moreover, back trajectory runs using the HYS-PLIT model for every hour of sampling served to rule out significant air mass transfer

from other potential emission sources outside of the targeted wind sector, as discussed in section 2.2 of the original article.

Comment 4 (MAJOR): Sampling Protocols: I'm not convinced that nitrate speciation (HNO3 ad p-NO3-) was actually achieved with the sampling filter pack method. The problem, as the authors have pointed out, that the collected p-NO3- can easily volatize as HNO3 that is collected on the filter designated for HNO3(g) collection. Additionally, there is the possibility of gaseous reactions on the PM filter especially as the filter accumulates PM. The authors point out that they don't expect volatilization of p-NO3- to play a major role on their results because p-NO3- isotope ratios are higher during winter than during summer and that the difference between p-NO3- and HNO3 is opposite than expected, but this is not proper justification. First, I wonder what the expected p-NO3- and HNO3 isotope difference is? This process is likely driven by an equilibrium effect rather than a kinetic effect since the volatilization of p-NO3- as HNO3 is due to the system being at non-equilibrium. I think this would change the authors expectation that the difference p-NO3- and HNO3 is driven by a kinetic effect (I assumed this was the authors assumption). Additionally, which "isotopic ratios" did the authors use to evaluate the p-NO3- and HNO3 difference? I'm assuming mass-dependent _18O but this information is not provided in text. Also, suggesting that this filter pack method has previously been used for isotopic analysis of p-NO3- and HNO3 (Elliot et al., 2009) does not mean the sampling method is correct and optimal for this isotopic analysis. Elliot et al., 2009 did not quantify this method for phase separation for isotopic analysis; thus, this argument should not be used to justify the work in this manuscript. Also, there is a general lack of information regarding the authors sampling protocols. What types of filters were used (and size)? How were these filters prepared and processed? Were field and laboratory blanks taken? At what flow rates was sampling conducted? How was the requirement that sample air volumes be within 15% quantitatively determined and could the authors elaborated on the experienced flow problems, specifically was this related to mechanical malfunctions or filter build-up? If samplers were placed out for an extended period of time, how might potential for passive HNO3(g) absorption on

the Nylon filter play a role in their results?

REP - A potential complication from the air sampling method could arise if there was significant dissociation of ammonium nitrate on the particle filter into nitric acid and ammonia, with subsequent collection on the downstream nitric acid and ammonia filters. This may result in isotopic fractionation between the particle and gaseous components (Heaton et al., 1997; Kundu et al., 2010). For instance, ammonium nitrate formed on the Teflon particle filters may dissociate and release NH3 and HNO3, particularly with changes in the ambient temperature and humidity. In that equilibrium process, the p-NH4+ and p-NO3- remaining on the Teflon filters would preferentially retain 15N, and 15N-depleted HNO3 and NH3 would accumulate on the downstream nylon and citrated Whatman filters. This phenomenon could generate artificially low $\delta$15N values in HNO3 and NH3 and high values in the particulate ions. At higher temperature, the effect should increase due to enhanced volatilization, assuming that the effect of temperature on the solid/gas or solution/gas equilibrium is larger than the effect of temperature on the fractionation factor of the atmospheric reactions (typically smaller as the temperature increases; Savard et al., 2017; Atm Env). In the present study, when all source types are considered, or when several data from a single source are examined, the opposite relationship is observed; NH4 $\delta$15N values are low at higher temperature (10 to 20 C), and high at low temperature (-10 to 0 C). In contrast, the NH3 values are slightly higher or practically unchanged at high temperature. In addition, there is no observed systematic change of HNO3 $\delta$15N values with temperature. We therefore conclude that any dissociation of ammonium nitrate from the Teflon filter did not significantly alter the isotopic values in the samples. While we acknowledge that small part of the HNO3 is may derive from volatilized p-NO3, as discussed in the last paragraph of section 3.1 of the submitted article, fractionation during this process would be negligible during winter sampling and bias the HNO3 $\delta$18O values low relative to p-NO3 in summer, while the observations showed the opposite seasonal pattern (Figure 3 of submitted article). In addition, this would be a mass-dependent process and therefore have no effect on the $\Delta$17O signals.

Interactive
comment
To wrap up, both $\delta 15N$ and $\delta 18O$ values suggest that the data set is valid: dissociation of ammonium nitrate and volatilization are reduced to a minimum during winter (below $0°C$), and these processes cannot create the $\delta 18O$ trends observed during summer.

Comment 5: Analytical procedures: I also mind a general lack of information in the analytical procedures. First, how were concentrations measured? Importantly, was nitrite detected and removed from the samples? Even if $NO2-$ was minor say 5% relative to $NO3-$, due to its rapid exchange with water below pH of 10, this could have artificially lowered the measured _17O of the interpreted $HNO3(g)$ or $p-NO3-$. As an example, a 5% $NO2-$ contribution would have lowered the _17O of the targeted nitrate (assuming _17O _ 30‰ by 1.5‰ and could be the primary driver behind the _17O difference in the "speciated" nitrate. Additionally, which chemical conversion method was used? Azide/Acetic Acid Buffer? If so, please cite the appropriate references (i.e. McIlvin and Altabet, 2005). How might using the long-form _17O definition (ln(1+d17O/1000)-0.516*ln(1+_18O/1000)) impact the authors comparison to data using the more commonly used linear _17O definition (d17O-0.52*_18O) (i.e. Table 4)?

REP – We can expand the analytical description as follows: We characterized the $\delta 15N$ ratios of NH3, w-NH4, p-NH4 and NOx, and the triple isotopic ratios ($\delta 15N$, $\delta 17O$, $\delta 18O$) of w-NO3, HNO3 and p-NO3-. The present article deals solely with the $\delta 18O$ values obtained for oxidized species. We treated the samples with the chemical conversion and thermal decomposition of N2O protocols, providing the ability to simultaneously analyze low-concentration N- and O-containing species (Smirnoff et al., 2012). The different preparation steps involved conversion of ammonium-containing and nitrate-containing samples into nitrite (NO2-), using sodium bromate and bromide, and a cadmium column, respectively. The final preparation step involved using sodium azide to ultimately produce N2O (Smirnoff et al., 2012). In addition to these procedures, which prepared samples for the NH3, NH4, NOx, p-NO3-, and precipitation NO3 isotopic analyses, HNO3 samples collected on nylon filters were treated as well. They were reduced to NO2- using the Cd-column before being converted into N2O.

NO2 resin cartridges and NO filters were sealed and shipped to Maxxam, along with field blanks, for extraction by a proprietary method. The resulting solutions of NO2- were converted into N2O using sodium azide (for details consult Smirnoff et al., 2012). All extracted N2O was analyzed with the pre-concentration/gold furnace-IRMS system developed at the Geological Survey of Canada (Smirnoff et al., 2012). This approach allows the spectrum of $\delta$15N, $\delta$17O and $\delta$18O ratios from O-bearing N-species to be determined in samples containing as little as 37.5 nmol of N (15 mL final solution). The USGS-34, USGS-35, USGS-32 nitrate reference materials were used and processed exactly the same way as the samples, i.e., converted from nitrate to nitrite, then to N2O. The laboratory analytical precision (average of replicates) determined during the present study using the described analytical procedures was 0.6 ‰ for $\delta$18O and $\delta$17O values in gaseous (n=12) and solid nitrates (n=20). For w-NO3, analytical duplicates gave 0.6 and 0.5 ‰ for $\delta$18O (n=3) and $\delta$17O (n=4) values, respectively. The $\Delta$17O values are defined as ln (1+$\delta$17O/1000) $-$ 0.516 x ln (1+$\delta$18O/1000), relative to Vienna Standard Mean Ocean (VSMOW).

Comment 6: Sections 3.2-3.3: I'm a little unsure as to why the authors are spending so much time on the _18O + _17O "source effect". Their collections were from polluted air masses not direct stack emissions, such that _18O and _17O should effectively be wiped of any "source effect" due to the rapid equilibration of NOx and its oxidants. The authors suggest in 3.2, that due to elevated _18O + _17O there isn't a source effect (this is not surprising or a novel finding but expected), but extend the discussion of source effects in 3.3 despite ruling them out in 3.2. This seems a bit odd to me, and I think it would serve this manuscript better to simplify these sections into 1, removing source effect discussions. Instead the authors should focus on the unique oxidation chemistries of the polluted air masses to understand how differences in NOx oxidation cycling and post NO2 reactions would have altered _18O + _17O rather than a source derived _18O + _17O effect, but again this is complicated as previously mentioned because of the nitrate lifetime problem. We don't know that the nitrate sampled is from the targeted source area.

REP – The final version of the article will show former Figure 1 in supplementary information (replaced in article by a location map), remove the emphasis previously put on the individual sources, and merge together sections 3.2 and former section 3.3 (for which the length is now much reduced).

Comment 7 (MAJOR): (a) Section 4.1: The Alberta nitrate _18O and _17O relationships appear linear despite the author's claim to the contrary. Can the authors include regression statistics so that their argument is supported quantitively rather than qualitatively?

REP – If the reviewer refers to Figure 2, data per seasons are not numerous enough to play with statistics. However this is a very minor point and the sentence can be erased without consequences.

(b) Much of the authors _17O range calculations and justifications are ad hoc. Can the authors properly justify their assumptions made in this calculation, specifically "50% contribution from each pathway for summer"?

REP – This assumption is just meant to help present alternatives. We do not claim that 50% is the proportion applicable for the collected samples.

(c) Additionally, can the authors propagate the error made in the suggested _17O ranges? The authors indicate that the _17O range "shrinks" during winter but their calculations indicate a larger range during the winter (winter: 26 to 37‰ summer: 20 to 29‰

REP – The "shrinks" referred to the wider range estimated from the extreme cases discussed earlier in the paragraph (100% from the OH pathway or 100% for the N2O5 pathway; 25-45 ‰ for winter samples). This can be clarified.

(d) The authors conclusion that _17O of NO2 is not equal to the asymmetrical O3 is not new but rather expected, due to VOC oxidation contributions that have an NO oxidation branching ratio of 70 to 80%. Perhaps the authors should retry their calculations

utilizing a more realistic approximated _17O of NO2. Also, could the authors compare their _17O in this region with the global _17O model?

REP – The goal of the calculation was to constrain the source NO2 Δ17O values based on those of nitrate, and it is not clear what the reviewer is suggesting. Use an approximate value of NO2 Δ17O to calculate the contributions from the different NO2 oxidation pathways in different samples? That would require assuming the NO2 Δ17O value was constant, which would also be unrealistic. The contribution of RO2 oxidation of NO is discussed in the paragraph following the calculation. The nitrate Δ17O are compared with the global model in section 4.4; NO2 Δ17O was not explicitly mapped in Alexander et al. (2009) for comparison.

(e) Again, the calculated transit times of 9 minutes to 4 hours, indicates that not all of the sampled nitrate is derived from the targeted upwind region due to the chemical lifetimes of NOx and atmospheric lifetime of nitrate. I find it hard to believe any of the interpretation on _18O and _17O differences between p-NO3- and HNO3(g) because this speciation was likely not truly achieved given the method concerns already raised in this review and others. I recommend that this speciation discussion should be removed and _18O and _17O interruption should focus on total nitrate relative to wet-nitrate (which was hardly discussed in this manuscript!)

REP – See reply to comment 4.

Comment 8: In general, I find the figures and tables difficult to read and interrupt (especially Figs. 3 and 4)

REP - We are planning to remove Figure 4 (see reply I to reviewer II) and willing to make the necessary changes to Figure 3 so it becomes easy to read.

Comment 9: The authors findings that meteorological parameters often correlate with HNO3, p-NO3- and their isotopic compositions isn't surprising (particularly phase separation) due to the well-established thermodynamic equilibrium of HNO3 and p-NO3-

that determines this phase separation. This point however, directly contradicts that authors claim that reaction pathways (i.e. NO2 + OH vs N2O5 heterogenous rxn) had a significant role on the observed speciation and isotopic composition in Section 4.1.

REP – Thermodynamic equilibrium will contribute to change the $\delta 18O$ values, but not the $\Delta 17O$ signals.

Comment 10: Overall, I'm not convinced that during the summer, source effects lowered the anthropogenic originating nitrate _17O values. The simplest explanation for this observation should be NO oxidation contributions from RO2. Until the authors can explicitly rule out the "oxidation chemistry effect" by modeling or empirical evidence, I don't think the authors suggested conclusion should be drawn.

REP – The RO2 oxidation contribution is not ruled out, but we believe it is valuable to raise an alternative hypothesis in order to spur further testing
* * *

---

## Referee Comment (RC4) · Anonymous Referee #2 · 10 Apr 2018

The manuscript by Savard and colleagues presents interesting data worthy of publication. However, I found the results and discussion sections muddled and the important points worthy of highlighting buried. The manuscript could be improved by a focus on key findings and condensing or eliminating repetitive sections. For example, page 10 lines 24-35 are potential explanations for low observed d18O, D17O values with important implications multiple communities. Additionally, challenging the assumption of NOx isotopic steady state with O3 is a key takeaway (not mentioned until page 14). Why are these not highlighted more prominently in the abstract? The current conclusions in the Abstract and Conclusion by comparison are weak "isotopic signals of.... are not interchangeable", and "invariably interchangable".

[Figure]

The manuscript needs a map to put the respective sampling sites and the surround potential sources in a spatial context. Without this information, it is not possible to discern how far sites are from each other.

The authors points out "very few" air masses passed over other sources outside the targeted ones in the preceding 24 hours. For those that did, are they removed from the analysis? Why is this data not shown- as it seems relevant.

The long variation in sampling times is concerning. For example, individual sample deployments ranged from 5 to 113 days. The authors should explore whether there is any evidence that the length of this sampling time caused any artifacts in their results.

Page 5, line 10: What is "preconized"?

Page 5: line 33: Why was MAD scaled by 0.6745? Where did this number come from? Needs justification.

The authors conclude Elliott et al found minimal fractionation between d18O of pNO3- and HNO3. Figure 2 from that paper shows $\sim$10 permil differences during summer in the d18O values of these two components.

Page 1, line 19: Add distance to state how far collection sites are from major sources (i.e., from x to y km).

Figures 2a-c. Include 1:1 line to clarify your conclusion that data show a "vertical extent". More clarification needed here in this analysis/conclusion.

Discussion: Lines 18-35. This reads like intro text/results. Revise to lead off with a topic sentence that highlights your major finds and built supporting text around this.

Authors state that "Anthropogenic emitters involving combustion (O2) may generate primary NOx at or near sources that tend to carry low d18O and D17O values". It is not clear whether this is in reference to prior published studies, or whether this is one of their conclusions. Either way, it needs more justification.

For the analysis on page 10 lines 7-22, it is not clear how the authors determined the relative proportion of R1, R2, R3 to calculate the influence of O3 on oxidation pathways from NO2 to HNO3 (R4, R7, R8).

I found the discussion on page 11 lines 13-25 very intriguing. How might seasonal differences in lifetimes affect how far different constituents travel? Is there any prior modeling work (e.g., GEOS-CHEM) that could support these ideas?

The authors refer to "seasonal changes in planetary boundary layer heights" but don't explicitly state what these changes are and how they could impact their results.

Section 4.2 This reads as Results rather than Discussion.

Conclusions: What is "invariably interchangeable"?

---

## Referee Report (RR1)

The authors have addressed my significant concerns with this manuscript.  In particular, it has been streamlined, sections reorganized, new spatial context added, and key points are now sufficiently highlighted.  While other reviewers have substantial concerns regarding interpretation, properly addressing those concerns is part of the conversation in the larger, scientific literature.  This paper will help propel that conversation forward.

---

## Author Response (AR3)

[revised manuscript text omitted]

Axis labels must be δ/‰ or 10^3 δ (ln(1+δ)/‰ to be numerically correct, e.g. δ18O(VSMOW)/‰ or 10^3
5   δ18O(VSMOW). Please also see the examples in the BIPM brochure and IUPAC Green Book. The unit must not
be included in the label in parentheses because it is not an argument of the quantity symbol.

Tables 2 to 3: Column headings need to be written as δ18O/‰ or 10^3 δ18O etc. to be numerically correct.

10  REPLY: All corrected, including Table 3.

Table 6: Replace "isotopic results (‰)" with "isotope delta" (without ‰ symbol).
REPLY: DONE
Also, it is not clear what "r" stands for - the slope of a linear regression fit? If so, it must have units, e.g. ‰/%
15  in the case of regressing δ18O vs. relative humidity, or ‰/°C in case of regressing δ18O vs. temperature,
"‰/(nmol mol-1)" in the case of δ18O vs. O3 mole fraction, etc.

REPLY: 'r' stands for correlation coefficient; now indicated in the caption.

State the quantities referred to by the symbols "PM", "SO2" and "O3", i.e. PM mass concentration, SO2 mole
fraction and O3 mole fraction.

25  REPLY: We believe that indicating the units of the parameters that were statistically analyzed could be
misleading, so we elected not to add this information (never shown in statistical papers).

6/4: What do you mean by 2 ppb? This may relate to the mass fraction of nitrate (M = 62 g mol-1) in solution,
but this is confusing, since this quantity is not used anywhere else in the paper. It should therefore best be
30  deleted.

REPLY: DONE

6/5: This value should be rounded appropriately, e.g. 1.1 μmol/L
35

REPLY: DONE

8/4-6: Please convert to μmol/L for consistency with page 6.

REPLY: DONE

30/5: The definition of D(17O) in Miller (2002) is mathematically wrong. As an example, a $\delta$ value of 1 ‰ is numerically identical to the number 0.001. It must not be divided by 1000 because this would alter it to a value of 0.000001.

Please change this equation to $\ln(1+\delta17O) - 0.516 \ln(1+\delta18O)$.

REPLY: DONE

Table SM-3: Please replace "ppb" with the SI unit "nmol mol-1"
REPLY: DONE

---

## Author Response (AR4)

[revised manuscript text omitted]

**Minor corrections suggested by Dr. Jan Kaiser (Editor)**

1) Please adhere to the International System of Units (SI), which has not always been applied in the present version of the manuscript. Please nsult the SI brochure published by BIPM (http://www.bipm.org/en/publications/si-brochure/) and chapter 1 of the IUPAC Green Book (http://media.iupac.org/publications/books/gbook/IUPAC-GB3-2ndPrinting-Online-22apr2011.pdf)

Specifically, chemical symbols and units should not be mixed, i.e. it is not permissible to write "10 mg N m-3". To correct this, the chemical species must be identified by the quantity symbol (not the unit), e.g. "c(NO3--N) = 10 mg m-3", or, in the text, "10 mg m-3 of NO3--N", or "10 mg m-3 as N equivalents", or "a nitrate-N concentration of 10 mg m-3"

REPLY: All corrected.
This also applies to the tables in the supplementary material.
REPLY: All corrected.

2) Table 3, Figs. 2, 3, 4, 6 & Figs. SM-3 & 5: Axis labels should be $\delta$/‰ or $10^3 \delta$ ($\ln(1+\delta)$/‰ in case of Fig. 4). See BIPM brochure and IUPAC Green Book for correct examples.

REPLY: The ‰ was previously indicated in the captions of all figures. We now have removed it from the captions and placed it on each axis. All corrected, including Figure 5.

1/13: Replace "ratios" with "delta values".
REPLY: DONE

3/10: Replace "/filter" with "per filter".
REPLY: CORRECTED on 4/25

6/4: The unit "ppb N-NO3/L" does not make sense. Also, please see comment 1 above and remove "N-NO3" here and from "0.03 N-NO3 umol/L".
REPLY: DONE on 4/6

6/5: Please convert 0.016 µg to µmol and remove "N".
REPLY: DONE

6/25: Replace "ratios" with "values."
REPLY: DONE

6/32: Delete extraneous "/1000" divisors.

REPLY: We have corrected the equation as in Miller (2002), by adding 1000x to the two parts: $1000 \times \ln(1+\delta^{17}O/1000) - 0.516 \times 1000 \times \ln(1+\delta^{18}O/1000)$.

7/20: Replace "ratios" with "delta values" or "deltas".

REPLY: DONE

7/21 & 7/23 & 10/6 "Isotopic difference" is meaningless as a quantitative term on its own. Please rephrase so as that it is clear which delta values are meant. Also, it would be clearer to say explicitly whether particulate nitrate or HNO3 have higher or lower delta values.

REPLY: DONE

8/13: Change to "not the highest delta values for w-NO3-".

REPLY: DONE

8/22: Subscript "x" in "NOx". "x" should be in italics.

REPLY: DONE

8/28: "NO3-".

REPLY: DONE

9/4: Add space between 0 and ºC. .

REPLY: DONE

9/19 & R12 Table 5: "NO3" (neutral radical) .

REPLY: DONE

12/22: The correct unit symbol for "hour" is "h"..

REPLY: DONE

12/25: Please replace "ppb" and "ppt" with the SI units "nmol mol-1" and "pmol mol-1"..

REPLY: DONE

14/2: "The negative isotopic differences between p-NO3- and HNO3 ..." is unclear. Please rephrase so that it is clear which species has lower delta values.

[revised manuscript text omitted]

**Minor corrections suggested by Dr. Jan Kaiser (Editor)**

1) Please adhere to the International System of Units (SI), which has not always been applied in the present version of the manuscript. Please nsult the SI brochure published by BIPM

5  (http://www.bipm.org/en/publications/si-brochure/) and chapter 1 of the IUPAC Green Book (http://media.iupac.org/publications/books/gbook/IUPAC-GB3-2ndPrinting-Online-22apr2011.pdf)

Specifically, chemical symbols and units should not be mixed, i.e. it is not permissible to write "10 mg N m-3". To correct this, the chemical species must be identified by the quantity symbol (not the unit), e.g. "c(NO3--N)

10 = 10 mg m-3", or, in the text, "10 mg m-3 of NO3--N", or "10 mg m-3 as N equivalents", or "a nitrate-N concentration of 10 mg m-3"

REPLY: All corrected.
This also applies to the tables in the supplementary material.

15 REPLY: All corrected.

2) Table 3, Figs. 2, 3, 4, 6 & Figs. SM-3 & 5: Axis labels should be δ/‰ or $10^3$ δ (ln(1+δ)/‰ in case of Fig. 4). See BIPM brochure and IUPAC Green Book for correct examples.

REPLY: The ‰ was previously indicated in the captions of all figures. We now have removed it from the captions

20 and placed it on each axis. All corrected, including Figure 5.

1/13: Replace "ratios" with "delta values".
REPLY: DONE

3/10: Replace "/filter" with "per filter".
REPLY: CORRECTED on 4/25

6/4: The unit "ppb N-NO3/L" does not make sense. Also, please see comment 1 above and remove "N-NO3"

30 here and from "0.03 N-NO3 umol/L".
REPLY: DONE on 4/6

6/5: Please convert 0.016 µg to µmol and remove "N".
REPLY: DONE

6/25: Replace "ratios" with "values."
REPLY: DONE

6/32: Delete extraneous "/1000" divisors.

REPLY: We have corrected the equation as in Miller (2002), by adding 1000x to the two parts: 1000 x ln $(1+\delta^{17}O/1000) - 0.516 \times 1000 \times \ln (1+\delta^{18}O/1000)$.

7/20: Replace "ratios" with "delta values" or "deltas".
REPLY: DONE
7/21 & 7/23 & 10/6 "Isotopic difference" is meaningless as a quantitative term on its own. Please rephrase so as that it is clear which delta values are meant. Also, it would be clearer to say explicitly whether particulate nitrate or HNO3 have higher or lower delta values.
REPLY: DONE
8/13: Change to "not the highest delta values for w-NO3-".
REPLY: DONE
8/22: Subscript "x" in "NOx". "x" should be in italics.
REPLY: DONE
8/28: "NO3-".
REPLY: DONE
9/4: Add space between 0 and ºC. .
REPLY: DONE
9/19 & R12 Table 5: "NO3" (neutral radical) .
REPLY: DONE
12/22: The correct unit symbol for "hour" is "h"..
REPLY: DONE
12/25: Please replace "ppb" and "ppt" with the SI units "nmol mol-1" and "pmol mol-1"..
REPLY: DONE
14/2: "The negative isotopic differences between p-NO3- and HNO3 ..." is unclear. Please rephrase so that it is clear which species has lower delta values.

[revised manuscript text omitted]

*Q is a stable molecule of high energy; M is either O₂ or N₂; RO₂ stands for both HO₂ and alkyl peroxy. HC stand for hydrocarbons. *This aqueous nitrate may be on a particle.*

**Table 6. Correlations of NO$_3^-$ isotopic results (‰) with meteorological parameters and concentration (or ratio) of co-contaminants.**

| | Relative Humidity | | Temperature | | Daylight (fraction) | | PM | SO$_2$ | O$_3$ | |
|---|---|---|---|---|---|---|---|---|---|---|
| | r | R$^2$ | r | R$^2$ | r | R$^2$ | r | r | r | R$^2$ |
| HNO$_3$ | | | | | | | | | | |
| δ$^{18}$O | **0.8** | 0.59 | -0.4 | | -0.3 | | 0.1 | 0.0 | -0.29 | |
| n | 8 | | 15 | | 15 | | 13 | 13 | 13 | |
| Δ$^{17}$O | 0.6 | | **-0.5** | 0.24 | -0.4 | | 0.4 | 0.3 | -0.03 | |
| n | 8 | | 15 | | 15 | | 13 | 13 | 13 | |
| p-NO$_3^-$ | | | | | | | | | | |
| δ$^{18}$O | **0.9** | 0.79 | **-0.6** | 0.34 | **-0.6** | 0.35 | 0.1 | 0.5 | **-0.61** | 0.38 |
| n | 7 | | 15 | | 15 | | 12 | 12 | 12 | |
| Δ$^{17}$O | **0.9** | 0.73 | **-0.6** | 0.34 | **-0.7** | 0.44 | 0.0 | 0.5 | -0.47 | |
| n | 7 | | 15 | | 15 | | 12 | 12 | 12 | |

In **bold** are the significant correlation coefficients, equal or above the 95 % significance value

**A**

[Figure]

**B**

[Figure]

Figure 1. Aerial images showing sampling sites (green triangles) in central and southern Alberta (A), and in the greater Edmonton area (B), along with emissions of NOx as tonnes of $NO_2$ reported to the National Pollutant Release Inventory for 2013 (Environment and Climate Change Canada, 2018b).

[Figure]

5    **Figure 2: Triple O isotopic results obtained for simultaneously collected atmospheric $HNO_3$ (A), w-$NO_3^-$ (B) and p-$NO_3^-$ (C), in Alberta, identified by sampling periods (cold months - blue; warm months - red).**

[Figure]

[Figure]

**Figure 3: Line-connected $\delta^{18}$O and $\Delta^{17}$O values for simultaneously collected HNO$_3$ (empty symbols) and p-NO$_3^-$ (solid symbols) from cold (blue) and warm (red) sampling periods.**

[Figure]

**Figure 4: Isotopic results for p-NO$_3^-$ identified by sampling periods (solid lines), compared with summer and winter trends obtained for Arctic sites (dashed lines; derived from ln (1+ δ) in Morin et al., 2008).**

[Figure]

[Figure]

**Figure 5: Weighted** Δ¹⁷O **average for the sum of dry nitrates as a function of** NO₂ **concentration divided by p-NO₃ plus** HNO₃ **concentrations, a ratio indicative of the maturity of a plume.**

[Figure]

**Figure 6: Isotopic ratios for atmospheric p-NO₃⁻, w-NO₃⁻ and HNO₃ samples in cold and warm periods from central and southern Alberta (this study), compared with previously published winter and summer bulk and throughfall deposition samples from the oil sands (OS) region from northern Alberta (Proemse et al., 2013), and p-NO₃⁻ in-stack emissions data for an OS upgrader located in the same region (Proemse et al., 2012). The grey dotted line connects NOₓ from theoretical combustion with O₂ isotopic composition and at isotopic equilibrium with tropospheric O₃ (Michalski et al., 2014).**

**Minor corrections suggested by Dr. Jan Kaiser (Editor)**

1) Please adhere to the International System of Units (SI), which has not always been applied in the present version of the manuscript. Please nsult the SI brochure published by BIPM
5 (http://www.bipm.org/en/publications/si-brochure/) and chapter 1 of the IUPAC Green Book (http://media.iupac.org/publications/books/gbook/IUPAC-GB3-2ndPrinting-Online-22apr2011.pdf)

Specifically, chemical symbols and units should not be mixed, i.e. it is not permissible to write "10 mg N m-3". To correct this, the chemical species must be identified by the quantity symbol (not the unit), e.g. "c(NO3--N)
10 = 10 mg m-3", or, in the text, "10 mg m-3 of NO3--N", or "10 mg m-3 as N equivalents", or "a nitrate-N concentration of 10 mg m-3"

REPLY: All corrected.
This also applies to the tables in the supplementary material.
15 REPLY: All corrected.

2) Table 3, Figs. 2, 3, 4, 6 & Figs. SM-3 & 5: Axis labels should be $\delta$/‰ or $10^3 \delta$ (ln(1+$\delta$)/‰ in case of Fig. 4). See BIPM brochure and IUPAC Green Book for correct examples.

REPLY: The ‰ was previously indicated in the captions of all figures. We now have removed it from the captions
20 and placed it on each axis. All corrected, including Figure 5.

1/13: Replace "ratios" with "delta values".
REPLY: DONE

3/10: Replace "/filter" with "per filter".
REPLY: CORRECTED on 4/25

6/4: The unit "ppb N-NO3/L" does not make sense. Also, please see comment 1 above and remove "N-NO3"
30 here and from "0.03 N-NO3 umol/L".
REPLY: DONE on 4/6

6/5: Please convert 0.016 µg to µmol and remove "N".
REPLY: DONE

6/25: Replace "ratios" with "values."
REPLY: DONE

6/32: Delete extraneous "/1000" divisors.

REPLY: We have corrected the equation as in Miller (2002), by adding 1000x to the two parts: 1000 x ln $(1+\delta^{17}O/1000) - 0.516$ x 1000 x ln $(1+\delta^{18}O/1000)$.

7/20: Replace "ratios" with "delta values" or "deltas".

REPLY: DONE

7/21 & 7/23 & 10/6 "Isotopic difference" is meaningless as a quantitative term on its own. Please rephrase so as that it is clear which delta values are meant. Also, it would be clearer to say explicitly whether particulate nitrate or HNO3 have higher or lower delta values.

REPLY: DONE

8/13: Change to "not the highest delta values for w-NO3-".

REPLY: DONE

8/22: Subscript "x" in "NOx". "x" should be in italics.

REPLY: DONE

8/28: "NO3-".

REPLY: DONE

9/4: Add space between 0 and ºC. .

REPLY: DONE

9/19 & R12 Table 5: "NO3" (neutral radical) .

REPLY: DONE

12/22: The correct unit symbol for "hour" is "h"..

REPLY: DONE

12/25: Please replace "ppb" and "ppt" with the SI units "nmol mol-1" and "pmol mol-1"..

REPLY: DONE

14/2: "The negative isotopic differences between p-NO3- and HNO3 ..." is unclear. Please rephrase so that it is clear which species has lower delta values.

[revised manuscript text omitted]

Q is a stable molecule of high energy; M is either *O$_2$* or *N$_2$*; *RO$_2$* stands for both *HO$_2$* and alkyl peroxy. HC stand for hydrocarbons. *This aqueous nitrate may be on a particle.*

**Table 6. Correlations of $NO_3^-$ isotopic results (‰) with meteorological parameters and concentration (or ratio) of co-contaminants.**

| | Relative Humidity | | Temperature | | Daylight (fraction) | | PM | SO$_2$ | O$_3$ | |
|---|---|---|---|---|---|---|---|---|---|---|
| | r | R$^2$ | r | R$^2$ | r | R$^2$ | r | r | r | R$^2$ |
| HNO$_3$ | | | | | | | | | | |
| $\delta^{18}O$ | **0.8** | 0.59 | -0.4 | | -0.3 | | 0.1 | 0.0 | -0.29 | |
| n | 8 | | 15 | | 15 | | 13 | 13 | 13 | |
| $\Delta^{17}O$ | 0.6 | | **-0.5** | 0.24 | -0.4 | | 0.4 | 0.3 | -0.03 | |
| n | 8 | | 15 | | 15 | | 13 | 13 | 13 | |
| p-NO$_3^-$ | | | | | | | | | | |
| $\delta^{18}O$ | **0.9** | 0.79 | **-0.6** | 0.34 | **-0.6** | 0.35 | 0.1 | 0.5 | **-0.61** | 0.38 |
| n | 7 | | 15 | | 15 | | 12 | 12 | 12 | |
| $\Delta^{17}O$ | **0.9** | 0.73 | **-0.6** | 0.34 | **-0.7** | 0.44 | 0.0 | 0.5 | -0.47 | |
| n | 7 | | 15 | | 15 | | 12 | 12 | 12 | |

In **bold** are the significant correlation coefficients, equal or above the 95 % significance value

**A**

[Figure]

**B**

5   **Figure 1**. Aerial images showing sampling sites (green triangles) in central and southern Alberta (A), and in the greater Edmonton area (B), along with emissions of NO$_x$ as tonnes of NO$_2$ reported to the National Pollutant Release Inventory for 2013 (Environment and Climate Change Canada, 2018b).

[Figure]

5 **Figure 2: Triple O isotopic results (‰) obtained for simultaneously collected atmospheric HNO₃ (A), w-NO₃⁻ (B) and p-NO₃⁻ (C), in Alberta, identified by sampling periods (cold months - blue; warm months - red).**

[Figure]

**Figure 3:** Line-connected $\delta^{18}O$ and $\Delta^{17}O$ values (‰) for simultaneously collected HNO$_3$ (empty symbols) and p-NO$_3^-$ (solid symbols) from cold (blue) and warm (red) sampling periods.

[Figure]

[Figure]

**Figure 4: Schematic outline of main steps in the production of Alberta nitrates: NOx-O₂-O₃ photochemical cycle (1) and reaction with RO₂ (2) modify NOx source signals (R2, R3, R9); oxidation of NOx produces HNO₃ along the N₂O₅ (4) or OH (5) pathways. The grey line represents NOx from photochemical cycling with O₂ and O₃ (Michalski et al., 2014). The direction of arrows 1 to 5 indicates how the isotopic values would evolve along the different chemical reactions; the positions of these arrows are arbitrary.¶**

**Figure 4:** Isotopic results (‰) for p-NO$_3^-$ identified by sampling periods (solid lines), compared with summer and winter trends obtained for Arctic sites (dashed lines; derived from ln (1+ δ) in Morin et al., 2008).

[Figure]

**Figure 5: Weighted average Δ¹⁷O (‰) for the sum of dry nitrates as a function of NO₂ concentration divided by p-NO₃ plus HNO₃ concentrations, a ratio indicative of the maturity of a plume.**

[Figure]

**Figure 6: Isotopic ratios (‰) for atmospheric p-NO₃⁻, w-NO₃⁻ and HNO₃ samples in cold and warm periods from central and southern Alberta (this study), compared with previously published winter and summer bulk and throughfall deposition samples from the oil sands (OS) region from northern Alberta (Proemse et al., 2013), and p-NO₃⁻ in-stack emissions data for an OS upgrader located in the same region (Proemse et al., 2012). The grey dotted line connects NOₓ from theoretical combustion with O₂ isotopic composition and at isotopic equilibrium with tropospheric O₃ (Michalski et al., 2014).**

[Figure]

**Figure 6: Isotopic results (‰) for p-NO₃⁻ identified by sampling periods (solid lines), compared with summer and winter trends obtained for Arctic sites (dashed lines; derived from ln (1+ δ) in Morin et al., 2008).**

| Page 1 : [1] Supprimé | Cole,Amanda [Ontario] | 08/05/2018 3:29:00 PM |
|---|---|---|

r

| Page 1 : [1] Supprimé | Cole,Amanda [Ontario] | 08/05/2018 3:29:00 PM |
|---|---|---|

r

| Page 1 : [1] Supprimé | Cole,Amanda [Ontario] | 08/05/2018 3:29:00 PM |
|---|---|---|

r

| Page 1 : [1] Supprimé | Cole,Amanda [Ontario] | 08/05/2018 3:29:00 PM |
|---|---|---|

r

| Page 1 : [1] Supprimé | Cole,Amanda [Ontario] | 08/05/2018 3:29:00 PM |
|---|---|---|

r

| Page 1 : [1] Supprimé | Cole,Amanda [Ontario] | 08/05/2018 3:29:00 PM |
|---|---|---|

r

| Page 1 : [1] Supprimé | Cole,Amanda [Ontario] | 08/05/2018 3:29:00 PM |
|---|---|---|

r

| Page 1 : [1] Supprimé | Cole,Amanda [Ontario] | 08/05/2018 3:29:00 PM |
|---|---|---|

r

| Page 1 : [1] Supprimé | Cole,Amanda [Ontario] | 08/05/2018 3:29:00 PM |
|---|---|---|

r

| Page 1 : [2] Supprimé | Savard, Martine | 23/03/2018 2:23:00 PM |
|---|---|---|

polluted

| Page 1 : [2] Supprimé | Savard, Martine | 23/03/2018 2:23:00 PM |
|---|---|---|

polluted

| Page 1 : [2] Supprimé | Savard, Martine | 23/03/2018 2:23:00 PM |
|---|---|---|

polluted

| Page 1 : [3] Supprimé | Cole,Amanda [Ontario] | 07/05/2018 4:08:00 PM |
|---|---|---|

pathways

| Page 1 : [3] Supprimé | Cole,Amanda [Ontario] | 07/05/2018 4:08:00 PM |
|---|---|---|

pathways

| Page 1 : [3] Supprimé | Cole,Amanda [Ontario] | 07/05/2018 4:08:00 PM |
|---|---|---|

pathways

| Page 1 : [3] Supprimé | Cole,Amanda [Ontario] | 07/05/2018 4:08:00 PM |
|---|---|---|

pathways

| Page 1 : [4] Supprimé | Savard, Martine | 24/01/2018 1:11:00 PM |

five different

| Page 1 : [5] Supprimé | Cole,Amanda [Ontario] | 07/05/2018 4:08:00 PM |

the

| Page 1 : [6] Supprimé | Savard, Martine | 25/04/2018 4:31:00 PM |

The summer pattern

| Page 1 : [7] Supprimé | Cole,Amanda [Ontario] | 07/05/2018 4:09:00 PM |

valid

| Page 1 : [8] Supprimé | Savard, Martine | 25/04/2018 10:02:00 AM |

while the l

| Page 1 : [8] Supprimé | Savard, Martine | 25/04/2018 10:02:00 AM |

while the l

| Page 1 : [8] Supprimé | Savard, Martine | 25/04/2018 10:02:00 AM |

while the l

| Page 1 : [8] Supprimé | Savard, Martine | 25/04/2018 10:02:00 AM |

while the l

| Page 1 : [8] Supprimé | Savard, Martine | 25/04/2018 10:02:00 AM |

while the l

| Page 6 : [9] Supprimé | Savard, Martine | 24/05/2018 9:49:00 AM |

Concentration of nitrates on Teflon and Nylon filter extracts, [AC1]and in precipitation samples were determined at Institut national de la recherche scientifique–Eau, Terre, Environnement (INRS-ETE). The determinations used an automated QuikChem 8000 FIA+ analyzer (Lachat Instruments), equipped with an ASX-260 series autosampler. The detection limit for the method (# 31-107-04-1-A with sulfanilamide) was 2 ppb $N-NO_3$/L (0.03 $N-NO_3$ umol/L). The concentration for NO and $NO_2$ collected with samplers of Maxxam Analytics were determined by Maxxam Analytics using an ion chromatograph.

[revised manuscript text omitted]

Another argument supporting this interpretation is the strong correlation between $\Delta^{17}O$ values and the maturity of a plume as expressed by the $NO_2$ content divided by sum of dry nitrates (Fig. 6). The results reflect the higher content of $O_3$-derived O in dry nitrates from mature plumes, i.e., with relatively low $NO_2$ contents. Seasonal changes in the planetary boundary layer height may also affect the impingement of emission plumes on the measurement sites, and thereby the relative amounts of fresh vs background nitrates.

**Point-by-point REPLY to Anonymous Referee #1**

Comment 1: Title: I find the title to be somewhat misleading specifically "from anthropogenic sources – Implications for polluted air masses" As written this title implies direct plume emissions were sampled rather than air mass originating from anthropogenic emission regions. Due to the numerous sampling sites, I think the authors should instead highlight that this work is spatially-resolved.

REPLY – We have modified the title.

Change in manuscript: "The $\Delta^{17}O$ and $\delta^{18}O$ values of atmospheric nitrates simultaneously collected downwind from anthropogenic sources – Implications for polluted air masses"

Comment 2: Intro: I think the introduction could use a better framework. If speciating HNO3(g) and p-NO3- is a main motivation of this study (however likely not achieved), then I think the introduction needs to have a better critical analysis of HNO3(g) and p-NO3- dynamics in the atmosphere. What might _18O and _17O of speciated nitrate tell the atmospheric community that isn't already known? Will this speciated work shed light into their separate formation pathways or will the phase separation reflect the inorganic thermodynamics of trace gases and PM driven by RH, temperature, [sulfate], [ammonia], etc.

REPLY – We have modified the introduction to better illustrate the motivation for the research.

Change in manuscript: « In the past, due to sampling challenges, $HNO_3$ and p-$NO_3^-$ were generally collected together (without differentiation). Therefore, no studies have separately and simultaneously collected and analyzed the $HNO_3$ and p-$NO_3^-$ $\delta^{18}O$ and $\Delta^{17}O$ values, and discussed these isotopic characteristics of nitrate collected downwind of anthropogenic emitters. While $HNO_3$ and p-$NO_3^-$ can be in equilibrium (e.g. if p-$NO_3^-$ is in the form of solid $NH_4NO_3$), this is not always the case, for example, if nitrate is bonded to calcium or dissolved in liquid water on a wet particle (see section 3.3). They also have different lifetimes with respect to wet scavenging and dry deposition, and may differ in their formation pathways as well. Therefore, investigating the mass independent and dependent oxygen fractionations in nitrates separately collected may help identifying their respective formation and loss pathways, and provide additional constraints on processes controlling their distribution.»

Comment 3: Sampling Methodology: I think it would be useful to quantitatively demonstrate that the collected nitrate truly originated from the targeted upwind region. Specifically, the lifetimes both for NOx oxidation to nitrate and nitrate lifetime should be considered. Based on these lifetimes, how much of the sampled nitrate reasonable originated from NOx oxidized from the upwind region? Was the emitted NOx converted to nitrate from these emission regions are is the sampled nitrate a mixture of "background" nitrate originated from NOx emitted further upwind that the targeted region due to a relatively long nitrate lifetime (3-5 days). I think this is incredible important especially considered the sampling distance from the targeted sources (1 to 125 km).

REPLY – We aimed at isolating emissions from sources at their respective location, with the emissions subject to some atmospheric processing (i.e., not stack sampling). By necessity, it is clear that background contributed to the sampled load, but not in significant proportions. A few pieces of evidence support this. One is the low background concentrations; background particle nitrate and nitric acid concentrations at Wood Buffalo National Park in northern Alberta, where CAPMoN began sampling in 2014, averaged 0.016 and 0.020 µg N m$^{-3}$, respectively, for >2 years of monitoring. Concentrations from the conditional sampling at the sampling sites studied here were up to 20 times higher, suggesting that the collected samples have greater p-NO3 and HNO$_3$ concentrations than background. In addition, we obtained some model output from a 2-week simulation in 2013 (Makar et al., 2018) illustrating the significantly higher concentrations of nitric acid downwind of the coal-fired power plants near Genesee. In this snapshot, the three power plants are shown as brown dots and the three Edmonton-area sampling locations are shown as purple stars. The NO$_2$ plumes are shown as well to illustrate the change in direction of the plume. It is clear that at the Genesee site, the bulk of the nitric acid at this particular time resulted from the nearby power plants. Using the conditional sampling weighted the samples more heavily when this was the case.

[Figure]

[Figure]

Changes in manuscript:

Text added to section 3.2: *"Background sites for this region are sparse, but concentrations at Cree Lake in neighbouring Saskatchewan were the lowest in Canada reported up to 2011 (Cheng and Zhang, 2017), and 2014-2016 measurements at Wood Buffalo National Park on the northern Alberta border revealed similar average concentrations of 0.02 µg N/m³ for both HNO₃ and p-NO₃⁻ (preliminary internal data). Therefore, the lowest concentrations in our samples approached average background concentrations, while the highest were 20 or more times higher than regional background."*

Text added to section 4.1: *"While the fraction of NOₓ converted to nitrate in this transit time may be small, these are large sources of NOₓ in an area with very low background nitrates. For example, a plume containing 10 ppb of NO₂ mixing with background air with 0.1 ppt of OH (Howell et al., 2014) would produce HNO₃ via R6 at a rate of 0.011µg N m⁻³ min⁻¹ at T = 7 °C (Burkholder et al., 2015), or an equivalent amount of a typical sample in 10 minutes. Even if equilibration with O₃ is established within a few minutes, the nitrate produced in the interim can form a substantial fraction of the sample collected nearby."*

Comment 4 (MAJOR): Sampling Protocols: I'm not convinced that nitrate speciation (HNO3 ad p-NO3-) was actually achieved with the sampling filter pack method. The problem, as the authors have pointed out, that the collected p-NO3- can easily volatize as HNO3 that is collected on the filter designated for HNO3(g) collection. Additionally, there is the possibility of gaseous reactions on the PM filter especially as the filter accumulates PM. The authors point out that they don't expect volatilization of p-NO3- to play a major role on their results because p-NO3- isotope ratios are higher during winter than during summer and that the difference between p-NO3- and HNO3 is opposite than expected, but this is not proper justification. First, I wonder what the expected p-NO3- and HNO3 isotope difference is? This process is likely driven by an equilibrium effect rather than a kinetic effect since the volatilization of p-NO3- as HNO3 is due to the system being at non-equilibrium. I think this would change the authors expectation that the difference p-NO3- and HNO3 is driven by a kinetic effect (I assumed this was the authors assumption). Additionally, which "isotopic ratios" did the authors use to evaluate the p-NO3- and HNO3 difference? I'm assuming mass-dependent _18O but this information is not provided in text. Also, suggesting that this filter pack method has previously been used for

isotopic analysis of p-NO3- and HNO3 (Elliot et al., 2009) does not mean the sampling method is correct and optimal for this isotopic analysis.

Elliot et al., 2009 did not quantify this method for phase separation for isotopic analysis; thus, this argument should not be used to justify the work in this manuscript. Also, there is a general lack of information regarding the authors sampling protocols. What types of filters were used (and size)? How were these filters prepared and processed? Were field and laboratory blanks taken? At what flow rates was sampling conducted? How was the requirement that sample air volumes be within 15% quantitatively determined and could the authors elaborated on the experienced flow problems, specifically was this related to mechanical malfunctions or filter build-up? If samplers were placed out for an extended period of time, how might potential for passive HNO3(g) absorption on the Nylon filter play a role in their results?

REPLY – A discussion of field tests and intercomparisons for the filter pack system has been added to section 2.2. Additional clarification of the isotopic evidence that volatilization was not the primary driver of the observed isotopic differences (i.e. that we assume that would be an equilibrium effect, and therefore the fractionation is not what would be expected) has also been provided. Finally, since much of the discussion hinges on $\Delta^{17}O$ values, this mass-dependent process would not create the observed $HNO_3$-$pNO_3$ differences.

Change in manuscript: Added to section 2.2: *"Ambient air was pulled through a three-stage filter pack system to collect, sequentially, particulate matter on a Teflon filter, gaseous nitric acid ($HNO_3$) on a Nylasorb nylon filter, and gaseous ammonia on a citric acid-coated Whatman 41 filter (all 47 mm). The Teflon-nylon filter method for p-$NO_3^-$ and $HNO_3$ has been extensively compared and evaluated, and is currently used by national monitoring networks targeting regional background sites, CAPMoN in Canada and CASTNet (Clean Air Status and Trends Network) in the United States. Previous testing showed negligible collection of $HNO_3$ on the Teflon filter, <3% breakthrough of $HNO_3$ from the nylon filter with loadings more than three times higher than reported here, and blanks for p-$NO_3^-$ and $HNO_3$ of approximately 0.2 μg N/filter (Anlauf et al., 1985; Anlauf et al., 1986). Intercomparisons with more labor-intensive methods, such as tunable diode laser absorption spectroscopy and annular denuder-filter pack systems, have shown evidence of some volatilization of ammonium nitrate from the Teflon filter leading to a negative bias in p-$NO_3^-$ and positive bias in $HNO_3$ under hot (> 25 °C) and dry conditions, particularly in high ambient concentrations (e.g., Appel et al., 1981). However, other field studies have shown no significant differences in $HNO_3$ between filter packs and denuder and/or TDLAS systems (Anlauf et al., 1986; Sickles Ii et al., 1990) or mixed results (Spicer et al., 1982; Zhang et al., 2009). While those studies used short-duration sampling, a comparison for weekly samples at a lower-concentration site showed good agreement between filter pack and denuder values for most of the study but potential interference from $HNO_2$ (nitrous acid) on the nylon filter in two samples(Sickles Ii et al., 1999). Based on the conditions in Alberta, we estimate that there is little or no volatilization of $NH_4NO_3$ for samples with mean temperatures below 5 °C, but there is a possibility for nitrate loss of up to 30% in the warmest sampling periods. "*

More detail about poor flows have also been added to 2.2: *"Flow issues were primarily due to pump failure, likely due to cycling the pumps on and off frequently in early samples. Therefore, for later samples the protocol was changed such that the pumps remained on and valves were*

*used to switch the pumps between sampling lines and non-sampling tubing based on the wind sector."*

The last paragraph of 3.1 has been edited as follows: *"…This could result in equilibrium isotopic fractionation between the particle and gaseous components, which would become artificially high and low, respectively, with more fractionation at higher temperatures (summer) relative to lower temperatures (winter) when volatilization is minimal (Keck and Wittmaack, 2005). We find the p-NO$_3^-$ isotopic ratios ($\delta^{17}O$ and $\delta^{18}O$ ) to be generally higher during winter than during summer (see Section 3.4). Moreover, the p-NO$_3^-$ minus HNO$_3$ isotopic differences are negative during summer, opposite to the expected isotopic artefact if particulate volatilization were the dominant factor in determining the particle-gas isotopic differences (the same was concluded for the $\delta^{15}N$ values in NH$_3$ and NH$_4$; Savard et al., 2017). We therefore conclude that, while volatilization may occur in the summer samples, other isotope effects must be larger in order to lead to the observed differences. In addition, volatilization would cause mass-dependent fractionation and would not affect the $^{17}O$ anomaly; therefore, $\Delta^{17}O$ values remain robust tracers in this situation…"*

Comment 5: Analytical procedures: I also mind a general lack of information in the analytical procedures. First, how were concentrations measured? Importantly, was nitrite detected and removed from the samples? Even if NO2- was minor say 5% relative to NO3-, due to its rapid exchange with water below pH of 10, this could have artificially lowered the measured _17O of the interpreted HNO3(g) or p-NO3-. As an example, a 5% NO2- contribution would have lowered the _17O of the targeted nitrate (assuming _17O _ 30‰ by 1.5‰ and could be the primary driver behind the _17O difference in the "speciated" nitrate. Additionally, which chemical conversion method was used? Azide/Acetic Acid Buffer? If so, please cite the appropriate references (i.e. McIlvin and Altabet, 2005). How might using the long-form _17O definition (ln(1+d17O/1000)-0.516*ln(1+_18O/1000)) impact the authors comparison to data using the more commonly used linear _17O definition (d17O-0.52*_18O) (i.e. Table 4)?

REPLY – We have expanded the analytical description.

Change in manuscript; section 2.3: « Nitric acid from nylon filters were extracted using 10 mL of 0.01M solution of NaCl. Particulate-NO$_3$ from Teflon filters were extracted in two portions of 6 mL of ultrapure water (ELGA). To reduce possible evaporation, filters were placed in an ultrasonic bath with ice. The extractions were performed during one hour and samples were left for 48 hours in a fridge to insure the complete extractions. The solutions were decanted and a small portion (1-2 mL) was used to determine concentrations. The remaining extracts were stored in the fridge for subsequent isotope analysis. The blanks from both filters were treated the same way.

Concentration of nitrates in Teflon and Nylon filter extracts, and in precipitation samples were determined at the Institut national de la recherche scientifique – Eau, Terre, Environnement (INRS-ETE). The determinations used an automated QuikChem 8000 FIA+ analyzer (Lachat Instruments), equipped with an ASX-260 series autosampler. The detection limit for the method with sulfanilamide (# 31-107-04-1-A with sulfanilamide) was 2 ppb N-NO$_3$/L (0.03 N-NO$_3$ umol/L). Nitrite concentrations were also measured in the extracts. Nitrite concentrations above the detection limit (0.016 mg N/L) were found in a handful of samples at Terrace Heights. These samples were excluded from the reported data. »

« The preparation steps involved conversion of nitrate-containing samples into nitrite ($NO_2^-$) using a custom-made cadmium column. The final preparation step involved using sodium azide to ultimately produce $N_2O$ (McIlvin and Altabet, 2005; Smirnoff et al., 2012). All extracted $N_2O$ was analyzed using a pre-concentrator (PreCon, Thermo Finnigan MAT) including a furnace with 'gold' wires, online with an Isotope Ratio Mass Spectrometer (Delta V Plus, Thermo Electron; Kaiser et al., 2007; Smirnoff et al., 2012). The utilized approach… »

« Extracts from filter blanks were processed in the same way. The blanks from nylon filters were not integrated for the calculation of the results due to their negligible size. Peak heights from the blanks resulting from Teflon filters were always below 10% of sample peaks and were also neglected. »

Comment 6: Sections 3.2-3.3: I'm a little unsure as to why the authors are spending so much time on the _18O + _17O "source effect". Their collections were from polluted air masses not direct stack emissions, such that _18O and _17O should effectively be wiped of any "source effect" due to the rapid equilibration of NOx and its oxidants. The authors suggest in 3.2, that due to elevated _18O + _17O there isn't a source effect (this is not surprising or a novel finding but expected), but extend the discussion of source effects in 3.3 despite ruling them out in 3.2. This seems a bit odd to me, and I think it would serve this manuscript better to simplify these sections into 1, removing source effect discussions. Instead the authors should focus on the unique oxidation chemistries of the polluted air masses to understand how differences in NOx oxidation cycling and post NO2 reactions would have altered _18O + _17O rather than a source derived _18O + _17O effect, but again this is complicated as previously mentioned because of the nitrate lifetime problem. We don't know that the nitrate sampled is from the targeted source area.

REPLY – Good piece of advice.

Change in manuscript: « We now show former Figure 1 in supplementary material (Fig. SM- 2). Figure 1 in the article now includes two location maps. We have removed the emphasis previously put on the individual sources by erasing former sub-section 3.3.

Comment 7 (MAJOR): (a) Section 4.1: The Alberta nitrate _18O and _17O relationships appear linear despite the author's claim to the contrary. Can the authors include regression statistics so that their argument is supported quantitively rather than qualitatively?

REPLY – If the reviewer refers to Figure 2, data per seasons are not numerous enough to assess non-linearity with confidence. However, this is a very minor point.

Change in manuscript: The sentence was erased.

(b) Much of the authors _17O range calculations and justifications are ad hoc. Can the authors properly justify their assumptions made in this calculation, specifically "50% contribution from each pathway for summer"?

REPLY – This assumption is just meant to help present alternatives. We do not claim that 50% is the proportion applicable for the collected samples.

Change in manuscript: none required.

(c) Additionally, can the authors propagate the error made in the suggested _17O ranges? The authors indicate that the _17O range "shrinks" during winter but their calculations indicate a larger range during the winter (winter: 26 to 37‰ summer: 20 to 29‰.

REPLY – The "shrinks" referred to the wider range estimated from the extreme cases discussed earlier in the paragraph (100% from the OH pathway or 100% for the N2O5 pathway; 25-45 ‰ for winter samples), however we realize the placement was misleading. We have clarified.

Change in manuscript in Subsection 4.1, end of first paragraph: «... the range is 26-37 ‰.»

(d) The authors conclusion that _17O of NO2 is not equal to the asymmetrical O3 is not new but rather expected, due to VOC oxidation contributions that have an NO oxidation branching ratio of 70 to 80%. Perhaps the authors should retry their calculations utilizing a more realistic approximated _17O of NO2. Also, could the authors compare their _17O in this region with the global _17O model?

REPLY – The goal of the calculation was to constrain the source $NO_2$ $\Delta^{17}O$ values based on those of nitrates, and it is not clear what the reviewer is suggesting. Use an approximate value of $NO_2$ $\Delta^{17}O$ to calculate the contributions from the different $NO_2$ oxidation pathways in different samples? That would require assuming the $NO_2$ $\Delta^{17}O$ value was constant, which would also be unrealistic. The contribution of $RO_2$ oxidation of NO is discussed in the paragraph following the calculation. The nitrate $\Delta^{17}O$ are compared with the global model in sub-section 3.5; $NO_2$ $\Delta^{17}O$ was not explicitly mapped in Alexander et al. (2009) for comparison.

Change in manuscript: the calculations and discussion of their implications are now presented in sub-section 4.1.

(e) Again, the calculated transit times of 9 minutes to 4 hours, indicates that not all of the sampled nitrate is derived from the targeted upwind region due to the chemical lifetimes of NOx and atmospheric lifetime of nitrate. I find it hard to believe any of the interpretation on _18O and _17O differences between p-NO3- and HNO3(g) because this speciation was likely not truly achieved given the method concerns already raised in this review and others. I recommend that this speciation discussion should be removed and _18O and _17O interruption should focus on total nitrate relative to wet-nitrate (which was hardly discussed in this manuscript!)

REPLY – See reply to comment 4 for speciation. At section 3.2 second paragraph, we added: « Though the number of samples were limited, w-$NO_3^-$ $\Delta^{17}O$ values were roughly correlated with the weighted average $\Delta^{17}O$ values of p-$NO_3$ and $HNO_3$ in samples covering the same time periods, consistent with scavenging of both $HNO_3$ and p-$NO_3$ by wet deposition. »

Comment 8: In general, I find the figures and tables difficult to read and interrupt (especially Figs. 3 and 4)

REP - We have modified caption of Figure 3 so the trends shown by the different symbols are easier to appreciate, and removed former Figure 4.

Change in manuscript, caption Figure 3: « Line-connected $\delta^{18}O$ and $\Delta^{17}O$ values (‰) for simultaneously collected $HNO_3$ (empty symbols) and p-$NO_3^-$ (solid symbols) from cold (blue) and warm (red) sampling periods. » Figure 4 is now removed.

Comment 9: The authors findings that meteorological parameters often correlate with HNO3, p-NO3- and their isotopic compositions isn't surprising (particularly phase separation) due to the well-established thermodynamic equilibrium of HNO3 and p-NO3- that determines this phase separation. This point however, directly contradicts that authors claim that reaction pathways (i.e. NO2 + OH vs N2O5 heterogenous rxn) had a significant role on the observed speciation and isotopic composition in Section 4.1.

REPLY – Actually the second point supports the first. Thermodynamic equilibrium will contribute to change the $\delta^{18}O$ values, but not the $\Delta^{17}O$ signals.

Change in manuscript: none really required, but we now present the comparison with other parameters in sub-section 3.4 and include the relevant data in Table SM-3.

Comment 10: Overall, I'm not convinced that during the summer, source effects lowered the anthropogenic originating nitrate _17O values. The simplest explanation for this observation should be NO oxidation contributions from RO2. Until the authors can explicitly rule out the "oxidation chemistry effect" by modeling or empirical evidence, I don't think the authors suggested conclusion should be drawn.

REPLY – We do not invoke a specific source effect, but sampling of p-$NO_3$ in early plumes characterized by $\Delta^{17}O$ values lower relative to those of $HNO_3$ due to differential depositional rates. The $RO_2$ oxidation contribution is not ruled out, but likely concomitant with differential deposition. We believe it is valuable to raise an alternative hypothesis in order to spur further testing of this common explanation.

Change in manuscript: Sub-section 4.2 has a new title and has been modified.

1-The manuscript by Savard and colleagues presents interesting data worthy of publication. However, I found the results and discussion sections muddled and the important points worthy of highlighting buried. (A) The manuscript could be improved by a focus on key findings and condensing or eliminating repetitive sections. For example, page 10 lines 24-35 are potential explanations for low observed d18O, D17O values with important implications multiple communities. (B) Additionally, challenging the assumption of NOx isotopic steady state with O3 is a key takeaway (not mentioned until page 14). Why are these not highlighted more prominently in the abstract? The current conclusions in the Abstract and Conclusion by comparison are weak "isotopic signals of.... are not interchangeable", and "invariably interchangable".

Reply: (A) We have removed repetitive text parts, and after all changes we have reduced the overall text length by one page. We agree with the reviewer that our suggestion about NOx isotopes not at steady state with $O_3$ should be given more importance at the beginning of the article. NOTE that this is in total disagreement with the main concern of reviewer 3. Here we decide to keep our first suggestion on 'no steady state for some samples', as no other suggestion has been put forward to explain the low $\Delta^{17}O$ values in non-winter p-$NO_3$. We now explain better the rationale for studying the anthropogenic nitrate samples and highlight better the importance of studying separately nitric acid in the introduction (also briefly summarized in the abstract). (B) We have modified the content of the conclusion to better emphasize the implications of our results and interpretation.

Changes in manuscript: (A) in the abstract: « Recent laboratory experiments suggest that the isotopic equilibrium between $NO_2$ (the main precursor of $NO_3^-$) and $O_3$ may take long enough under certain field conditions that, nitrates may be formed near emission sources with lower isotopic values than those formed further downwind. Indeed, previously published field measurements of oxygen isotopes in w-$NO_3^-$ and p-$NO_3^-$ samples suggest that abnormally low isotopic values might characterize polluted air masses. However, none of the air studies have deployed systems allowing collection of samples specific to anthropogenic sources in order to avoid shifts in isotopic signature due to changing wind directions, or separately characterized $HNO_3$ with $\Delta^{17}O$ values. Here we have used a wind-sector-based, multi-stage filter sampling system and precipitation collector to simultaneously sample $HNO_3$ and p-$NO_3^-$, and co-collect w-$NO_3^-$. The nitrates are from various distances (<1 to >125 km) downwind of different anthropogenic emitters, and consequently from varying time lapses after emission. »

(B) in the conclusion « The $HNO_3$, w-$NO_3$ and p-$NO_3$ from anthropogenic sources in central and southern Alberta, simultaneously collected with wind sector-based conditional sampling systems produced $\delta^{18}O$ and $\Delta^{17}O$ trends confirming the previous contention that regional ambient conditions (e.g., light intensity, oxidant concentrations, RH, temperature) dictate the triple isotopic characteristics and oxidation pathways of nitrates.

The gaseous form of nitrate ($HNO_3$) having distinct isotopic characteristics relative to the wet and particulate forms implies that understanding nitrate formation and loss requires characterizing the nitrate species individually. Particulate-$NO_3^-$ in these samples generally shows

higher $\delta^{18}O$ and $\Delta^{17}O$ values than $HNO_3$ in the fall-winter period as the heterogeneous $N_2O_5$ pathway favours the production of p-$NO_3^-$. In contrast, $HNO_3$ has higher $\delta^{18}O$ and $\Delta^{17}O$ values during warm periods, which we propose is due to faster dry deposition rates relative to p-$NO_3^-$ in the event that low-$\Delta^{17}O$ $NO_2$ is oxidized in the plume. The mechanisms conferring nitrate with relatively low isotopic values, whether oxidation before $NO_x$-$O_3$ equilibrium is reached or higher contributions from $RO_2$, are likely to be observed in anthropogenic polluted air masses. »

2-The manuscript needs a map to put the respective sampling sites and the surround potential sources in a spatial context. Without this information, it is not possible to discern how far sites are from each other.

Reply: Positioning the sites was requested by reviewers 1 and 3. We have added two maps showing the various sites (new Fig. 1).

Change in manuscript: The location maps are now shown in Figure 1.

3-The authors points out "very few" air masses passed over other sources outside the targeted ones in the preceding 24 hours. For those that did, are they removed from the analysis? Why is this data not shown- as it seems relevant.
Reply: It's not possible to remove back trajectories from the integrated samples. This analysis was to determine if the conditional sampling method was indeed capturing air masses from the source region(s). Since we have de-emphasized the exact source types, the sentence has been revised to be more accurate.

Change in manuscript: Revised this sentence as follows: "*Back trajectories run using the HYSPLIT model (Stein et al., 2015; Rolph, 2017) for every hour of sampling verified that the conditional sampling approach collected air masses that had primarily passed over or near the targeted source (i.e. there was no landscape feature that decoupled wind direction from back trajectories). A sample plot of back trajectories from Genesee is show in Figure SM-2. A sample plot is now shown in Figure SM-2*"

4-The long variation in sampling times is concerning. For example, individual sample deployments ranged from 5 to 113 days. The authors should explore whether there is any evidence that the length of this sampling time caused any artifacts in their results.

Reply: If the sampling protocols had generated artefacts, they would be reflected in the isotopic difference between p-$NO_3$ and $HNO_3$: p-$NO_3$ would have higher $\delta^{18}O$ values than $HNO_3$ particularly during summer. But as explained in the article, the trends we observe are opposite to this prediction. Therefore, if there are effects due to sampling in our results, they perhaps minimize the ranges that lead to our conclusions. In other words, removing potential effects would just strengthen our conclusions. Additionally, if artefacts were created by the sampling procedure, the impact would increase with the duration of sampling. Now, we have plotted our $\delta^{18}O$ and $\delta^{17}O$ values against duration of sampling for nitrate types, and we find no systematic increase (or decrease) of values with duration. the Alberta data are shown below for all sampling periods in the 6 upper panels; the results previously obtained (Smirnoff et al., 2012) using the same sampling protocols for Highway 401 traffic are shown in the lower two panels.

[Figure]

Change in manuscript: no change required.

5- Page 5, line 10: What is "preconized"?

Reply: we have now clarified.

Change in manuscript: «… for the protocols used here.»

6- Page 5: line 33: Why was MAD scaled by 0.6745? Where did this number come from? Needs justification.

Reply: it is the normal divisor widely used for allowing the comparison of the mean absolute deviation (MAD) with the regular standard deviation (SD).

Change in manuscript: « Finally, for comparability with the more familiar standard deviation, the MAD was scaled using the standard 0.6745 divisor to give the modified median absolute deviation (M.MAD), which is consistent with the standard deviation in the event that the distribution is Gaussian… »

7-The authors conclude Elliott et al found minimal fractionation between d18O of pNO3- and HNO3. Figure 2 from that paper shows _10 permil differences during summer in the d18O values of these two components.

Reply: Good catch. This just not explained properly. We indeed know about figure 2 (see paragraph before last of section 3.5 in first manuscript submitted). We have now corrected.

Change in manuscript: «… minimal fractionation for p-NO$_3^-$ and HNO$_3$…»

8-Page 1, line 19: Add distance to state how far collection sites are from major sources (i.e., from x to y km).

Reply: Done.

Change in manuscript: «…and co-collect w-NO$_3^-$, at various distances (<1 to >125 km) downwind from five different anthropogenic emitters.»

9-Figures 2a-c. Include 1:1 line to clarify your conclusion that data show a "vertical extent". More clarification needed here in this analysis/conclusion.

Reply: this part is removed to avoid confusion.

Change in manuscript: Sentence removed.

10-Discussion: Lines 18-35. This reads like intro text/results. Revise to lead off with a topic sentence that highlights your major finds and built supporting text around this.

Reply: Good point. This entire sub-section is now presented in the section 3.

Change in manuscript: See new sub-section 3.3.

11-Authors state that "Anthropogenic emitters involving combustion (O2) may generate primary NOx at or near sources that tend to carry low d18O and D17O values". It is not clear whether this is in reference to prior published studies, or whether this is one

of their conclusions. Either way, it needs more justification.

Reply: Clearly not one of our interpretation, but a widely accepted information from the literature. The references solely appears in caption of Table 5 in the version the reviewer worked with. We now have removed that part of the text during the restructuration of the article.

Change in manuscript: sentence removed.

12-For the analysis on page 10 lines 7-22, it is not clear how the authors determined the relative proportion of R1, R2, R3 to calculate the influence of O3 on oxidation pathways from NO2 to HNO3 (R4, R7, R8).

Reply: These calculations did not make any assumptions about the relative contributions of R1, R2 and R3; rather, the purpose was to derive the possible range of $\Delta^{17}O$ values of $NO_2$ from which the observed $HNO_3$ and $p-NO_3^-$ could have been derived. We've added a topic sentence to this section (in 4.1) to make this more clear.

Change in manuscript: Text added in section 4.1: "*In the present sub-section, we estimate the $\Delta^{17}O$ values of $NO_2$ involved during the production of the Alberta nitrates based on the observed values and discuss the implications of these estimations.*"

13-I found the discussion on page 11 lines 13-25 very intriguing. How might seasonal differences in lifetimes affect how far different constituents travel? Is there any prior modeling work (e.g., GEOS-CHEM) that could support these ideas?

Reply: Longer lifetimes with respect to deposition would certainly allow for greater dispersal, depending on the wind speeds and chemical lifetimes (relatively long for nitrates). We note that dry deposition velocities are highly variable in space and time since they depend on both meteorological and detailed surface characteristics. These are quite challenging to model, and estimates from different models typically differ by at least a factor of 2 even with the same input data. Our main point is that $HNO_3$ has a deposition velocity consistently higher than that of $p-NO_3$ (though it depends on the size of the particle) as calculated using the multiple resistor model at numerous sites in Canada (Zhang et al., 2009).

Change in manuscript: none required

14-The authors refer to "seasonal changes in planetary boundary layer heights" but don't explicitly state what these changes are and how they could impact their results.

Reply: Seasonal changes in the planetary boundary layer height may also affect the impingement of emission plumes on the measurement sites, and thereby the relative amounts of fresh vs background nitrates. For example, winter boundary layer heights can be lower than the 138-155 m stacks at the three coal-fired power plants. During those periods, fresh plumes from those stacks may be emitted into the free troposphere where they are much less likely to mix back down to the surface where the samples are collected. This would be consistent with the observed higher $\Delta^{17}O$ values in winter, if the air masses reaching the sampling site are older than when the plume is emitted within the boundary layer. On the other hand, a relatively

shallow boundary layer that DOES encompass the source stacks would concentrate the plume within the layer and lead to faster nitrate formation. Since we do not have data on the PBL height to determine which was most likely during our sampling periods, we decided to remove the sentence.

Change in manuscript: Sentence removed from manuscript.

15-Section 4.2 This reads as Results rather than Discussion.

Reply: We have now placed this part in the results.

Change in manuscript: See new sub-section 3.4.

16-Conclusions: What is "invariably interchangeable"?

Reply: The isotopic values of $HNO_3$ and p-$NO_3$.

Change in manuscript: «... and disavows the assumption that isotopic values of the dry nitrate phases are invariably interchangeable.»

**Point-by-point REPLY to Anonymous Referee #3**

The manuscript presents a new data set on the isotopic oxygen composition of nitrates in the Alberta region, Canada. It focuses specifically on the speciation of nitrate (aerosols, gases, wet phases) in conjunction with a potential source effect. The manuscript can be considered as the second part of a previous manuscript published in Atmospheric Environment (Savard et al. 2017, doi: 10.1016/j. atmosenv. 2017.05.010) which dealt only with the 15N/14N ratio of the same samples.

1- As a first question, I wonder why the authors did not submit this second part to AE for coherency reasons or add this part to above mentioned reference.

Reply: The option of adding this data set and interpretation to the AE paper on $\delta^{15}N$ values of all N-species investigated ($NH_3/NH_4$ and all nitrates) was not feasible, as it would have made a much too long article. We are convinced there is a natural separation of the two articles. They are addressing different questions: the AE article aims at evaluating the source fingerprinting potential of $\delta^{15}N$ values in all forms of N emission (reduced and oxidized) from various anthropogenic sources; whereas the article submitted to ACP aims at understanding better the NOx oxidation pathways and the $NOx/HNO_3/p\text{-}NO_3$ relationships.

Change in manuscript: not applicable.

2- Generally speaking, I find the article unclear and confusing, with too many figures and tables that are not all very informative and easy to read. The explanations given are often ad hoc and not supported by strong observations, experiments or theory. Overall, the article is not of sufficient interest with new and strong novelty to recommend its publication in ACP.

Reply: This article presents the first $\Delta^{17}O$ values in $HNO_3$, simultaneously sampled with $p\text{-}NO_3$. These measurements are difficult to obtain, as they require elaborated field collection campaigns and state-of-the-art analytical systems. The data presented are new and they prompted a new interpretation in terms of non-equilibrated $NOx\text{-}O_3$, suggested for the first time for field samples. For these reasons, we believe the article is worth publishing in ACP.

Change in manuscript: In this new version of the article, we have restructured the sequence of data presentation and interpretation: all results are in section 3 and all discussion and the main interpretations are in section4. We have also modified the introduction to better express the rationale: « While $HNO_3$ and $p\text{-}NO_3^-$ can be in equilibrium (e.g. if $p\text{-}NO_3^-$ is in the form of solid $NH_4NO_3$), this is not always the case, for example, if nitrate is bonded to calcium or dissolved in liquid water on a wet particle (see section 3.3). They also have different lifetimes with respect to wet scavenging and dry deposition, and may differ in their formation pathways as well. Therefore, investigating the mass independent and dependent oxygen fractionations in nitrates separately collected may help identifying their respective formation and loss pathways, and provide additional constraints on processes controlling their distribution. »

3- A major flaw of the paper is the angle taken by the authors to present and interpret their data in relation with a source effect as they did in Savard et al. (2017). It is well accepted by the community that the oxygen isotopes of nitrate are driven by oxidations and not by

source effect, an idea back up by a large number of experiments and observations from the first studies (Michalski et al. 2003) to most recent ones (Guha et al. 2017).

The reviewer agrees with a key statement of the introduction in the originally submitted article that O isotopes should reflect oxidation pathways (see Introduction second paragraph; and sub-section 4.4 line 4). Confirming no direct source effect on the O isotopes was expected, and this confirmation IS NOT the main contribution highlighted in the article. The original introduction clearly states the rationale for sampling downwind from anthropogenic source: «In those studies, $\delta^{18}O$ and $\Delta^{17}O$ values were suggested to be useful to apportion the contribution of emission sources to regional atmospheric nitrate loads. However, the signals of precursor $NO_x$ emitted from the same sources may quickly get modified through isotopic equilibration with $O_3$, so that the original source signals may be difficult to recognize.» In the new version of the article, we further explain the pertinence of evaluating source effects, not in terms of distinguishing the ultimate sources among themselves, but for assessing if low $\Delta^{17}O$ values previously suggested as indicative of anthropogenic emissions are characterizing anthropogenic emissions, generally speaking. Do low $\Delta^{17}O$ values reflect a larger role of $RO_2$ in the oxidation of anthropogenic NOx emissions in fresh plumes? This question is of interest to the scientific community as it is still debated in the literature (Proemse et al., 2013; Guha et al., 2017).

Change in manuscript: Introduction «. Triple oxygen isotope characterizations of field $NO_3^-$ samples are not yet widespread. Also rare are the nitrate $\delta^{18}O$ and $\Delta^{17}O$ values of field samples downwind from $NO_x$-emitting sources at mid-latitudes (Kendall et al., 2007; Proemse et al., 2013). The few existing studies have chiefly characterized w-$NO_3^-$ or the sum of p-$NO_3^-$ and $HNO_3$ (Michalski et al., 2004; Morin et al., 2007; Morin et al., 2008; Alexander et al., 2009; Morin et al., 2009; Proemse et al., 2012; Guha et al., 2017), and suggested these indicators would be useful to trace atmospheric nitrate in water (Kendall et al., 2007; Tsunogai et al., 2010; Dahal and Hastings, 2016), or to apportion the contribution of anthropogenic emissions to regional atmospheric nitrate loads (Proemse et al., 2013). »

4- (A) The authors should have eliminated the source effect in one or two sentences (B) and concentrated on the oxidation mechanism by adding ancillary data such as NOx, O3 concentrations, photo-dissociation rates such JNO2 and/or modeling.

Reply: (A) We significantly shortened all text parts related to the individual anthropogenic emissions. We have used $O_3$ and NOx mixing ratios and presented our statistics in Table 6. The fraction of each sample collected during daylight hours (correlations also shown in Table 6) was judged as a reasonable proxy for the amount of sample collected during active photochemistry. Detailed $jNO_2$ calculations are of limited value for effort considering that we do not have radiation data on site to account for cloud cover. We do recognize the importance of modelling, but it was not the purpose of our research, and our data can be made available for modellers when the article is accepted (a table with all pertinent information can be placed in the supplemental information).

Changes in manuscript: (A) not applicable.

(B) p. 6. First paragraph of 3.2. «As expected, there is no systematic tendency when looking at the samples collected from the anthropogenic sources: CFPP $HNO_3$ and p-$NO_3^-$ have the highest $\delta^{18}O$ and $\Delta^{17}O$ averages, but not the highest w-$NO_3^-$ values; chemical industries show the lowest $\delta^{18}O$ and $\Delta^{17}O$ averages for w- and p-$NO_3$, but not for $HNO_3$. This observation indicates that the oxygen isotopes in the three nitrate species are not predominantly source-dependent (see also Fig. SI-2), as previously suggested in the literature (Michalski et al., 2003).»

In addition, former sub-section 3.3 has been removed and part of the text merged with sub-section 3.2.

  5-  The sampling protocols are poorly described. Blanks are not given, neither pumped volumes. No filter breakthrough, saturation, interference, efficiency is evaluated (see Talbot et al. 1990 for the use of nylon filter), especially in response to RH which is known to greatly influence volatilization of p-NO3 (Cheng et al., 2012) and HNO3 collection efficiency (Appel et al., 1980) on filters. Actually, such samplings artefacts can alternatively be an argument to explain the tight correlation observed between HNO3/p-NO3 isotopes and RH (Table6). It is also surprising to see the use of filter pack system to differentiate p-NO3 and HNO3 collection as most modern systems and networks use impregnated denuder systems (Cheng et al., 2012, ChemComb (Thermo Fisher scientific), MARGA (Metrohm) or URG gas-aerosols denuder samplers)) to avoid loss p-NO3 by H2SO4 acidification or gain of HNO3 by adsorption on collected alkaline aerosols.

Reply: The filter pack system is based on the ones used by two long-standing networks (Environment and Climate Change Canada's CAPMoN and the U.S. Environmental Protection Agency's CASTNET), but we can certainly provide more background about the historical testing of these filters and the rationale for their use in this study. For example, Anlauf et al. (1986) found that breakthrough was ~3% for filter loadings up to 3 times higher than the maximum loading in this study. Filter loadings and pumped volumes can be reported with the tabulated sample and ancillary data mentioned above.

Denuders were considered but were not used, partly because of the lack of capacity and established quality control protocols at the CAPMoN laboratory. Also because of the higher potential complications due to the longer deployments in these remote locations (necessary to collect sufficient material for isotopic analysis at low ambient concentrations) compared to the typical urban networks with high concentrations that allow using denuders. We had concerns about: (a) the likely positive artefact of "passive" sampling due to diffusion into the denuder during the periods without pumping in this sector-based approach; (b) the likelihood of capturing coarse PM on the denuder if no size-selective inlet was used (which was not wanted due to the desire to capture p-NO$_3$ on coarse PM); and (c) the higher potential for condensation and dripping within the denuders during multiple day/night cycles and resulting loss of coating/sample. While we acknowledge that small part of the HNO$_3$ is likely volatilized p-NO$_3$, as discussed in the last paragraph of sub-section 3.1, fractionation during this process would be negligible during winter sampling and bias the HNO$_3$ $\delta^{18}O$ values low relative to p-NO$_3$ in summer, while the observations showed the opposite seasonal pattern. In addition, this would be a mass-dependent process and therefore have no effect on the $\Delta^{17}O$ signals, so it cannot explain the correlations between RH and $\Delta^{17}O$ values.

Changes in manuscript: A discussion of field tests and intercomparisons has been added to section 2.2: *"Ambient air was pulled through a three-stage filter pack system to collect, sequentially, particulate matter on a Teflon filter, gaseous nitric acid (HNO$_3$) on a Nylasorb nylon filter, and gaseous ammonia on a citric acid-coated Whatman 41 filter. The Teflon-nylon filter method for p-NO$_3^-$ and HNO$_3$ has been extensively compared and evaluated, and is currently used by national monitoring networks targeting regional background sites, CAPMoN in Canada and CASTNet (Clean Air Status and Trends Network) in the United States. Previous testing by Anlauf et al. (1985, 1986) showed negligible collection of HNO$_3$ on the Teflon filter, <3% breakthrough of HNO$_3$ from the nylon filter with loadings more than three times higher than reported here, and blanks for p-NO$_3^-$ and HNO$_3$ of approximately 0.2 µg N/filter. Intercomparisons with more labor-intensive methods, such as tunable diode laser absorption*

*spectroscopy and annular denuder-filter pack systems, have shown evidence of some volatilization of ammonium nitrate from the Teflon filter leading to a negative bias in p-NO₃⁻ and positive bias in HNO₃ under hot (> 25 °C) and dry conditions, particularly in high ambient concentrations (e.g., Appel et al., 1981). However, other field studies have shown no significant differences in HNO₃ between filter packs and denuder and/or TDLAS systems (Anlauf et al., 1986; Sickles et al., 1990) or mixed results (Spicer et al., 1982; Zhang et al., 2009). While those studies used short-duration sampling, a comparison for weekly samples at a lower-concentration site showed good agreement between filter pack and denuder values for most of the study but potential interference from HNO₂ (nitrous acid) on the nylon filter in two samples (Sickles et al., 1999). Based on the conditions in Alberta, we estimate that there is little or no volatilization of NH₄NO₃ for samples with mean temperatures below 5 °C, but that there may be up to 30% nitrate loss in the warmest sampling periods."*

6- Location descriptions and context refers systematically to the Savard et al, 2017 papers which does not help to contextualize what the data plotted really mean. Samples cover different total air sampling time, from 21 to 360h and deployment times. We don't know if the sampling is dominated by nighttime or daytime chemistry, if they are rich/poor NOx/O3 atmospheres.

Reply: We now present two location maps as Figure 1 and briefly describe the locations and contexts of sampling (not much added because we have removed the former emphasis that was placed on the various source types; see further).

We have explored the relationship between the isotopic results and daylight fraction and found a significant inverse correlation with isotopic values of p-NO₃, but not with HNO₃ (Table 6). We now provide all the data, including available O₃ and NOₓ concentrations, in summary tables.

Changes in manuscript: New Figure 1 for locations and all data presented in Tables SM-1, 2 and 3 in the supplementary material.

7- Replicated samples were pooled at two sites (Genesee and Vauxhall) making even more difficult to know what plotted data really represent.

Reply: Each point on the plots represents a single sampling period at a given site, whether several samples were pooled or not. Where samples were not pooled, the individual data were used to estimate the reproducibility of the combined sampling and analytical approach (Table 2), but in the end, the average values (Tables SM-1 and 2) were plotted. In brief, last paragraph of sub-section 2.2 of the submitted article clearly explains what the data represent.

Changes in manuscript: See Tables SM-1, 2 and 3 in Supplementary Material.

8- Section 3.3 is useless considering what the authors say in the first line of 3.2. It is thus detrimental to the understanding of the study to see an idea accepted by the whole community, namely that oxygen isotopes of nitrate are controlled by oxidation, starting to appear in the middle of the discussion. Discussion about source-driven effect should be evacuated as soon as possible with no more than one/two sentences, such as "we did not observe any significant correlations between O-isotopes and source types or wind direction".

Reply: We have now placed former Figure 1 in Supplementary Material, and replaced it in article by a location map (new Fig. 1), removed the emphasis previously put on the individual sources by eliminating sub-section 3.3.

Changes in manuscript: see new sub-section 3.2, Figure 1, and Figure SM-2.

9- The discussion about the different oxidation pathways to explain the season trends is classic and does not bring any new idea or interpretation. The only original observation is the difference in isotopic compositions between HNO3 and nitrate but it is questionable given the above reserve mentioned.

Reply: We specifically write in the article that the interpretation in terms of chemical pathways for cold and warm periods agree with former studies. We do no claim this is a novel interpretation. However, this aspect is crucial for estimating the isotopic signals of precursor NOx, and for further explaining the causes for the changes in isotopic differences between $pNO_3$ and $HNO_3$.

As mentioned in reply to point 5, as well as in the text, the mentioned sampling artefacts cannot cause the observed $\Delta^{17}O$ differences between $pNO_3$ and $HNO_3$. Additionally, if such effects had imprinted the isotopic results, the sampling duration would show a systematic influence on the results. We do not observe such an influence on our results (see reply to comment 4 of referee #2). Therefore, we maintain that our conclusions rely on high quality data.

Changes in manuscript: see sub-sections 4.1 and 4.2.

10- Moreover, there is no systematic trend about HNO3 being enriched or depleted as function of season and with respect to p-NO3. In figure 3, there is few cases where summer p-NO3 have higher _17O than HNO3. It is thus difficult to understand why authors want to explain the greater _17O of HNO3 in summer over p-NO3. Furthermore, the discussion falls short to give an acceptable explanation (lines 10 to 25 of page 11).

Reply: The data on Fig. 3 show clearly both positive and negative values of $\Delta^{17}O_{(HNO3)}$-$\Delta^{17}O_{(pNO3)}$, with a somewhat positive trend with temperature. While we hypothesize that negative values may be due to the larger contribution of the $N_2O_5+H_2O$ heterogeneous reaction to p-$NO_3$, we felt it was necessary to propose a mechanism for the positive differences also observed in most spring and summer samples. At this stage, to our knowledge, the best hypothesis for explaining higher $\Delta^{17}O$ values in $HNO_3$ is that the deposition of $HNO_3$ is greater than the one of p-$NO_3$, a difference in rates that is much stronger during summer than winter. This mechanism combined with the NOx-$O_3$ isotopic equilibrium can explain our data set. If readers want to suggest different lines of interpretation, we will gladly receive them.

Changes in manuscript: see new caption of Figure 3 and new sub-section 4.2.

11- The idea that NO2 is not in isotopic equilibrium with O3 in summer is odd. First if equilibrium is not reached, it should be amplified in winter, not in summer when O3 is at max (Angle et al., 1989) and photolysis at its peak.

The point is taken, though $O_3$ is at its peak in April/May in Alberta (NAPS), but we should note that there was not a full year of data at any single site, and the two sites where summer samples were primarily gathered were the closest to the $NO_x$ sources (Table 1). We added Fig. 5 illustrating our examination of the relationship between the maturity of a plume relatively to the isotopic signal of the nitrates (graph of $\Delta^{17}O$ values (weighted average) as a function of $NO_2$ concentration divided by p-$NO_3$ plus $HNO_3$ concentrations). The relationship is strong ($r^2$=0.81) and supports our interpretation: the more mature is a plume (low $NO_2$ concentration), the higher are the $\Delta^{17}O$ values in the nitrates, i.e. the higher is the $O_3$-derived O content in the nitrates).

Changes in manuscript: in sub-section 4.1: « The fact that we find the lowest isotopic values in summer p-NO$_3^-$ samples collected from various anthropogenic sources at distance less than 16 km supports this suggestion (Table 1). Another argument supporting this interpretation is the strong correlation between $\Delta^{17}O$ values and the maturity of a plume as expressed by the NO$_2$ content divided by sum of dry nitrates (Fig. 6). The results reflect the higher content of O$_3$-derived O in dry nitrates from mature plumes, i.e., with relatively low NO$_2$ contents. »

[Figure]

**Figure 5: Weighted $\Delta^{17}O$ average (‰) for the sum of dry nitrates as a function of NO$_2$ concentration divided by p-NO$_3$ plus HNO$_3$ concentrations, a ratio indicative of the maturity of a plume.**

12- Moreover, NO2 is the precursor of HNO3 and p-NO3, if not in equilibrium it should impact equally HNO3 and p-NO3. To twist this basic idea, the authors claim that HNO3 is faster scavenged from the atmosphere than p-NO3 but they have no quantitative data to show that is realistic in their environmental context.

Reply: The reader should keep in mind that the main precursor of p-NO$_3$ is HNO$_3$. Additionally, we refer the reader again to the articles cited in the original manuscript showing higher dry deposition rates for HNO$_3$ (Zhang et al 2009; Benedict et al., 2013).

Changes in manuscript: none required.

13- Neither the authors tested the hypothesis that NO2 is indeed not in equilibrium with O3. If Michalski et al. (2014) showed that the time-scale for equilibrium is strongly dependent on local sunlight conditions and NOx/O3 ratio and can be longer than 1h, they fall short to tell us why isotope equilibrium will take longer than chemical steady state (is it due to the time for ozone or NO2 to reach its isotopic equilibrium composition? or unrealistic O/O3/NO/NO2 ratios after model initialization since chemical steady state will be reached in min and will radically change the NO2/O3 ratio?). In another study, Morin et al. (2011) using a true atmospheric model modeled _17O of NO2 using different realistic atmospheric conditions and environments. They showed that NO2 is largely at isotopic equilibrium except during few night hours but with little impact on prognosticated _17O of nitrate (1 to 2 ‰ at most). Clearly, this section needs more and deeper investigations and critical review of published works.

Reply: It is important to note here that Morin et al. (2011) did not model any fresh NOx emissions and they used a 24-hour model spin up before reporting isotopic composition of $NO_2$. Therefore, their modeling results are not comparable with the nitrates collected within minutes to hours of fresh $NO_x$ emissions. It would be interesting to see their spin-up period.

The field measurements reported here were not specifically designed to test Michalski's proposition on NOx-$NO_3$ isotopic disequilibrium (Michalski et al. 2014 paper), but they provide an opportunity to verify whether or not non steady-state is possible in the field by allowing examination of anthropogenic nitrates collected after various duration of sampling. The results are such that they suggest the nitrate loads are at least partly reflecting NOx-$NO_3^-$ isotopic disequilibrium.

Considering that this article represents the first investigation of simultaneously sampled nitrates in precipitation, gas and particulate forms for their $\delta^{18}O$ and $\Delta^{17}O$ values, we think it deserves to be available to the large readership of ACP. We have now clarified the lines of interpretation the article provides by placing less emphasis on the types of sources from which the plumes were sampled, and by reworking the results and interpretation (section 3) and discussion (section 4).

Changes in manuscript: Former sub-sections 4.2 and 4.4 are now sub-sections 3.4 and 3.5, respectively; abstract lines 16-22, and new sub-section 3.2; new title and trimmed sub-section 4.2; new title for sub-section 4.3.

   14- Explanation of correlations with meteorological parameters are ad hoc and rough with a weak constrain on possible mechanisms. For instance, correlations with RH and T can be the result of the winter/summer meteorology. Summer is more oxidant but also warmer, sunnier and lower RH. Should all correlations be interpreted, as much of them are not independently related?

Reply: We judge pertinent suggesting an interpretation for these correlations as they relate to reactions summarized in Table 5. The text describing this interpretation is short, and it is now placed with the results and interpretation section (3), right after the interpretation in terms of oxidation pathways (sub-section 3.3).

Changes in manuscript: see new section 3.4

   15- Correlations with co-pollutants are contradictory as mentioned by the authors (lines 27-35, page 12) and lead to no strong conclusions. In this regard and in my view, the authors should have reported O3, NOx and JNO2 time-series to give some context. Only gross correlations are reported with most the variables interdependent.

REPLY: We are not convinced that time series of $O_3$, $NO_x$ and $jNO_2$ would be meaningful in interpreting these integrated and intermittent samples, which is why we used average values over the sampling times for $O_3$ and $NO_x$ analysis. However, those average values are now reported in the new data table of the supplementary information. The use of daylight fraction rather than $jNO_2$ is discussed in point 4. Graphs of the $O_3$ time series at (a) Genesee, (b) Terrace Heights, and (c) Fort Saskatchewan, including during sampling periods, are below. Filled circles identify periods when nitrate sampling was taking place for at least half of the hour, with alternating black and red to indicate different samples. Due to expected large spatial gradients in $NO_x$, we decided the comparison with $NO_x$ 5 km away was problematic and removed it. However, $NO_2$ data measured on-site now appear reported where available.

[Figure]

Changes in manuscript: average values are in Table SM-3.

16- There is other imperfection that bother me. For instance, what was a hypothesis at the beginning (the none equilibrium of NO2 with O3) has now become a certainty (line 6 page 13).

REPLY: Good point. The previous sentence was : « However, $NO_2$ not in isotopic equilibrium with $O_3$, and/or NO reacted with $RO_2$ significantly influenced the overall results.»

Changes in manuscript: sub-section 4.2, p. 12, line 25: *However, $NO_2$ not in isotopic equilibrium with $O_3$, and/or NO reacted with $RO_2$* **may have** *significantly influenced the overall results.*»

17- Finally, the idea that low values of _17O can be linked to the rapid oxidation of anthropogenic NOx is attractive but would have merited more investigation such as following for example the NOx/NO3- ratio to give some clue about the aging of the air masses.

REPLY: This is a good suggestion. A technique for actively sampling integrated $NO_2$ and NO concentrations was developed with some success through the course of the study, but since it was an evolving methodology, we have acceptable $NO_2$ concentrations only at 3 of the 4 sites, both in the Edmonton urban area.

We have plotted the $\Delta^{17}O$ values (weighed average) of p-$NO_3$ and $HNO_3$ as a function of $NO_2$ concentrations / (p-$NO_3$ plus $HNO_3$ concentrations), and there is clear inverse correlation. In other words, when $NO_2$ is high (low nitrates in early plume), the $\Delta^{17}O$ values are low, and when $NO_2$ is low (late plumes), the nitrates $\Delta^{17}O$ values are high. These $NO_2$ concentrations also appear in Table SM-1.

Changes in manuscript: See new Figure 5 and reply to comment 11.

**Cited references**

Anlauf, K.G., Fellin, P., Wiebe, H.A., Schiff, H.I., Mackay, G.I., Braman, R.S., Gilbert, R. A comparison of three methods for measurement of atmospheric nitric acid and aerosol nitrate and ammonium (1985) Atmospheric Environment (1967), 19 (2), pp. 325-333.

Anlauf, K.G., Wiebe, H.A., Fellin, P. Characterization of Several Integrative Sampling Methods for Nitric Acid, Sulphur Dioxide and Atmospheric Particles (1986) Journal of the Air Pollution Control Association, 36 (6), pp. 715-723.

**Point-by-point REPLY-II to Anonymous Referee #3**

1-I maintain that this article should have been submitted to AE as a Part II for coherency but this is a minor comment
REPLY- We do not see the advantage for the readership in this proposition. Publishing two articles dealing with distinct issues of atmospheric science isotopic applications, with several months in between, is commonly done through different journals, even if reporting data from a single region.

Change in manuscript: none required.

2-I don't think that "new and novel" data are sufficient arguments to guaranty their publications. New and novel does not mean correct and I have major reserves about their correctness (see point 5)
REPLY- We mean 'New and novel' implying that the data is QA/QC checked, i.e., valid.

Change in manuscript: see point 5.

3-I don't think that the authors demonstrated in any way that they have collected nitrate from specific sources whatever O isotopes track or not these sources. To pretend that, they need to provide observations that either NOx, nitrate (or any other tracers, CO, O3) are different than background atmosphere. According to the set-up of their experiment, I have serious doubts that sampling air from hours to days will guaranty a permanent sampling of the plume emissions. Conditional sampling based on wind direction is not enough. In this way, I found the title misleading, firstly because as said above, there is no guaranty they have sampled specific anthropogenic sources and secondly, as they mentioned, the scrambling of the oxygen atoms erases source fingerprints.

REPLY- We did not claim a "permanent sampling" of plume emissions, as we agree that would be unrealistic. The goal was to isolate anthropogenic emissions at various locations (distances), downwind from the sources, with the emissions subject to field conditions and atmospheric processing (i.e. not stack sampling). By necessity, it is clear that background contributed to the sampled load, but not in significant proportions. A few pieces of evidence support this. One is the low background concentrations; background particle nitrate and nitric acid concentrations at Wood Buffalo National Park in northern Alberta, where CAPMoN began sampling in 2014, averaged 0.016 and 0.020 µg N m$^{-3}$, respectively, for >2 years of monitoring. Concentrations from the conditional sampling at the sampling sites studied here were up to 20 times higher, suggesting that the collected samples have greater p-NO$_3$ and HNO$_3$ concentrations than background. Second, back trajectories confirmed that the conditional sampling method sampled air masses that predominantly passed over or near the sources. In addition, we obtained some model output from a 2-week simulation in 2013 (Makar et al., 2018) illustrating the significantly higher concentrations of nitric acid downwind of the coal-fired power plants near Genesee. In

this snapshot, the three power plants are shown as brown dots and the three Edmonton-area sampling locations are shown as purple stars. The NO2 plumes are shown as well to illustrate the change in direction of the plume. It is clear that at the Genesee site, the bulk of the nitric acid at this particular time resulted from the nearby power plants. Using the conditional sampling weighted the samples more heavily when this was the case.

[Figure]

Changes in manuscript: The title is changed to "The δ$^{17}$O and δ$^{18}$O values of atmospheric nitrates simultaneously collected *downwind of* anthropogenic sources – Implications for polluted air masses."

Text edited in section 2.2: *"Back trajectories run using the HYSPLIT model (Stein et al., 2015; Rolph, 2017) for every hour of sampling verified that the conditional sampling approach collected air masses that had primarily passed over or near the targeted source (i.e. there was no landscape feature that decoupled wind direction from back trajectories). A sample plot of back trajectories from Genesee is show in Figure SM-2."*

Text added to section 3.2: *"Background sites for this region are sparse, but concentrations at Cree Lake in neighbouring Saskatchewan were the lowest in Canada reported up to 2011 (Cheng and Zhang, 2017), and 2014-2016 measurements at Wood Buffalo National Park on the northern Alberta border revealed similar average concentrations of 0.02 µg N/m$^3$ for both HNO$_3$ and p-NO$_3^-$ (preliminary internal data). Therefore, the lowest concentrations in our samples approached average background concentrations, while the highest were 20 or more times higher than regional background."*

4-Giving the Pearson's correlation in a table is not enough to judge the correctness of the correlation. Readers need to see the dispersion of the data and species time-series within the sampling time windows to connect sources with sampling.

REP- It is unclear to us how informative it will be to see hourly time series, which must then be compared with an integrated sample, however, in the interests of fulfilling the request we have added the plots for O3.

Change in manuscript: The data for each sampling period are now reported in Tables SM-1 through SM-3. In addition, plots to show the O$_3$ measured at nearby stations including during sampling periods are shown in reply I to reviewer 3, comment 15. Due to expected large spatial gradients in NO$_x$, we decided the comparison with NO$_x$ 5 km away was problematic and removed it. However, NO$_2$ measured on-site is now reported where available.

5 (merged with 9; see below) -It is wrong to think that denuders are best used in urban area. Denuders to collect HNO3 are used in the most remote regions of world (eg Antarctica, Jourdain and Legrand, 2002, Legrand et al., 2017). Denuders that are operational at 1m3/h exists (URG or Thermo Chemcomb), thus minimizing the collection time. Proper set up can limit passive sampling and restricted it to gas diffusion, exactly their purpose. The denuder tubes are the norm to collect acid gases with minimal interferences. They are promoted by the largest atmospheric aerosol networks (EMEP, EPA-method IO4-2). The method used by the authors (1st filter for p-NO3 and 2nd nylon filter for HNO3) is not the reference set up used to separate p-NO3 and HNO3. It is a set up used mainly to collect total nitrate. The difference in  17O between p-NO3 and HNO3 is not a guaranty that the different phases are sampled correctly. Finally, as already mentioned, the fact that a method is published and accepted does not exempt the authors to show us that they can correctly reproduce it. Authors should be able to provide the data and demonstrate that blanks, interferences, efficiencies etc. can be quantified and/or corrected (Finlayson-Pitts&Pitts, 2000). Jourdain, B., and Legrand, M.: Year-round records of bulk and size-segregated aerosol composition and HCl and HNO3 levels in the Dumont d'Urville (coastal Antarctica) atmosphere: Implications for sea-salt aerosol fractionation in the winter and summer, J. Geophys. Res., 107, 4645, 10.1029/2002jd002471, 2002. Legrand, M., Preunkert, S., Wolff, E., Weller, R., Jourdain, B., and Wagenbach, D.: Year-round records of bulk and size-segregated aerosol composition in central Antarctica (Concordia site) – Part 1: Fractionation of sea-salt particles, Atmos. Chem. Phys., 17, 14039-14054, 10.5194/acp-17-14039-2017, 2017. EMEP manual for sampling and chemical analysis, Norwegian Institute for Air Research, Kjeller, NorwayEMEP/CCC-Report 1/95, 2001. Compendium of Methods for the Determination of Inorganic Compounds in Ambient Air (EP A/625/R-96/010a) – method IO4-2 Finlayson-Pitts, B.

J., and Pitts, J. N.: Chemistry of the upper and lower atmosphere: Theory, experiments and applications, Academic Press, San Diego, CA, 969 pp., 2000.

9- I will give one example where 17O of nitrate can be modified. If a nitrate particles seating on the filter is hit by a sulfuric acid droplet and the pH of this sulfuric acid is low enough, then isotopic exchange between HNO3 and H2O can be triggered. I'm not saying it is what is happening with the author's sampling system but again my main point is that 17O cannot be at the same time the causal and the effect, i.e. the variable to be explained and the variable to explain: the observed difference between 17O HNO3 and p-NO3 can't be used as an argument to validate a sampling system. Where is the constrain showing me that such difference simply exists and it is not an artifact? For me it is a self-realization observation.

REPLY- We have responded to points 2, 5 and 9 together since we interpreted them as raising closely-related issues.

We acknowledge that it is possible to use denuders in remote areas, our point was that there are specific and well-regarded networks of rural and remote stations that continue to use filter-based sampling. Since our system was using the established methods of one of those networks (CAPMoN), and evaluation of the method blanks, collection efficiencies and interferences have been previously reported, it seems excessive to us to require repetition of these tests in every report using the same method. Where we developed a new method (for NO and $NO_2$ active sampling, not reported here), blanks and breakthrough tests were done and evaluated before reporting results. Again, denuders were considered, but we chose not to use them for several reasons: (1) we were not certain of the denuder capacity or the ambient levels of $HNO_3$ in this region prior to the study; (2) given the potential for long periods without flow in the conditional sampling setup, denuders open to the atmosphere would be likely to passively sample during non-pumped periods; while (3) denuders with size-selective impactors at the inlets would result in screening out nitrate on some particles, with the size cutoff varying as the pumps cycled on and off in (sometimes) 5-minutes periods. Note that isotopic results based on collection with filter packs are not new. For instance, isotopic values for dry deposition (p-$NO_3$ and $HNO_3$) actively collected with filter packs over a week have previously been validated in eastern USA (Elliott et al., JGR, vol 114, 2009).

Our primary concern with this system was the volatilization that is well documented, and that would affect both the O and N isotopes in a mass dependent and highly temperature dependent way. Therefore, as we stated, we evaluated the relative $HNO_3$ and p-$NO_3$ $\delta^{18}O$ and $\delta^{15}N$ values (as well as $\delta^{15}N$ in $NH_3$ and p-$NH_4$), and their pattern with temperature, to judge whether this was strongly affecting the results. We did not draw conclusions about the artifact based on $\Delta^{17}O$ values, just stated that mass-dependent volatilization would not affect the $\Delta^{17}O$ value, which is correct. While the reviewer does suggest a possible mechanism that would affect $\Delta^{17}O$ values (exchange with $H_2O$ due to highly acidic particles), this scenario is unlikely in the studied region. Where we analyzed a complete suite of major ion data from the particle filter (2 of the 4 sites), the charge balance was always positive due to both relatively high $Ca^{2+}$ and $NH_4^+$. In any case, this scenario would similarly influence p-$NO_3$ collected in a denuder-filter pack sampling system.

Change in manuscript: A discussion of field tests and intercomparisons has been added to section 2.2: *"Ambient air was pulled through a three-stage filter pack system to collect, sequentially, particulate matter on a Teflon filter, gaseous nitric acid ($HNO_3$) on a Nylasorb nylon filter, and gaseous ammonia on a citric acid-coated Whatman 41 filter. The Teflon-nylon filter method for p-$NO_3^-$ and $HNO_3$ has been extensively compared and evaluated, and is currently used by national monitoring networks targeting regional background sites, CAPMoN in Canada and CASTNet (Clean Air Status and Trends Network) in the United States. Previous testing by Anlauf et al. (1985, 1986) showed negligible collection of $HNO_3$ on the Teflon filter, <3% breakthrough of $HNO_3$ from the nylon filter with loadings more than three times higher than reported here, and blanks for p-$NO_3^-$ and $HNO_3$ of approximately 0.2 µg N/filter. Intercomparisons with more labor-intensive methods, such as tunable diode laser absorption spectroscopy and annular denuder-filter pack systems, have shown evidence of some volatilization of ammonium nitrate from the Teflon filter leading to a negative bias in p-$NO_3^-$ and positive bias in $HNO_3$ under hot (> 25 °C) and dry conditions, particularly in high ambient concentrations (e.g., Appel et al., 1981). However, other field studies have shown no significant differences in $HNO_3$ between filter packs and denuder and/or TDLAS systems (Anlauf et al., 1986; Sickles et al., 1990) or mixed results (Spicer et al., 1982; Zhang et al., 2009). While those studies used short-duration sampling, a comparison for weekly samples at a lower-concentration site showed good agreement between filter pack and denuder values for most of the study but potential interference from $HNO_2$ (nitrous acid) on the nylon filter in two samples (Sickles et al., 1999). Based on the conditions in Alberta, we estimate that there is little or no volatilization of $NH_4NO_3$ for samples with mean temperatures below 5 °C, but that there may be up to 30% nitrate loss in the warmest sampling periods."*

Revised text end of section 3.1: *"We therefore conclude that, while volatilization may occur in the summer samples, other isotope effects must be larger in order to lead to the observed differences. In addition, volatilization would cause mass-dependent fractionation and would not affect the $^{17}O$ anomaly; therefore, $\Delta^{17}O$ values remain robust tracers in this situation."*

6-If the main point of the paper has nothing to do with targeted source types, title of the paper should not give the opposite impression. The authors did not convince me that they have sampled "true" anthropogenic plumes. Nothing in the presented data indicate such thing
REPLY- The main point of the article is not to address differences between various anthropogenic sources, but to examine isotopic trends in anthropogenic plumes sampled at different periods (various distances from sources, i.e., various delays after emissions), with the specific objective of verifying if low $\Delta^{17}O$ values exist in some contexts. The results clearly show that they do. This finding has implications for interpreting isotopic data collected downwind from anthropogenic sources in general. The title refers to this aspect, which the article largely discusses.

Change in manuscript: title reworded « The $\Delta^{17}O$ and $\delta^{18}O$ values of atmospheric nitrates simultaneously collected downwind of anthropogenic sources – Implications for polluted air masses »

7-When I said what the data mean, I mean what atmospheric context are they representing? Not how have they been obtained? Plotting altogether data that represent averaged hours, averaged

days, mix of nighttime or daytime in different proportion etc. does not help the reader to contextualize the observations.

REPLY- Merged parallel samples (Genesee and Vauxhall) constitutes a physical average of atmospheric characteristics at a given area. This operation can be compared with the calculated average through 4 parallel samples (4 other sites), which only aimed at determining the reproducibility of our sampling and analytical protocols. We have judged this type of care determinant and crucial in guaranteeing the quality of the data. Not clear what the reviewer means in the second point. There is no other way to plot the data since each sample is integrated over a variety of conditions. We would agree that higher-frequency field measurements would add to our understanding of the processes, though it would be challenging to collect enough material for isotopic analysis as methods currently stand.

Change in manuscript: additional details for each sample (air volumes, daylight fraction, etc.) are included in Tables SM-1 to SM-3.

10- Again I do not see any systematic trend in 17O difference between p-NO3 and HNO3 with season (fig3). In summer, two out of four have 17O nitrate > 17O HNO3 and in winter they have only two events, a very weak statistic. I may not see the same data than the authors and any help from the other reviewers will be welcome. I have no explanation (as I'm not convinced by the correctness of the data by the way) but I can easily found one if I pile up few none demonstrated hypothesis, like the authors did with 1- HNO3 is formed from non-equilibrated NOx/O3 system and 2- HNO3 is faster scavenged. I can propose the formation of lower 17O p-NO3 by the heterogeneous reaction 2NO2 + H2O(s) –> HNO3(ads) + HONO (Finlayson-Pitts, 2009), or higher 17O HNO3 by NO3 + RH –> HNO3 in gas phase nighttime oxidation. Finlayson-Pitts, B. J.: Reactions at surfaces in the atmosphere: integration of experiments and theory as necessary (but not necessarily sufficient) for predicting the physical chemistry of aerosols, PCCP, 11, 7760-7779, 10.1039/b906540g, 2009.

REPLY- The trends are various and each deserves attention. We discuss all of them in the article.

Change in manuscript: none required.

13- Well, I disagree again with the authors. One of the strongest argument used in this paper is to claim that NOx-O3 are not in isotopic equilibrium, using mainly Michalski paper as support. So, it is up to the authors to first question Michalski's paper and its conclusions. In Michalski, the atmospheric application of their model is really poorly described. It is not mentioned if at initialization, ozone has already its isotopes at equilibrium (as it should be in the atmosphere considering the life-time of O3 vs NOx). Yet ozone formation is the only reaction creating 17O-excess, and since chemical steady state is quickly reached, equilibrium of 17O among all species can't be reached faster than O3 own equilibrium time in Michalski's model. Clearly, the limiting step in Michalski's model to propagate 17O is ozone formation and not NOx/O3 interaction. If ozone is in isotopic equilibrium, any new population of NO2 formed by O3+NO (modulo the

two-to-one atom transfer) will have the same isotopic composition that the O-atom transfer (if kinetic fractionation is neglected). It is thus simply a question of reservoir of NO2 versus flux of NO2 to reach equilibrium. Isotopic abundance has nothing to do here. Let's imagine that O3 is already in isotopic equilibrium, further formation/destruction have no effect on ozone 17O. Let's imagine further that NOx and O3 are in chemical/isotopic equilibrium (new O3 formed has the same isotopic composition than consumed O3 as O3 isotope is controlled by pressure and temperature only). Suddenly, a new pool of NO is emitted. NO will be converted to NO2 by O3 contained in the surrounding atmosphere upon mixing and thus NO2 will be formed at the rate of the Leighton cycle in this system. The characteristic time of the isotopic transfer from O3 to NO2 is simply twice the time of the Leighton cycle. Obviously, a plume model is necessary to calculate air mass mixing but as a first approximation, we can assume that the plume is continuously replenished by surrounding O3 so that O3 stays constant. The characteristic time, Tau, at which the non-equilibrated isotopic NOx reservoir is replaced by the isotopic equilibrated NO2 is simply twice the size of NO2 reservoir divided by the speed of Leighton cycle, either NO+O3 reaction or JNO2 depending on the chemistry context, as one of these reactions is the limiting step. Using Michalski first simulations, NO = 23 ppbv (assumed NO2/NOx = 0.3 for fresh plume), NO2 = 10 ppbv, O3 = 50 ppbv and k = 2e-14 molecules cm-3 s-1, J = 0,007 s; then Tau = 2/J = 4,8 min. In 20 min NO2 is at 98 % in isotopic equilibrium. Using Michalski second simulations NO2= 0,03ppb, NO=0,003 ppb (assumed NO2/NOx = 0.9 for remote place), O3 = 5 ppb, Tau = 2 [NO2]/(k[NO][O3]) = 120 min; 8h to reach 98 % of equilibrium. Apparently, a much less favorable situation (due to the very low NO, strongly limiting the recycle speed) but this simulation at low ozone, 5 ppb, is taken as an illustration of Morin's observation (Morin et al., 2007). However, such situation corresponds to an ozone depletion event (due to the high concentration of bromine) for which NOx are recycled through the BrO + NO and not NO+O3 reaction. In a more rural situation (Rohrer et al., 1998), NO2 = 1,4 ppb, NO = 0,3 ppb, O3 = 25 ppb, Tau = 11 min Rohrer, F., Brüning, D., Grobler, E. S., Weber, M., Ehhalt, D. H., Neubert, R., Schüßler, W., and Levin, I.: Mixing Ratios and Photostationary State of NO and NO2 Observed During the POPCORN Field Campaign at a Rural Site in Germany, Journal of Atmospheric Chemistry, 31, 119-137, 10.1023/a:1006166116242, 1998.

REPLY- We do not want to discuss the fundamentals of Michalski et al.'s paper here, this is not the place. However, we trust that the conclusion of Michalski's experiments open up the possibility of seeing isotopic disequilibrium in natural samples under certain conditions. In fact, given the unknowns, the back-of-envelope calculations above (20 min and 8 h to 98% of equilibrium in the two scenarios) are roughly in agreement with the timescales shown in Michalski et al. (Fig. 8). Therefore, it is not clear why the reviewer is not comfortable with the results of their simulations. Given that transit times from the closest point sources to our measurement sites averaged 25 minutes (range 9-55), and that we were sampling the fraction of NO$_x$ that had been converted to nitrate and therefore "frozen" in $\Delta^{17}O$ at the point of conversion, contributions from unequilibrated NO$_x$ are not ruled out by the tau of 11 minutes suggested by the reviewer for similar conditions.

Based on this discussion and comments from another reviewer, we have realized that the salient point is whether the amount of nitrate formed shortly after emission (before the NO$_2$-O$_3$

isotopic equilibration) is enough to contribute significantly to the samples. Our calculations indicate that it is feasible, and we've updated the manuscript to include those results.

We would like to be clear that we are not claiming to present definitive evidence of this phenomenon in the atmosphere. Indeed, we do suggest in the manuscript that the contribution from enhanced $RO_2$ could also give a similar result, as has been previously hypothesized. However, since the possibility of incomplete $NO_x$ equilibration retained in nitrate field samples was a new idea supported by winter data when RO2 cannot play significantly, NOx-O3 disequilibrium was also highlighted. We have carefully reviewed the wording of the document to be sure not to overstate our confidence in the mechanism, as was suggested in the earlier comments.

Change in manuscript:.
     Text added to section 4.1: "*While the fraction of $NO_x$ converted to nitrate in this transit time may be small, these are large sources of $NO_x$ in an area with very low background nitrates. For example, a plume containing 10 ppb of $NO_2$ mixing with background air with 0.1 ppt of OH (Howell et al., 2014) would produce $HNO_3$ via R6 at a rate of 0.011μg N $m^{-3}$ $min^{-1}$ at T = 7 °C (Burkholder et al., 2015), or an equivalent amount of a typical sample in 10 minutes. Even if equilibration with $O_3$ is established within a few minutes, the nitrate produced in the interim can constitute a substantial fraction of the sample collected nearby*."

     Subsection 4.3; first paragraph: « As expected, the measured oxygen isotopes of the various nitrate groups are consistent with exchange with $O_3$ and oxidation through the well-known OH and $N_2O_5$ oxidation paths. However, $NO_2$ not in isotopic equilibrium with $O_3$, and/or NO reacted with $RO_2$ may have significantly influenced the overall results. »

     Subsection 4.3; end of first paragraph: «Meanwhile, our results raise the question: are these overall effects observable in triple oxygen isotopes of nitrates from other polluted sites?»

In summary, authors' reply did not change my position and did not convince me. Because the idea that 1- HNO3 has a different 17O composition than p-NO2 and 2- NOx is not in isotopic equilibrium are strong and important conclusions, before propagating these idea in the literature, strong lines of evidence should be provided. I don't think the current work carries such guaranty.
REPLY- Point 1 refers to measurements; the difference in isotopic signals is an observation, it is not an idea inferred through an interpretation. We have shown that the data are valid. Point 2 is a suggestion for which all arguments are exposed in the article; the reader gets substantial information allowing for a personal opinion to be made; this suggestion may create a debate (indeed, it already has) and spur further testing of the hypothesis through additional measurements and plume modelling, a healthy outcome in science.

[revised manuscript text omitted]

bonded to calcium or dissolved in liquid water on a wet particle (see section 3.3). They have different lifetimes with respect to wet scavenging (Cheng and Zhang, 2017) and dry deposition velocities (Zhang et al., 2009), and may differ in their formation pathways as well. Therefore, investigating the mass independent and dependent oxygen fractionations in nitrates separately collected may help identifying their respective formation and loss pathways, and provide additional constraints on processes controlling their distribution.

Here we have characterized nitrate collected downwind of five emission sources in central and southern Alberta, Canada, namely: (1) coal-fired power plants, (2) city traffic, (3) chemical industries and metal refining, (4) fertilizer plant and oil refinery, and (5) gas compressors plus cattle and swine feedlots. To this end, we employed wind-sector-based active samplers to collect $HNO_3$ and p-$NO_3^-$ as well as w-$NO_3^-$ downwind of the source types. The objective of this work was to assess the atmospheric $NO_x$ reaction pathways and determine processes responsible for the distribution of $HNO_3$, and w- and p-$NO_3^-$ in a mid-latitudinal region.

**2 Methodology**

**2.1 Regional context**

[revised manuscript text omitted]

[Figure]

5    **Figure 2: Triple O isotopic results (‰) obtained for simultaneously collected atmospheric HNO₃ (A), w-NO₃⁻ (B) and p-NO₃⁻ (C), in Alberta, identified by sampling periods (cold months - blue; warm months - red).**

[Figure]

**Figure 3:** Line-connected δ¹⁸O and Δ¹⁷O values (‰) for simultaneously collected HNO₃ (empty symbols) and p-NO₃⁻ (solid symbols) from cold (blue) and warm (red) sampling periods.

[Figure]

[Figure]

**Figure 4:** Schematic outline of main steps in the production of Alberta nitrates: NOₓ-O₂-O₃ photochemical cycle (1) and reaction with RO₂ (2) modify NOₓ source signals (R2, R3, R9); oxidation of NOₓ produces HNO₃ along the N₂O₅ (4) or OH (5) pathways. The grey line represents NOₓ from photochemical cycling with O₂ and O₃ (Michalski et al., 2014). The direction of arrows 1 to 5 indicates how the isotopic values would evolve along the different chemical reactions; the positions of these arrows are arbitrary.¶

**Figure 4:** Isotopic results (‰) for p-NO₃⁻ identified by sampling periods (solid lines), compared with summer and winter trends obtained for Arctic sites (dashed lines; derived from ln (1+ δ) in Morin et al., 2008).

[Figure]

**Figure 5: Weighted average Δ$^{17}$O (‰) for the sum of dry nitrates as a function of NO$_2$ concentration divided by p-NO$_3$ plus HNO$_3$ concentrations, a ratio indicative of the maturity of a plume.**

[Figure]

[Figure]

**Figure 6: Isotopic results (‰) for p-NO$_3^-$ identified by sampling periods (solid lines), compared with summer and winter trends obtained for Arctic sites (dashed lines; derived from ln (1+ δ) in Morin et al., 2008).**

**Figure 6: Isotopic ratios (‰) for atmospheric p-NO$_3^-$, w-NO$_3^-$ and HNO$_3$ samples in cold and warm periods from central and southern Alberta (this study), compared with previously published winter and summer bulk and throughfall deposition samples from the oil sands (OS) region from northern Alberta (Proemse et al., 2013), and p-NO$_3^-$ in-stack emissions data for an OS upgrader located in the same region (Proemse et al., 2012). The grey dotted line connects NO$_x$ from theoretical combustion with O$_2$**

10 **isotopic composition and at isotopic equilibrium with tropospheric O$_3$ (Michalski et al., 2014).**

| Page 1 : [1] Supprimé | Cole,Amanda [Ontario] | 08/05/2018 3:29:00 PM |
|---|---|---|

r

| Page 1 : [1] Supprimé | Cole,Amanda [Ontario] | 08/05/2018 3:29:00 PM |
|---|---|---|

r

| Page 1 : [1] Supprimé | Cole,Amanda [Ontario] | 08/05/2018 3:29:00 PM |
|---|---|---|

r

| Page 1 : [1] Supprimé | Cole,Amanda [Ontario] | 08/05/2018 3:29:00 PM |
|---|---|---|

r

| Page 1 : [1] Supprimé | Cole,Amanda [Ontario] | 08/05/2018 3:29:00 PM |
|---|---|---|

r

| Page 1 : [1] Supprimé | Cole,Amanda [Ontario] | 08/05/2018 3:29:00 PM |
|---|---|---|

r

| Page 1 : [1] Supprimé | Cole,Amanda [Ontario] | 08/05/2018 3:29:00 PM |
|---|---|---|

r

| Page 1 : [1] Supprimé | Cole,Amanda [Ontario] | 08/05/2018 3:29:00 PM |
|---|---|---|

r

| Page 1 : [1] Supprimé | Cole,Amanda [Ontario] | 08/05/2018 3:29:00 PM |
|---|---|---|

r

| Page 1 : [2] Supprimé | Savard, Martine | 23/03/2018 2:23:00 PM |
|---|---|---|

polluted

| Page 1 : [2] Supprimé | Savard, Martine | 23/03/2018 2:23:00 PM |
|---|---|---|

polluted

| Page 1 : [2] Supprimé | Savard, Martine | 23/03/2018 2:23:00 PM |
|---|---|---|

polluted

| Page 1 : [3] Supprimé | Cole,Amanda [Ontario] | 07/05/2018 4:08:00 PM |
|---|---|---|

pathways

| Page 1 : [3] Supprimé | Cole,Amanda [Ontario] | 07/05/2018 4:08:00 PM |
|---|---|---|

pathways

| Page 1 : [3] Supprimé | Cole,Amanda [Ontario] | 07/05/2018 4:08:00 PM |
|---|---|---|

pathways

| Page 1 : [3] Supprimé | Cole,Amanda [Ontario] | 07/05/2018 4:08:00 PM |
|---|---|---|

pathways

| Page 1 : [4] Supprimé | Savard, Martine | 24/01/2018 1:11:00 PM |

five different

| Page 1 : [5] Supprimé | Cole,Amanda [Ontario] | 07/05/2018 4:08:00 PM |

the

| Page 1 : [6] Supprimé | Savard, Martine | 25/04/2018 4:31:00 PM |

The summer pattern

| Page 1 : [7] Supprimé | Cole,Amanda [Ontario] | 07/05/2018 4:09:00 PM |

valid

| Page 1 : [8] Supprimé | Savard, Martine | 25/04/2018 10:02:00 AM |
| --- | --- | --- |

while the l

| Page 1 : [8] Supprimé | Savard, Martine | 25/04/2018 10:02:00 AM |
| --- | --- | --- |

while the l

| Page 1 : [8] Supprimé | Savard, Martine | 25/04/2018 10:02:00 AM |
| --- | --- | --- |

while the l

| Page 1 : [8] Supprimé | Savard, Martine | 25/04/2018 10:02:00 AM |
| --- | --- | --- |

while the l

| Page 1 : [8] Supprimé | Savard, Martine | 25/04/2018 10:02:00 AM |
| --- | --- | --- |

while the l

| Page 6 : [9] Supprimé | Savard, Martine | 24/05/2018 9:49:00 AM |
| --- | --- | --- |

Concentration of nitrates on Teflon and Nylon filter extracts, [AC1]and in precipitation samples were determined at Institut national de la recherche scientifique–Eau, Terre, Environnement (INRS-ETE). The determinations used an automated QuikChem 8000 FIA+ analyzer (Lachat Instruments), equipped with an ASX-260 series autosampler. The detection limit for the method (# 31-107-04-1-A with sulfanilamide) was 2 ppb $N-NO_3/L$ (0.03 $N-NO_3$ umol/L). The concentration for NO and $NO_2$ collected with samplers of Maxxam Analytics were determined by Maxxam Analytics using an ion chromatograph.

| Page 8 : [10] Supprimé | Savard, Martine | 22/03/2018 2:55:00 PM |
| --- | --- | --- |

**3.3 Covariations of $\Delta^{17}O$ and $\delta^{18}O$ values in nitrates from individual sources**

The $p-NO_3^-$, $w-NO_3^-$ and $HNO_3$ values co-vary when identified by source type in the $\delta^{18}O$ and $\Delta^{17}O$ space (Fig. 1). The isotopic range for a single source can be as large as 6 ‰ for $\Delta^{17}O$ values and 19 ‰ for $\delta^{18}O$ values in $HNO_3$, 12 and 17 ‰ in $w-NO_3^-$, and 7 and 21 ‰ in $p-NO_3$. Each source type clearly exhibits nitrate $\Delta^{17}O$ and $\delta^{18}O$ with a specific grouping. The CFPP $w-NO_3^-$ results show a range similar to the $HNO_3$ results, but lower $\delta^{18}O$ values than the $HNO_3$ and $p-NO_3^-$ groups. The few other precipitation samples show $\delta^{18}O$ and $\Delta^{17}O$ values generally higher than the $p-NO_3^-$ and $HNO_3$ samples, again with exception of the chemical and metal industries.

The $HNO_3$ samples from a given source type tend to have a higher $\delta^{18}O$ value for a given $\Delta^{17}O$ value than $p-NO_3^-$ (or *vice versa*; Fig. 1). These observations suggest that the contribution of oxidation pathways leading to $HNO_3$ and $p-NO_3^-$ are not identical, or that there is an isotope fractionation in the conversion of $HNO_3$ to $p-NO_3^-$.

Regarding the potential for identifying nitrate sources, it appears that using $\delta^{18}O$ and $\Delta^{17}O$ values for such a task is not feasible, as previously suggested in the literature (Michalski et al., 2003). This interpretation stems from the fact

that nitrate species show either continuous trends regardless of their sources (p- and w- NO$_3$) or overlapping source results (HNO$_3$; Fig. 1).

The individual range of points identified by source may partly reflect different initial ambient conditions and rates of changes in ambient conditions during NO$_x$ oxidation (Fig. 1; see Section 3.5). Specifically, each isotopic range may depict the progressively changing influence of ozone due to ambient conditions through time. Indeed, the atmospheric samples were collected repeatedly over several weeks or months at a given site (near a given source), and consecutively from one site to the other over more than three years; samples undeniably incorporate N-species produced under significantly changing ambient conditions.

| Page 11 : [11] Supprimé | Cole,Amanda [Ontario] | 22/05/2018 12:29:00 AM |
|---|---|---|

Sub-sections 3.3 to 3.6 determined the main atmospheric chemical pathways responsible for the production of the Alberta nitrates. In the next section, we will further peruse the isotopic specificities of the produced data set and outline other key mechanisms affecting the distribution of nitrates.

| Page 11 : [12] Supprimé | Savard, Martine | 22/03/2018 10:37:00 AM |
|---|---|---|

The Alberta nitrate values do not fall on a single line but, rather, show a vertical extent in the $\delta^{18}O$ and $\Delta^{17}O$ space (Fig. 2) that exceeds the precision of the data (Section 2.3 and Table 2). This observation differs from several studies that measured bulk nitrate or a single nitrate species and reported $\delta^{18}O$ and $\Delta^{17}O$ sets as linear.

| Page 11 : [13] Supprimé | Cole,Amanda [Ontario] | 24/05/2018 12:28:00 AM |
|---|---|---|

Considering the relevant oxidation reactions shown in Table 5, anthropogenic atmospheric nitrates incorporate O atoms from three main molecules, O$_2$ (via RO$_2$ - R3, and possibly source NO$_x$- R1), O$_3$ (via NO$_2$, NO$_3^-$ and N$_2$O$_5$ – R2, R4, R7-R8) and H$_2$O (via OH, R5-R6). These molecules carry distinct isotopic signals that will partly determine the final $\delta^{18}O$ and $\Delta^{17}O$ values of the nitrate products. The $\delta^{18}O$ and $\Delta^{17}O$ values of O$_2$ are 23.5 and 0 ‰, respectively. Anthropogenic emitters involving combustion (O$_2$) may generate primary NO$_x$ at or near sources that tend to carry low $\delta^{18}O$ and $\Delta^{17}O$ values(Zel'dovich, 1946; Lavoie et al., 1969). This primary NO$_x$ (>90 % emitted as NO) cycles through NO-NO$_2$-O$_3$-NO numerous times before it is removed in R6. OH typically has negative $\delta^{18}O$ values and a $\Delta^{17}O$ value equal to 0 ‰ as it rapidly exchanges O isotopes with water vapour (Dubey et al., 1997; Röckmann et al., 1998). We obtained the average of precipitation $\delta^{18}O$ values for each sampling period at the studied sites (OPIC, 2017), and calculated the vapour signal using water-vapour fractionation factors (Clark and Fritz, 1997). Next, using fractionation factors between OH in equilibrium with H$_2$O vapour (Walters and Michalski, 2016), we calculated that the $\delta^{18}O$ values would range between -83 and -62 ‰. Peroxy radicals mostly derive from O$_2$ at mid latitudes (Michalski et al., 2003; Morin et al., 2007; Alexander et al., 2009), but they have a non-zero $\Delta^{17}O$ signal (1-2 ‰) due to the role of ozone in the HO$_x$ cycle (Morin et al., 2011). However, their $\delta^{18}O$ values are difficult to measure, so they can only be inferred based on assumptions (+23.9 ‰; Fang et al., 2011; Guha et al., 2017). The $\delta^{18}O$ and $\Delta^{17}O$ values of bulk

O$_3$ are generally between 90 and 120 ‰, and 30 and 34 ‰, respectively, but the transferable signals are suggested to be around 130 and 39 ‰ at mid-latitudes (Vicars and Savarino, 2014). Moreover, NO$_x$ modelled at isotopic steady state with tropospheric O$_3$ yields 117 and 45 ‰ in $\delta^{18}$O and $\Delta^{17}$O, respectively (Michalski et al., 2014). This neglects the contribution of NO oxidation by RO$_2$ (R3), which will reduce the steady-state $\Delta^{17}$O and $\delta^{18}$O of NO$_x$ below the O$_3$-only oxidation value. The foregoing review of isotopic signals provides context to the interpretation of our data, keeping in mind that mass-dependent fractionation has likely played a role in determining nitrate $\delta^{18}$O values[AC2].[AC3]Amanda to revisit… (see section SM.1)

| Page 13 : [14] Supprimé | Cole,Amanda [Ontario] | 22/05/2018 12:58:00 PM |
|---|---|---|

Another argument supporting this interpretation is the strong correlation between $\Delta^{17}$O values and the maturity of a plume as expressed by the NO$_2$ content divided by sum of dry nitrates (Fig. 6). The results reflect the higher content of O$_3$-derived O in dry nitrates from mature plumes, i.e., with relatively low NO$_2$ contents. Seasonal changes in the planetary boundary layer height may also affect the impingement of emission plumes on the measurement sites, and thereby the relative amounts of fresh vs background nitrates.

**Supplementary Material – ACP 1103**

**The $\Delta^{17}O$ and $\delta^{18}O$ values of atmospheric nitrates simultaneously collected downwind of anthropogenic sources – Implications for polluted air masses**

5  Martine M. Savard[1*], Amanda Cole[2], Robert Vet[2], Anna Smirnoff[1]

[1] Geological Survey of Canada (Natural Resources Canada), 490 de la Couronne, Québec (QC), G1K 9A9, Canada
[2] Air Quality Research Division, Environment and Climate Change Canada, 4905 Dufferin St., Toronto (ON), M3H 5T4, Canada

*Correspondence to*: Martine M. Savard (martinem.savard@canada.ca)

**Table SM-1. Sampling sites, duration of sampling, average results obtained for the parallel sampling with the CAPMoN systems.**

| # | Site | Deployment | Collection | Days | Sampling hours | Mean (total for pooled) air volume (m³) | p-NO$_3^-$ Mean (total for pooled) loading (ug N) | Air conc (ug N/m³) | δ$^{18}$O (‰) | δ$^{17}$O (‰) | Δ$^{17}$O (‰) | HNO$_3$ Mean (total for pooled) loading (ug N) | Air conc (ug N/m3) | δ$^{18}$O (‰) | δ$^{17}$O (‰) | Δ$^{17}$O (‰) | Daylight fraction (sunrise to sunset) | NO$_2$ in sample train* (ug N/m³) |
|---|------|-----------|-----------|------|----------------|------------------------------------------|------|------|------|------|------|------|------|------|------|------|------|------|
| 1 | Genesee | 2010-09-30 | 2010-10-05 | 5 | 23 | 59 | 8.0 | 0.14 | 70.3 | 63.8 | 26.9 | 8.4 | 0.15 | 66.9 | 59.4 | 24.3 | 0.49 | |
| 2 | Genesee | 2010-10-08 | 2010-10-18 | 10 | 42 | 99 | | | | | | 6.6 | 0.07 | 63.3 | 56.1 | 22.9 | 0.65 | |
| 3 | Genesee | 2010-10-18 | 2010-11-08 | 21 | 39 | 87 | 13.4 | 0.15 | 62.1 | 55.9 | 23.3 | 10.2 | 0.12 | 69.2 | 61.1 | 24.8 | 0.33 | |
| 4 | Genesee | 2010-11-08 | 2011-01-31 | 84 | 360 | 517 | 181.6 | 0.35 | 83.2 | 74.2 | 30.5 | 40.2 | 0.08 | 81.7 | 71.9 | 29.0 | 0.31 | |
| 5 | Genesee | 2011-02-22 | 2011-04-28 | 65 | | | | | | | | | | | | | | |
| 6 | Genesee | 2011-04-28 | 2011-06-20 | 53 | 144 | 189 | 19.4 | 0.10 | 67.3 | 60.9 | 25.6 | 24.9 | 0.13 | 67.6 | 60.2 | 24.7 | 0.71 | *0.82* |
| 7 | Vauxhall | 2011-10-25 | 2011-11-17 | 23 | 152 | 255 | 22.1 | 0.09 | 64.8 | 61.3 | 27.3 | 6.9 | 0.03 | 68.3 | 61.4 | 25.5 | 0.35 | *0.05* |
| 8 | Vauxhall | 2011-11-17 | 2011-12-01 | 14 | 89 | 176 | 15.9 | 0.09 | 60.7 | 57.0 | 25.1 | 4.8 | 0.03 | 61.7 | 55.9 | 23.5 | 0.28 | *0.12* |
| 9 | Vauxhall | 2011-12-01 | 2011-12-13 | 12 | 128 | 235 | 31.1 | 0.13 | 63.9 | 60.2 | 26.7 | | | | | | 0.34 | |
| 10 | Terrace Heights | 2012-07-24 | 2012-08-12 | 19 | 213 | 112 | 3.2 | 0.03 | 56.6 | 50.6 | 20.9 | 3.7 | 0.03 | 62.4 | 52.9 | 20.2 | 0.73 | |
| 11 | Terrace Heights | 2012-08-23 | 2012-09-10 | 18 | 288 | 103 | 2.5 | 0.02 | 62.9 | 56.6 | 23.6 | 1.4 | 0.01 | 68.7 | 57.5 | 21.4 | 0.64 | 1.1 |
| 12 | Terrace Heights | 2012-10-01 | 2012-10-10 | 9 | 128 | 39 | 1.4 | 0.04 | 59.3 | 54.1 | 23.0 | 0.8 | 0.02 | 65.9 | 56.5 | 21.9 | 0.51 | 1.2 |
| 13 | Terrace Heights | 2012-10-19 | 2012-10-25 | 6 | | | | | | | | | | | | | | 1.8 |
| 14 | Fort Saskatchew an | 2013-04-14 | 2013-04-28 | 15 | 115 | 37 | 4.3 | 0.11 | 51.8 | 46.0 | 18.8 | 3.0 | 0.09 | 67.7 | 60.1 | 24.6 | 0.75 | 5.2 |
| 15 | Fort Saskatchew an | 2013-05-02 | 2013-06-03 | 32 | 108 | 39 | 9.6 | 0.26 | 58.0 | 53.2 | 22.8 | 4.8 | 0.13 | 64.9 | 56.8 | 22.8 | 0.84 | 14.9 |
| 16 | Fort Saskatchew an | 2013-06-14 | 2013-07-11 | 27 | 151 | 44 | 1.9 | 0.06 | 48.4 | 39.2 | 13.8 | 2.4 | 0.07 | 64.4 | 53.2 | 19.4 | 0.74 | 9.1 |
| 17 | Fort Saskatchew an | 2013-07-19 | 2013-09-06 | 49 | 223 | 76 | 4.4 | 0.06 | 60.2 | 50.6 | 19.0 | 4.5 | 0.06 | 65.8 | 55.0 | 20.5 | 0.74 | 4.4 |
| 18 | Fort Saskatchew an | 2013-09-29 | 2014-01-19 | 113 | 107 | 31 | 7.3 | 0.24 | 69.5 | 60.3 | 23.8 | 3.3 | 0.05 | 63.2 | 52.5 | 19.3 | 0.33 | 1.1 |

*Sum of upstream and dow nstream NO$_2$ filters w here breakthrough <15%. Sum of NO$_2$ and NO filters w here there w as no dow nstream filter to check for breakthrough (samples in italics).

**Table SM-2. Sampling sites, duration of sampling, and results obtained for precipitation.**

| # | Site | Sampling duration | | | w-NO$_3$ | | | |
| | | Deployment | End | Days | Concentration (mg N/L) | $\delta^{18}$O (‰) | $\delta^{17}$O (‰) | $\Delta^{17}$O (‰) |
|---|---|---|---|---|---|---|---|---|
| 4 | Genesee | 2010-11-14 | 2011-01-06 | 49 | 0.18 | 69.2 | 64.3 | 28.0 |
| 4 | Genesee | 2011-01-06 | 2011-01-24 | 19 | 0.27 | 74.4 | 69.2 | 30.1 |
| 5 | Genesee | 2011-02-04 | 2011-04-28 | 79 | 0.48 | 57.4 | 49.3 | 19.2 |
| 6 | Genesee | 2011-04-28 | 2011-06-21 | 58 | 0.17 | 63.3 | 57.4 | 24.2 |
| 10 | Terr. Heights | 2012-07-24 | 2012-08-12 | 19 | 0.28 | 66.8 | 59.4 | 24.3 |
| 11 | Terr. Heights | 2012-08-23 | 2012-09-10 | 18 | 0.15 | 67.6 | 59.9 | 24.4 |
| 15 | F. Saskachewan | 2013-05-02 | 2013-06-05 | 33 | 0.46 | 60.0 | 55.1 | 23.6 |
| 16 | F. Saskachewan | 2013-06-14 | 2013-07-11 | 27 | 0.30 | 63.8 | 52.7 | 19.2 |
| 18 | F. Saskachewan | 2013-09-20 | 2014-01-20 | 122 | | 71.4 | 63.1 | 25.6 |

**Table SM-3. Sampling sites and other measured parameters.**

| sample | Site | Nearest air quality data site(s) | Distance from sampler (km) | Bearing from sampler | Mean T (°C) | Wind speed (km/h) | Wind direction (°) | RH (%) | $O_3$ (ppb) | $SO_2$ (ppb) | $PM_{2.5}$ (µg/m³) | Nearest source (km) |
|---|---|---|---|---|---|---|---|---|---|---|---|---|
| 1 | Genesee | | | | 11.7 | 15.8 | 316 | 62.9 | 15.5 | 4.3 | 4.9 | 7 |
| 2 | Genesee | | | | 12.2 | 16.2 | 304 | 43.5 | 26.1 | 1.8 | 3.3 | 7 |
| 3 | Genesee | Genesee | 5 | W | 5.5 | 9.4 | 308 | 64.3 | 18.7 | 1.5 | 2.9 | 7 |
| 4 | Genesee | | | | -9.8 | 13.9 | 313 | 76.1 | 16.3 | 1.0 | 4.7 | 7 |
| 5 | Genesee | | | | -0.9 | 7.7 | 298 | 62.6 | 36.6 | | | 7 |
| 6 | Genesee | | | | 12.2 | 16.3 | 312 | 64.8 | 27.4 | 2.2 | 5.0 | 7 |
| 7 | Vauxhall | | | | 2.6 | | | | | | | n/a |
| 8 | Vauxhall | Lethbridge | 65 | SW | -0.7 | - | - | - | - | - | - | n/a |
| 9 | Vauxhall | | | | -3.5 | | | | | | | n/a |
| 10 | Terrace Heights | | | | 20.3 | 11.0 | 276 | 60.3 | 25.5 | 1.1 | 7.5 | 4 |
| 11 | Terrace Heights | Edmonton East | 4.5 | E | 15.6 | 13.9 | 277 | 60.6 | 19.7 | 1.1 | 7.2 | 4 |
| 12 | Terrace Heights | | | | 7.9 | 11.3 | 313 | 63.3 | 21.1 | 0.4 | 1.6 | 4 |
| 13 | Terrace Heights | | | | -1.8 | 10.2 | 309 | 66.3 | 15.2 | 0.8 | 0.9 | 4 |
| 14 | Fort Saskatchew an | | | | 4.3 | 10.8 | 298 | | 42.5 | 0.2 | 2.3 | 3 |
| 15 | Fort Saskatchew an | Ross Creek; Fort Saskatchew an (O3) | 4.3; 6.1 | W | 15.7 | 8.8 | 300 | | 37.8 | 0.3 | 4.0 | 3 |
| 16 | Fort Saskatchew an | | | | 16.3 | 8.6 | 293 | | 25.5 | 0.4 | 5.7 | 3 |
| 17 | Fort Saskatchew an | | | | 17.7 | 6.3 | 302 | | 22.1 | 0.5 | 5.2 | 3 |
| 18 | Fort Saskatchew an | Range Rd 220 | 4.0 | N | -8.1 | 10.3 | 351 | | 18.6 | 0.8 | 5.8 | 9 |

**A**

[Figure]

**B**

[Figure]

**Figure SM-1. Monthly total precipitation (A) and mean temperature (B) for the Vauxhall region (feedlots and gas compressors), great Edmonton area (CFPP, chemical and metal industries, city traffic, fertilizers and oil refinery), and oil sands mining lower Athabasca region, recorded over the period of sampling. The Edmonton area and Vauxhall meteorological conditions only differ from the oil sands ones by having higher winter temperature.**

[Figure]

**Figure SM-2.** Twelve-hour HYSPLIT back trajectories during sample 6 (at Genesee), with red trajectories showing the times when winds were from the sampled sector.

[Figure]

**Figure SM-3. Triple oxygen isotopic results (‰) obtained for simultaneously sampled atmospheric HNO₃ (empty symbols), w-NO₃⁻ (crosses) and p-NO₃⁻ (full symbols) downwind of the various sources.**

[Figure]

**Figure SM-4.** Oxygen isotopic variations as a function of sampling period labelled by emitter types: p-NO₃⁻ (A) $\Delta^{17}O$ and (B) $\delta^{18}O$ values; and HNO₃ (C) $\Delta^{17}O$ and (D) $\delta^{18}O$ values.

[Figure]

**Figure SM-5.** Triple oxygen isotopic (‰) results for HNO₃ from Southern and central Alberta (solid symbols) and p-NO₃⁻ (empty symbols) for High Arctic (Morin et al., 2008).

**Supplementary Material – ACP 1103**

**The $\Delta^{17}O$ and $\delta^{18}O$ values of atmospheric nitrates simultaneously collected downwind of anthropogenic sources – Implications for polluted air masses**

5  Martine M. Savard[1*], Amanda Cole[2], Robert Vet[2], Anna Smirnoff[1]

[1] Geological Survey of Canada (Natural Resources Canada), 490 de la Couronne, Québec (QC), G1K 9A9, Canada
[2] Air Quality Research Division, Environment and Climate Change Canada, 4905 Dufferin St., Toronto (ON), M3H 5T4, Canada

*Correspondence to*: Martine M. Savard (martinem.savard@canada.ca)

**Table SM-1. Sampling sites, duration of sampling, average results obtained for the parallel sampling with the CAPMoN systems.**

| # | Site | Deployment | Collection | Days | Sampling hours | Mean (total for pooled) air volume (m³) | p-NO₃⁻ Mean (total for pooled) loading (ug N) | Air conc (ug N/m³) | $\delta^{18}O$ (‰) | $\delta^{17}O$ (‰) | $\Delta^{17}O$ (‰) | HNO₃ Mean (total for pooled) loading (ug N) | Air conc (ug N/m3) | $\delta^{18}O$ (‰) | $\delta^{17}O$ (‰) | $\Delta^{17}O$ (‰) | Daylight fraction (sunrise to sunset) | NO₂ in sample train* (ug N/m³) |
|---|------|-----------|-----------|------|----------------|------------------------------------------|-------|------|------|------|------|-------|------|------|------|------|------|------|
| 1 | Genesee | 2010-09-30 | 2010-10-05 | 5 | 23 | 59 | 8.0 | 0.14 | 70.3 | 63.8 | 26.9 | 8.4 | 0.15 | 66.9 | 59.4 | 24.3 | 0.49 | |
| 2 | Genesee | 2010-10-08 | 2010-10-18 | 10 | 42 | 99 | | | | | | 6.6 | 0.07 | 63.3 | 56.1 | 22.9 | 0.65 | |
| 3 | Genesee | 2010-10-18 | 2010-11-08 | 21 | 39 | 87 | 13.4 | 0.15 | 62.1 | 55.9 | 23.3 | 10.2 | 0.12 | 69.2 | 61.1 | 24.8 | 0.33 | |
| 4 | Genesee | 2010-11-08 | 2011-01-31 | 84 | 360 | 517 | 181.6 | 0.35 | 83.2 | 74.2 | 30.5 | 40.2 | 0.08 | 81.7 | 71.9 | 29.0 | 0.31 | |
| 5 | Genesee | 2011-02-22 | 2011-04-28 | 65 | | | | | | | | | | | | | | |
| 6 | Genesee | 2011-04-28 | 2011-06-20 | 53 | 144 | 189 | 19.4 | 0.10 | 67.3 | 60.9 | 25.6 | 24.9 | 0.13 | 67.6 | 60.2 | 24.7 | 0.71 | *0.82* |
| 7 | Vauxhall | 2011-10-25 | 2011-11-17 | 23 | 152 | 255 | 22.1 | 0.09 | 64.8 | 61.3 | 27.3 | 6.9 | 0.03 | 68.3 | 61.4 | 25.5 | 0.35 | *0.05* |
| 8 | Vauxhall | 2011-11-17 | 2011-12-01 | 14 | 89 | 176 | 15.9 | 0.09 | 60.7 | 57.0 | 25.1 | 4.8 | 0.03 | 61.7 | 55.9 | 23.5 | 0.28 | *0.12* |
| 9 | Vauxhall | 2011-12-01 | 2011-12-13 | 12 | 128 | 235 | 31.1 | 0.13 | 63.9 | 60.2 | 26.7 | | | | | | 0.34 | |
| 10 | Terrace Heights | 2012-07-24 | 2012-08-12 | 19 | 213 | 112 | 3.2 | 0.03 | 56.6 | 50.6 | 20.9 | 3.7 | 0.03 | 62.4 | 52.9 | 20.2 | 0.73 | |
| 11 | Terrace Heights | 2012-08-23 | 2012-09-10 | 18 | 288 | 103 | 2.5 | 0.02 | 62.9 | 56.6 | 23.6 | 1.4 | 0.01 | 68.7 | 57.5 | 21.4 | 0.64 | 1.1 |
| 12 | Terrace Heights | 2012-10-01 | 2012-10-10 | 9 | 128 | 39 | 1.4 | 0.04 | 59.3 | 54.1 | 23.0 | 0.8 | 0.02 | 65.9 | 56.5 | 21.9 | 0.51 | 1.2 |
| 13 | Terrace Heights | 2012-10-19 | 2012-10-25 | 6 | | | | | | | | | | | | | | 1.8 |
| 14 | Fort Saskatchew an | 2013-04-14 | 2013-04-28 | 15 | 115 | 37 | 4.3 | 0.11 | 51.8 | 46.0 | 18.8 | 3.0 | 0.09 | 67.7 | 60.1 | 24.6 | 0.75 | 5.2 |
| 15 | Fort Saskatchew an | 2013-05-02 | 2013-06-03 | 32 | 108 | 39 | 9.6 | 0.26 | 58.0 | 53.2 | 22.8 | 4.8 | 0.13 | 64.9 | 56.8 | 22.8 | 0.84 | 14.9 |
| 16 | Fort Saskatchew an | 2013-06-14 | 2013-07-11 | 27 | 151 | 44 | 1.9 | 0.06 | 48.4 | 39.2 | 13.8 | 2.4 | 0.07 | 64.4 | 53.2 | 19.4 | 0.74 | 9.1 |
| 17 | Fort Saskatchew an | 2013-07-19 | 2013-09-06 | 49 | 223 | 76 | 4.4 | 0.06 | 60.2 | 50.6 | 19.0 | 4.5 | 0.06 | 65.8 | 55.0 | 20.5 | 0.74 | 4.4 |
| 18 | Fort Saskatchew an | 2013-09-29 | 2014-01-19 | 113 | 107 | 31 | 7.3 | 0.24 | 69.5 | 60.3 | 23.8 | 3.3 | 0.05 | 63.2 | 52.5 | 19.3 | 0.33 | 1.1 |

*Sum of upstream and dow nstream NO₂ filters w here breakthrough <15%. Sum of NO₂ and NO filters w here there w as no dow nstream filter to check for breakthrough (samples in italics).

**Table SM-2. Sampling sites, duration of sampling, and results obtained for precipitation.**

| # | Site | Sampling duration Deployment | End | Days | w-NO$_3$ Concentration (mg N/L) | δ$^{18}$O (‰) | δ$^{17}$O (‰) | Δ$^{17}$O (‰) |
|---|------|------------|-----|------|---------------|------|------|------|
| 4 | Genesee | 2010-11-14 | 2011-01-06 | 49 | 0.18 | 69.2 | 64.3 | 28.0 |
| 4 | Genesee | 2011-01-06 | 2011-01-24 | 19 | 0.27 | 74.4 | 69.2 | 30.1 |
| 5 | Genesee | 2011-02-04 | 2011-04-28 | 79 | 0.48 | 57.4 | 49.3 | 19.2 |
| 6 | Genesee | 2011-04-28 | 2011-06-21 | 58 | 0.17 | 63.3 | 57.4 | 24.2 |
| 10 | Terr. Heights | 2012-07-24 | 2012-08-12 | 19 | 0.28 | 66.8 | 59.4 | 24.3 |
| 11 | Terr. Heights | 2012-08-23 | 2012-09-10 | 18 | 0.15 | 67.6 | 59.9 | 24.4 |
| 15 | F. Saskachewan | 2013-05-02 | 2013-06-05 | 33 | 0.46 | 60.0 | 55.1 | 23.6 |
| 16 | F. Saskachewan | 2013-06-14 | 2013-07-11 | 27 | 0.30 | 63.8 | 52.7 | 19.2 |
| 18 | F. Saskachewan | 2013-09-20 | 2014-01-20 | 122 | | 71.4 | 63.1 | 25.6 |

**Table SM-3. Sampling sites and other measured parameters.**

| sample | Site | Nearest air quality data site(s) | Distance from sampler (km) | Bearing from sampler | Mean T (°C) | Wind speed (km/h) | Wind direction (°) | RH (%) | $O_3$ (ppb) | $SO_2$ (ppb) | $PM_{2.5}$ (µg/m³) | Nearest source (km) |
|---|---|---|---|---|---|---|---|---|---|---|---|---|
| 1 | Genesee | | | | 11.7 | 15.8 | 316 | 62.9 | 15.5 | 4.3 | 4.9 | 7 |
| 2 | Genesee | | | | 12.2 | 16.2 | 304 | 43.5 | 26.1 | 1.8 | 3.3 | 7 |
| 3 | Genesee | Genesee | 5 | W | 5.5 | 9.4 | 308 | 64.3 | 18.7 | 1.5 | 2.9 | 7 |
| 4 | Genesee | | | | -9.8 | 13.9 | 313 | 76.1 | 16.3 | 1.0 | 4.7 | 7 |
| 5 | Genesee | | | | -0.9 | 7.7 | 298 | 62.6 | 36.6 | | | 7 |
| 6 | Genesee | | | | 12.2 | 16.3 | 312 | 64.8 | 27.4 | 2.2 | 5.0 | 7 |
| 7 | Vauxhall | | | | 2.6 | | | | | | | n/a |
| 8 | Vauxhall | Lethbridge | 65 | SW | -0.7 | - | - | - | - | - | - | n/a |
| 9 | Vauxhall | | | | -3.5 | | | | | | | n/a |
| 10 | Terrace Heights | | | | 20.3 | 11.0 | 276 | 60.3 | 25.5 | 1.1 | 7.5 | 4 |
| 11 | Terrace Heights | Edmonton East | 4.5 | E | 15.6 | 13.9 | 277 | 60.6 | 19.7 | 1.1 | 7.2 | 4 |
| 12 | Terrace Heights | | | | 7.9 | 11.3 | 313 | 63.3 | 21.1 | 0.4 | 1.6 | 4 |
| 13 | Terrace Heights | | | | -1.8 | 10.2 | 309 | 66.3 | 15.2 | 0.8 | 0.9 | 4 |
| 14 | Fort Saskatchew an | | | | 4.3 | 10.8 | 298 | | 42.5 | 0.2 | 2.3 | 3 |
| 15 | Fort Saskatchew an | Ross Creek; Fort Saskatchew an (O3) | 4.3; 6.1 | W | 15.7 | 8.8 | 300 | | 37.8 | 0.3 | 4.0 | 3 |
| 16 | Fort Saskatchew an | | | | 16.3 | 8.6 | 293 | | 25.5 | 0.4 | 5.7 | 3 |
| 17 | Fort Saskatchew an | | | | 17.7 | 6.3 | 302 | | 22.1 | 0.5 | 5.2 | 3 |
| 18 | Fort Saskatchew an | Range Rd 220 | 4.0 | N | -8.1 | 10.3 | 351 | | 18.6 | 0.8 | 5.8 | 9 |

**A**

[Figure]

**B**

[Figure]

**Figure SM-1. Monthly total precipitation (A) and mean temperature (B) for the Vauxhall region (feedlots and gas compressors), great Edmonton area (CFPP, chemical and metal industries, city traffic, fertilizers and oil refinery), and oil sands mining lower Athabasca region, recorded over the period of sampling. The Edmonton area and Vauxhall meteorological conditions only differ from the oil sands ones by having higher winter temperature.**

[Figure]

**Figure SM-2.** Twelve-hour HYSPLIT back trajectories during sample 6 (at Genesee), with red trajectories showing the times when winds were from the sampled sector.

[Figure]

**Figure SM-3. Triple oxygen isotopic results (‰) obtained for simultaneously sampled atmospheric HNO₃ (empty symbols), w-NO₃⁻ (crosses) and p-NO₃⁻ (full symbols) downwind of the various sources.**

[Figure]

**Figure SM-4.** Oxygen isotopic variations as a function of sampling period labelled by emitter types: p-NO$_3^-$ (A) $\Delta^{17}$O and (B) $\delta^{18}$O values; and HNO$_3$ (C) $\Delta^{17}$O and (D) $\delta^{18}$O values.

[Figure]

**Figure SM-5. Triple oxygen isotopic (‰) results for HNO₃ from Southern and central Alberta (solid symbols) and p-NO₃⁻ (empty symbols) for High Arctic (Morin et al., 2008).**